# Navigating the Labyrinth: Evaluating and Enhancing LLMs' Search Problems Reasoning Abilities

## Abstract

Recently, Large Language Models (LLMs) attained impressive performance in math and reasoning benchmarks. However, they still often struggle with multi-step reasoning which is relatively easy for humans. To further investigate this, we introduce a new benchmark, SearchBench, containing 11 unique combinatorial problems that avoid training contamination (each equipped with automated pipelines to generate an arbitrary number of instances) and analyze the feasibility, correctness, and optimality of LLM-generated solutions. We show that even the most advanced LLMs fail to solve these problems end-to-end in text, e.g., GPT4 and o1-preview respectively solve only 1.4% and 18.6% correctly. SearchBench problems require considering multiple pathways to the solution and backtracking, posing a significant challenge to auto-regressive models. Instructing LLMs to generate code that solves the problem helps only slightly. We next introduce an in-context learning approach that prompts the model to implement A*, an informed search algorithm, to comprehensively traverse the problem state space, improving the performance of models. We further extend this approach and propose the Multi-Stage-Multi-Try inference method which breaks down the A* algorithm implementation into two stages and auto-verifies the first stage against unit tests, raising GPT-4's performance above 57%.

## 1 Introduction

The advent of Large Language Models (LLMs) has revolutionized the field of natural language processing, with models like Llama3.1 (Meta, 2024), GPT4 (OpenAI, 2023), and o1-preview (OpenAI, 2024) demonstrating unprecedented performance on math and science QA benchmarks, such as GSM8k (Cobbe et al., 2021) and GPQA (Rein et al., 2023). However, LLMs still exhibit surprising failures on some intuitive tasks (Bian et al., 2023; Qin et al., 2023; Marcus, 2020) and struggle with multi-step compositional reasoning, combinatorial problems, and planning (Dziri et al., 2024; Valmeekam et al., 2022; Wu et al., 2023). Inspired by these observations and to further investigate LLMs' reasoning abilities, we offer a new benchmark of search problems, SearchBench. SearchBench is designed to evaluate the general reasoning capabilities of language models in performing search and backtracking to previous search states as part of it.

SearchBench is challenging to LLMs due to several factors. Current LLMs have an autoregressive architecture that forces them to solve problems sequentially, posing challenges for tasks that require backtracking (Dziri et al., 2024). Moreover, natural language is less suited for describing or updating accurate representations of complex intermediate states. Lastly, the number of feasible states in combinatorial problems grows exponentially with its size. Our empirical results show that even the most capable models can solve less than 20% of SearchBench problems. In order to successfully solve SearchBench, a model must backtrack to correct errors, consider multiple chains of reasoning, and determine the most optimal outcome among the many feasible options. These capabilities are required for robust reasoning, making SearchBench a valuable benchmark for evaluating LLM reasoning capabilities as they continue to evolve.

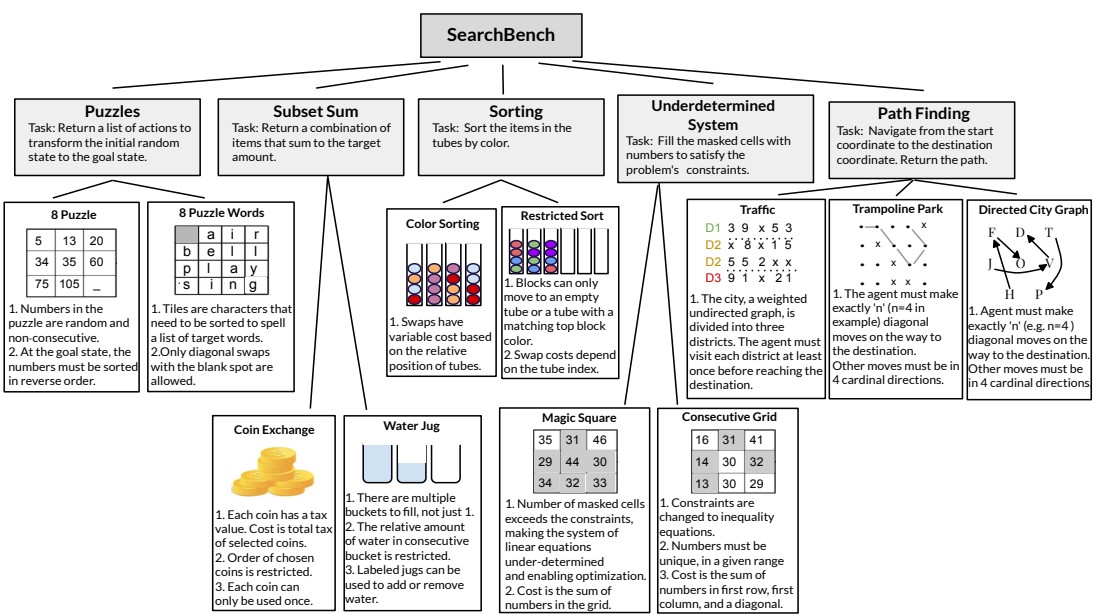

Figure 1: The taxonomy of SearchBench. The five nodes in level one represent the problem categories, and the 11 nodes in level two represent the problem types. We detail how the rules of known puzzles and combinatorial problems are modified in SearchBench to ensure that LLMs have not encountered a solved instance of the problem during their massive training.

SearchBench has five problem categories: (i) pathfinding, (ii) puzzles, (iii) subset sum, (iv) sorting, and (v) under-determined systems; further divided into 11 unique problem types. Each problem type is inspired by known puzzles and combinatorial problems but augmented with modified rules to ensure substantial differences from similar problems LLMs encountered during their training. We generate ~100 instances of varying difficulty per problem type using an automatic pipeline, resulting in 1107 fixed problem instances in total. Each problem type in SearchBench is equipped with an automatic pipeline that evaluates LLM-generated solutions on three dimensions: feasibility, correctness, and optimality. Feasibility checks whether the actions taken follow the problem's rules; correctness verifies if a feasible solution reaches the goal state; and optimality checks if the least cost solution was found.

To alleviate backtracking bottleneck of LLMs, we introduce A* prompting that uses code execution to find the solution. This method offloads some of the non-linear computations involved in searching the state-space from the model. Using this method, the task of the model is changed to devising a A* search strategy (Wikipedia, a), which has advantages over other search algorithms that are either computationally inefficient (BFS) (Wikipedia, b) or do not guarantee an optimal solution (DFS) (Wikipedia, c).

However, implementing the A* algorithm is complex and involves creating a correct search strategy and coding it without any errors. Our experiments show that the model often makes coding mistakes, such as syntax errors and type errors, with this approach (see Fig. 5). Recent work (Wang et al., 2022; Yao et al., 2023a; Long, 2023) shows that multiple inferences helps reduce LLM errors, and thus, we present the Multi-Stage-Multi-Try (MSMT) inference strategy. In this approach, we decompose code generation into two steps. First, we prompt the model to write a general A* algorithm for the problem type. Here, we verify the the A* implementation against a set of unit tests: (i) the code is executable; (ii) it returns a list as output; and (iii) data type of list elements is correct. Second, we instruct the model to implement the 'initialize'

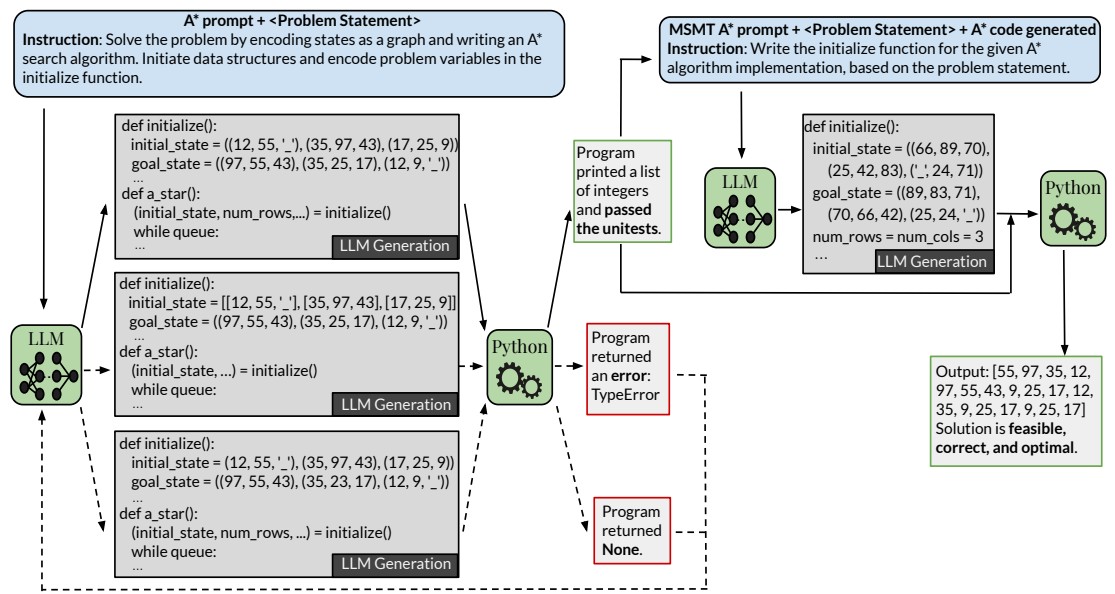

Figure 2: Our Multi-Stage-Multi-Try (MSMT) A* prompting approach.

function which encodes the conditions and state variables specific to each problem instance. Our MSMT A* method (Fig. 2) significantly enhances the LLMs' ability to solve search problems, outperforming all other prompting strategies, including 0-shot text, 4-shot Chain-of-Thought (CoT) (Wei et al., 2022) text, 0-shot code generation, and 4-shot A* prompting with the naive greedy decoding strategy.

To summarize, our main contributions are as follows: (i) We contribute the SearchBench benchmark designed to assess the capability of LLMs in solving state-based problems requiring combinatorial search; (ii) We introduce the A* Multi-Stage-Multi-Try (MSMT) framework that leverages the flexibility and natural language comprehension capabilities of LLMs, reliability and structured nature of the A* algorithm, along with unitests to verify LLMs' code generation in order to address search problems. (iii) We show a strong advantage of MSMT A* prompting for solving search problems robustly with LLMs, suggesting interesting future work directions and yet leaving significant room for future improvement on SearchBench.

## 2  SEARCHBENCH BENCHMARK

SearchBench includes five categories of problems: puzzles, subset sum, sorting, pathfinding, and under-determined systems. In theoretical computer science, combinatorial problems are classified into four types: existence, construction, enumeration, and optimization problems  (Wilson, 2016). To ensure broad representation, we selected one problem category from each of these types for SearchBench. Particularly, subset sum problems represent the existence category, where the task is to determine if a subset of a given set sums to a specified value (refer to Tab. 1 for an example problem in this category). The 8-puzzle and 8-puzzle words fall under construction problems, which involve solving puzzles. Sorting problems, such as color sort and restricted sorting, are enumeration problems. Pathfinding problems are categorized as optimization problems.

Additionally, we introduce a new category of NP-hard combinatorial problmes in SearchBench, under-determined system problems. These problems consist of constraint satisfaction problems which are typically solved by defining a system of linear equations, and do not require search over states. We modified them

to include fewer constraints than unknown variables, allowing for multiple correct solutions, and defined a unique cost function to enable search for a single optimal solution. This category was added in order to evaluate models' ability to generalize to novel combinatorial problems.

We selected 2-3 problem types for each category, resulting in 11 total problem types. Each type has a unique state space. For example, in 8-puzzle words, each state is an $n \times m$ table of characters, while in coin exchange, each state is an ordered subset of given coins (See Appendix sec. G for more examples). Generally, our problems involve an initial state, a goal state, and a set of possible actions, and the task is to find a sequence of actions from the initial to the goal state with minimum cost. We modified the rules to ensure that solved instances of SearchBench were not encountered during the LLMs' massive internet-scale training. The SearchBench taxonomy and rule modifications are illustrated in Fig. 1.

To construct SearchBench, we implemented an automatic generation pipeline for each problem type, ensuring each generated instance is solvable. We generated approximately 100 instances per type, resulting in a total of 1107 problem instances. The benchmark is then fixed. The generation pipelines can create instances with adjustable difficulty levels. Difficulty is defined by the state space size of the instance, with minimum difficulty requiring a few actions and maximum difficulty set such that problems could be solved correctly but not optimally by humans (See Appendix Sec. F for an analysis of the search space size). Hence, maximum human performance on SearchBench could be considered approximately 100%. Moreover, studies like Pizlo & Li (2005); Chronicle et al. (2006) show that humans can solve the classic versions of SearchBench problems, but their performance declines as the state space size increases.

In contrast to other reasoning benchmarks (Saparov & He, 2022; Cobbe et al., 2021; Hendrycks et al., 2021; Patel et al., 2021; Clark et al., 2020; Tafjord et al., 2020; Sap et al., 2019; Le et al., 2019) that only measure correctness, to gain a more comprehensive understanding of LLM performance on SearchBench, our evaluation pipeline assesses LLM solutions across 3 dimensions: Feasibility, Correctness, and Optimality. Feasibility determines if any of the actions chosen violate the problem rules (e.g. passing through labyrinth walls). Correctness requires that the solution is both feasible and reaches the goal state from the given start state. Optimality indicates that the solution is both correct and has the minimum cost w.r.t. known optimum. For each SearchBench problem, we implemented a fast A* algorithm with a provably admissible and consistent heuristic, to produce the optimal solution. We ran this implementation for each instance in the benchmark to obtain its unique optimal solution.

We note that even though correctness implies feasibility, and optimally implies correctness, feasibility and correctness are valuable intermediate metrics in determining how close the models are to generating the fully correct solution. For example, in traffic problems, GPT-4 often fails to record the first city visited, resulting in a feasible but incorrect solution. Defining feasibility helps distinguish this mostly correct implementation from more erroneous solutions. Correctness is stricter than feasibility and indicates that search-related tasks were implemented correctly, but the heuristic or recorded cost is incorrect, leading to non-optimal solutions.

## 3 EVALUATED METHODS

We use the following 3 baseline prompting methods to evaluate LLMs on SearchBench: 0-shot text, 4-shot CoT text, and 0-shot code. Additionally, we introduce two new code-based methods: 4-shot A* prompting and MSMT A*. The full prompts for each of the 5 approaches and GPT-4's responses for an example problem in SearchBench are provided in Appendix Sec. H.

To ensure the generality of our prompting methods, we selected one in-context example from each of the four SearchBench categories that are different from the category of the evaluated problem. This minimizes the similarity between the rules and context of the solved examples and the evaluated problem, and tests whether the model can solve unrelated problems. Thus, if a model finds an optimal solution using these methods, it demonstrates true generalization rather than prompt-specific improvements. In Sec. 6, we further analyze the

Table 1: An instance of the 'Coin Exchange' problem shown to LLMs. The green indicates instance-specific components, and the orange represents modifications to the classic problem rules specific to SearchBench. GPT-4 fails to generate a feasible solution using baseline methods (0-shot, 4-shot CoT text, 0-shot code) but produces a correct, non-optimal code solution using A* and MSMT A*.

| Problem statement |
|---|
| In the 'taxed coin exchange' problem, you are required to choose a subset of coins from this list [3, 6, 9, 10, 13, 15, 18, 5, 21, 19, 12, 15, 5, 9, 4, 16, 8, 4, 7, 7, 7, 2, 16, 14, 18, 3, 89, 21, 12, 10, 7, 14, 4, 11, 6, 20], such that the sum of the chosen coins adds up to 229. Each coin in the list is unique and can only be used once. Also coins carry a tax value. The tax values for each coin is 14: 1, 89: 13, 2: 2, 5: 2, 4: 4, 6: 6, 8: 2, 16: 5, 21: 4, 20: 2, 18: 9, 11: 10, 10: 3, 12: 12, 15: 5, 13: 1, 3: 1, 19: 19, 7: 7, 9: 3, where the tax for coins of the same value is the same. Also, if the coin chosen is smaller than the previous one, it must have an even value, otherwise, if the coin is larger than or equal to the previous coin chosen, it must have an odd value. The objective is to determine which subset of coins should be selected to minimize the total tax paid. The solution should be presented as a list of numbers, representing the value of the coins chosen in order, with the first coins chosen being in index 0, formatted in Python syntax. |

impact of including an example from the same top-level problem category. Additionally, 4-shot is the upper limit on the number of in-context examples due to the models' context length limit. For an analysis of the effect of fewer demonstrations (shots) on performance, see Appendix Sec. A.

**0-shot text and 4-shot CoT text prompting methods**. In the text-based prompting methods, we instruct the model to solve the problem in an end-to-end manner, using text only. In 4-shot CoT prompts, the in-context examples include a representation of the intermediate states drawn using ASCII characters after each action to prevent hallucinations and illogical leaps in reasoning.

**0-shot code prompting method**. This method instructs the LLM to produce a Python code that solves the given problem. The generated code is then executed to derive the final answer.

**A* Prompting**. In this approach, we prompt the LLM to implement an A* algorithm that solves $\mathcal{P}_i^C$ - a problem instance number $i$ of problem category $C$, providing four in-context examples of A* codes for four unrelated problems $\mathcal{P}_j^{\hat{C}}$ from different categories $\hat{C} \neq C$. To implement A* for the target SearchBench problem, the LLM must perform abstract reasoning to devise a search strategy applicable to any state within the search space. This contrasts with solving problems end-to-end in text, where the model has access to the variables of each state, eliminating the need for abstract reasoning or a general strategy. However, end-to-end approaches requires the model to perform every step of the non-linear computations involved in the search.

The in-context examples include detailed comments before each code segment, explaining the reasoning used to develop the strategy implemented within the code segment. These comments serves as CoT reasoning for devising the search strategy implemented in the code.

**Multi-Stage-Multi-Try (MSMT) A* Prompting**. In this method, the model receives the same in-context examples as the 'A* prompting', with different instructions. Here, the inference is done in two stages as demonstrated in Fig. 2. In the first stage, the model is instructed to implement the code as two functions: the 'a_star' function includes an instance-agnostic A* algorithm for the target problem type, and the 'initialize' function encodes the variables given in the problem statement. We then verify if the generated code satisfies the following set of unit tests: (i) code is executable; (ii) code returns a list; (iii) and the list elements match the data type specified by the problem statement. If the code fails any unit test, MSMT re-generate the code. Next, in the second stage, the LLM is instructed to implement an 'initialize' function, conditioned on the verified 'a_star' function from stage 1 for each instance of the problem type. The inclusion of simple unit tests, which can be expanded to more detailed tests if needed, offers a robust method for filtering out erroneous samples from the model's generations.

In our MSMT A* prompting approach, the model generates the full A* algorithm end-to-end without any external feedback, similar to how text-based prompting methods operate. Importantly, our MSMT A* does not rely on the majority vote of multiple solutions. Instead, the solution returned by the first model-generated code that passes the unit tests is taken as the final answer. This results in increased efficiency of MSMT A*, requiring only up to 1.5x number of inferences per problem on average compared to 5x-100x in majority vote approaches (Wang et al., 2022).

## 4 RELATED WORK

**Mathematical and Reasoning Benchmarks**: The evaluation of LLMs (Brown et al., 2020; OpenAI, 2023; 2022; Chung et al., 2024; Chowdhery et al., 2022; Rae et al., 2021; Taylor et al., 2022; Thoppilan et al., 2022) on mathematical and reasoning tasks has been a focus of recent research in natural language processing, leading to the development of benchmarks such as BIG-BENCH (Srivastava et al., 2022), GSM8K (Cobbe et al., 2021), AQUA (Ling et al., 2017), SVAMP (Patel et al., 2021), CommonsenseQA (Talmor et al., 2018), StrategyQA (Geva et al., 2021), and MATH (Hendrycks et al., 2021). However, these benchmarks have limitations. For instance, GSM8K problems are relatively simple and often require a repetitive reasoning pattern to solve. The MATH dataset, while more challenging, may not accurately reflect a model's reasoning or problem-solving capabilities due to the advanced mathematical skills required. Tasks in BIG-BENCH are mostly single-step reasoning tasks that don't challenge models to combine multiple steps for solving complex compositional problems. When prompted to solve problems end-to-end using CoT prompting, LLMs perform well on these tasks; however, they fail on our benchmark's problems, indicating that these benchmarks offer limited insight into LLMs' ability to systematically explore a state space.

**Application of LLMs to Combinatorial Problems**: Recent work (Yang et al., 2023; Liu et al., 2024; Masoud et al., 2024; Mittal et al., 2024; Iklassov et al., 2024) has explored solving combinatorial problems using LLMs. Yang et al. (2023); Liu et al. (2024); Masoud et al. (2024) investigated prompting LLMs to solve the Traveling Salesman Problem through multiple inferences, while Mittal et al. (2024) introduced a dataset of combinatorial problems, "PuzzleBench". However, they only selected problems that can be represented in a symbolic solver (SMT2.0) and assumed there exists fixed pre-defined symbolic representations for input problems and outputs, limiting their datasets' generalizability. Moreover, problems selected by Mittal et al. (2024) and Iklassov et al. (2024) are instances of the classical combinatorial problems, raising issues of memorization as algorithm implementations for instances of such problems are often available online.

SearchBench stands out in several ways (i) Generalizability: Unlike PuzzleBench, SearchBench problems are described only in natural language, with no restrictions on rules or actions, ensuring that a model capable of solving SearchBench can generalize to other combinatorial problems. (ii) Uniquely Modified Rules: This prevents memorization, as algorithms for classic versions of the problem are available online. (iii)Optimal Solutions: Each problem type has a uniquely defined cost, ensuring a single optimal solution and avoiding multiple valid answers. (iv)Multi-Dimensional Evaluation: This provides deeper insights into how close models are to deriving the unique optimal solution. (v)Automated Instance Generation: This avoids data leakage or contamination, as new instances can be generated on demand.

**Prompting and Inference Strategies**: Sophisticated prompting strategies have been developed to enhance models' reasoning abilities. One notable approach is Chain-of-Thought (CoT) prompting (Wei et al., 2022), which prompts LLMs to generate intermediate steps leading to the final output. This technique has led to advanced variations, including Tree-of-Thoughts (Yao et al., 2023a; Long, 2023), and Graph-of-Thought (Yao et al., 2023b; Lei et al., 2023; Besta et al., 2023) methods that maintain a tree of intermediate generations a to enable systematic exploration of "thoughts". However, these methods rely on evaluating and rejecting intermediate steps, which does not integrate well with our problems. In search problems, intermediate states can't be easily classified as correct or incorrect, and all possible actions must be considered to find the optimal

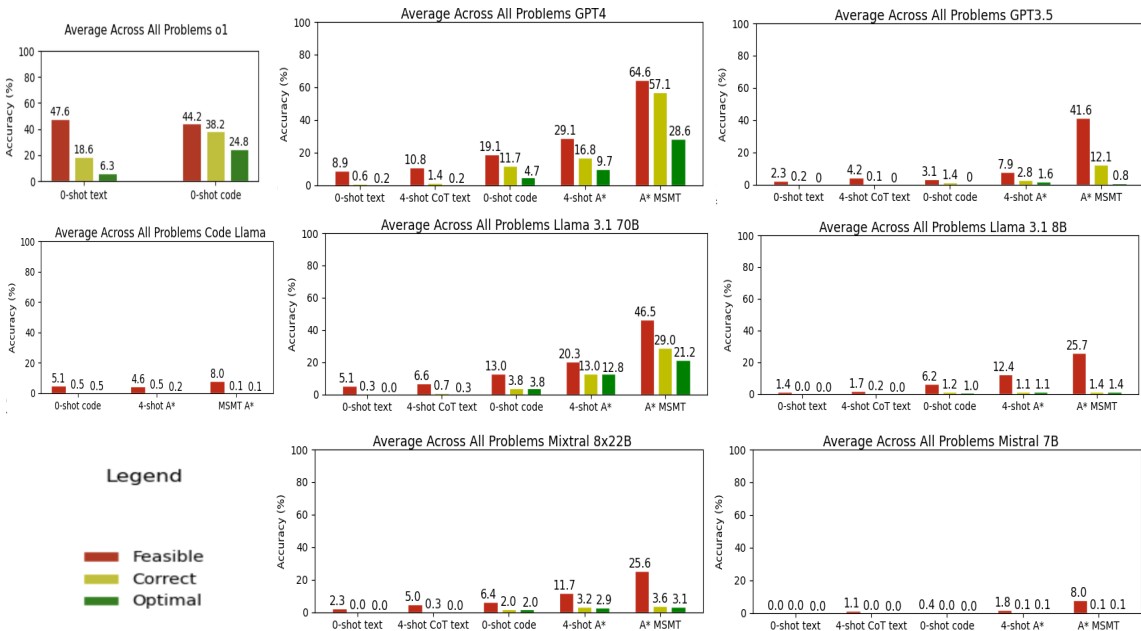

Figure 3: Average rate of feasible, correct, and optimal solutions for all problems using o1, GPT4, GPT3.5, Code Llama, Llama 3.1 70B, Llama 3.1 8B, Mixtral 8x22B, and Mistral 7B.

solution. Additionally, the state space of combinatorial problems grows exponentially, making it impractical for LLMs to navigate the frontier of the search tree without incorrectly disregarding most feasible states.

Other prompting methods, such as Decomposition strategies (Khot et al., 2022; Zhou et al., 2022; Zhang et al., 2023), simplify complex tasks into smaller, manageable subtasks using symbolic programs or structured algorithms. Additionally, systems like LLM-Augmenter (Peng et al., 2023) rely on external databases to consolidate evidence and verify segments of the LLM's output.

In this work, we propose the A* prompting strategy, where we prompt the model to solve problems by implementing a unique A* algorithm. Similarly, our A* MSMT approach decomposes the task of implementing the search algorithm into two stages and checks the model's generations against external validators; we use simple unit tests instead of external data sources or solved solution instances in our work.

## 5 EXPERIMENTS

We evaluated the performance of GPT-4, GPT-3.5, and Code Llama Instruct 34B (Roziere et al., 2023) [1], Llama 3.1 70B, Llama 3.1 8B, Mixtral 8x22B (Mistral, 2023b), and Mistral 7B (Mistral, 2023a) on SearchBench, using the following five prompting methods described in Sec. 3: 0-shot text, 4-shot CoT text, 0-shot code, 4-shot A*, and 4-shot MSMT A*. Results are summarized in Fig. 3.

**Implementation details**. GPT-4, GPT-3.5 Turbo (GPT3.5 hereafter), and o1-preview (o1 hereafter) were used through official Open-AI platform APIs. For all code evaluation experiments we used a machine with 96 64-bit Intel Xeon Gold 5220R CPUs with a maximum speed of 4GHz, and 71.5 MiB Level 3 cache.

---

[1] Finetuned on the Phind dataset (Phind)

**0-shot text and 4-shot CoT text prompting methods**. As shown in Fig. 3, the correct solutions rate is below 1% for all of the models using 0-shot text prompting, and less than 9% of GPT4 solutions are feasible (follow the problem rules) using this method. This is expected as the exponentially growing state space size of SearchBench problems and the difficulty of backtracking during auto-regressive generation make it challenging to solve SearchBench problems using text-based prompting, even with the strongest LLMs.

Moreover, 4-shot CoT text prompting only improves the rate of feasible solutions generated by less than 3% for all models, with almost no performance gain for Mistral 7B. This shows that the inherent complexity of search problems from SearchBench cannot be effectively addressed by text-based prompting alone.

Finally, we also report results with the recent o1 model (OpenAI, 2024), which is designed for comprehensive reasoning and trained to decompose tasks and correct its mistakes. As can be seen, this model still struggles with SearchBench problems, solving less than 19% correctly using 0-shot text. However, it significantly outperformed other models' end-to-end performance.

**0-shot code prompting method**. This prompting method improves performance over text-based prompting for all models except Mistral 7B, which remained close to 0%. This is expected, as using Python to compute intermediate steps and execute the iterations of the algorithms devised by the LLMs reduces the load on the models. As seen in Fig. 3, o1 solved 38.2% of the problems correctly, 19.1% of GPT-4's code generations result in a feasible solution, with only 11.7% being correct. The next best performance was achieved by Llama 3.1 70B, which solved 13% of the problems correctly. For an analysis of the computation time of programs generated by the LLMs, please refer to Appendix Sec. C.

**A\* Prompting**. As shown in Fig. 3, A\* prompting improves the performance of all models on SearchBench except for Code Llama, which shows almost no improvement, indicating potential limitations of this model in in-context learning or following the given instructions. GPT-4's feasible, correct, and optimal solution rates increase by 10%, 5%, and 5%, respectively, and Llama 3.1 70B's rates increase by 7%, 9%, and 9%.

**MSMT A\***. In Fig. 3, we see that MSMT A\* prompting significantly enhances the performance of all models. With MSMT A\*, GPT-4 correctly solved 57.1% of SearchBench problems and achieved a 28.6% rate of optimal solutions, outperforming o1. The performance increase of GPT-4 was consistent across all problem types compared to other prompting strategies (See Appendix Sec. B for a detailed analysis of GPT-4's performance on each problem type). Other LLMs also showed strong improvements (except for Code Llama, which only improved in feasibility, as it still struggles to follow instructions even with MSMT A\*).

The improvement of MSMT A\* over A\* prompting shows that while LLMs have the capability to generate the correct solution, they are prone to make mistakes. MSMT unit tests help filter out erroneous samples, selecting higher-quality ones. The overall improvement in performance of LLMs promoted with MSMT A\* demonstrates that emulating a structured algorithm in models and selecting a verified sample can significantly boost LLM's problem-solving capabilities. That said, the 28.6% optimal performance, although inspiring, still leaves room for further improvements, underlining the importance of SearchBench for future research.

## 6 ABLATIONS AND ANALYSIS

Here we provide a comprehensive analysis to further investigate our SearchBench using GPT4. For additional analysis, please refer to Appendix Sec. A, B, C, and F.

**Does including a more similar problem in prompt improve GPT-4's performance?** In our main experiments with A\* and MSMT A\* (Fig. 3), we used four in-context examples, each from a different category than the target problem (Sec. 3). This ensured no segment of the target problem solution was observed by the LLM in the prompt, hence better measuring LLM's reasoning generalization. Here, we evaluated GPT4's performance when a solved instance of a SearchBench problem from the same category but a different type as the evaluated problem, is included in the prompt. Results are summarized in Fig. 4. We observed small

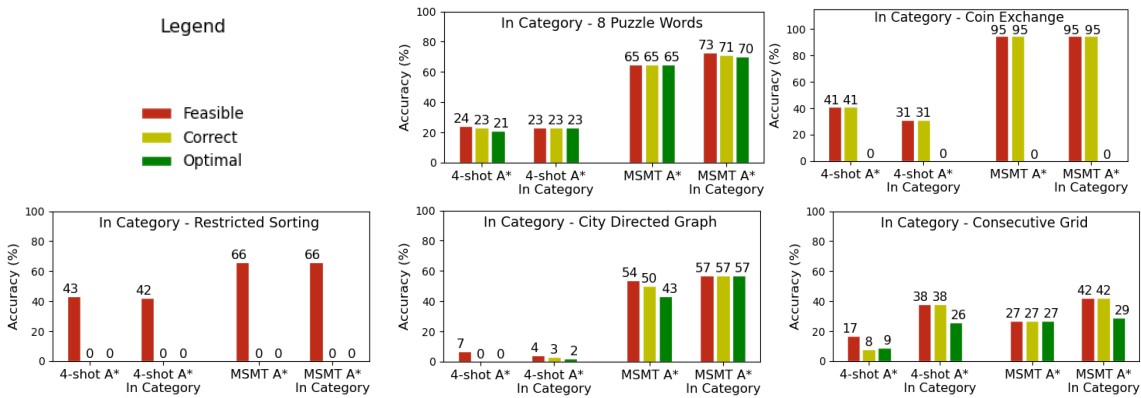

Figure 4: Comparing GPT-4's performance, using A* prompting approaches, when one of the in-context examples is switched to a problem that shares the same category as the inference problem.

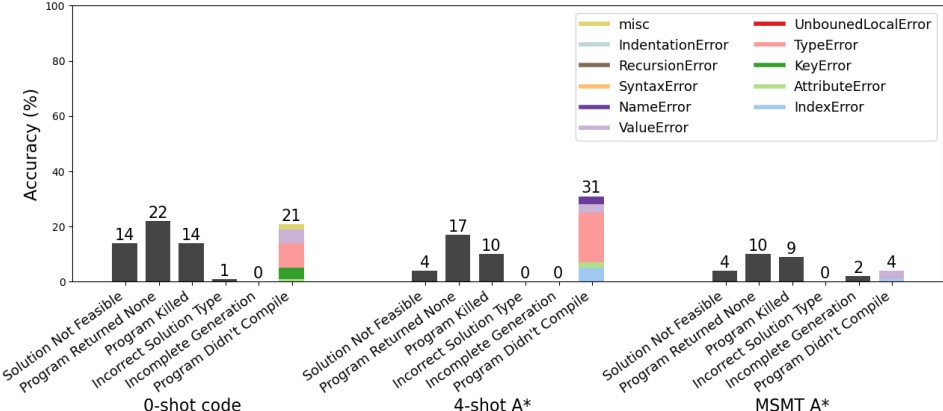

Figure 5: Rate of errors returned by python programs generated by GPT-4, categorized into 6 error types, calculated across all SearchBench problems with an infeasible solution.

improvements, with up to 15 additional instances solved. This indicates that SearchBench problems within the same category still differ significantly in rules, constraints, and target A* algorithm implementations.

The most significant improvement was observed for the Consecutive Grid problems from the under-determined systems category which involve searching over states that satisfy constraints on the order of integers in a table. This category differs more significantly from other combinatorial problems in terms of reasoning patterns, showing that including more similar problems in the prompt leads to greater improvement for novel task.

**What types of coding errors occur, and how often, when running GPT-4's code generations?** We analyzed the errors returned by GPT-4's generated codes that resulted in infeasible solutions. The results are shown in Fig. 5. We categorized errors into six types: (i) 'Solution Not Feasible' - code executed but returned an infeasible solution; (ii) 'Program Returned None' - code failed to find a solution; (iii) 'Program Killed' - code did not finish within the allotted time; (iv) 'Incorrect Solution Type' - returned solution had the wrong data type; (v) 'Incomplete Generation' - model ran out of tokens; and (vi) 'Program Didn't Compile'.

As shown in Fig. 5, prompting the model with the A* method results in more non-compiling code instances compared to 0-shot code prompting. This is expected as A* is more complex and requires generating longer code than the simpler algorithms typically used by the model using 0-shot code generation, such as the greedy algorithm, BFS, or DFS. However, the number of infeasible solutions significantly decreases with A* prompting, indicating that the model can better reason about the problem when prompted with the A* examples. When comparing A* prompting to the MSMT A* method, we notice that all of the errors that fail at least one unit test, including 'Program Returned None', 'Program Killed', 'Incorrect Solution Type', 'Incomplete Generation', and 'Program Didn't Compile', significantly decrease.

**What are the most common reasoning errors made in GPT-4's A* implementations?** We manually analyzed 50 A* codes generated by GPT-4 that returned non-optimal solutions across five problems: three pathfinding problems and two puzzle problems. These problems were chosen because GPT-4 showed the least and greatest performance improvement, respectively, using A* prompting compared to 0-shot code (see Appendix Sec. 3). We identified seven distinct failure modes in the GPT-4-generated A* implementations. Each failure mode corresponds to a critical subtask within the overall search strategy. Failing any one of these subtasks results in a suboptimal solution. The results are summarized in Tab. 2 where the percent of 'correct reasoning' (disregarding coding errors) is reported for each subtask. As shown, in pathfinding problems, the most common reasoning mistake was failing to record the list of coordinates visited (13% success rate). Specifically, the model often did not record the start coordinate in the list of visited states. This in turn led to feasible yet incorrect solutions. For the puzzle problems, the most frequent reasoning mistake was in encoding the goal state. This is likely because our puzzle problems featured unique expected goal states, e.g., different from the conventional 8-puzzle problem.

Table 2: The average accuracy of GPT-4 on the identified A* (failure modes) subtasks. This analysis was based on 50 codes implemented for pathfinding and puzzle problems, using A* prompting.

|                                     | **Pathfinding Problems** | **Puzzle Problems** |
|-------------------------------------|--------------------------|---------------------|
| **Encoding Initial State**          | 47%                      | 100%                |
| **Encoding Goal State**             | 74%                      | 20%                 |
| **Recording the Path/Actions**      | 13%                      | 70%                 |
| **Exit Condition**                  | 70%                      | 100%                |
| **Iterating Through Successor States** | 57%                   | 100%                |
| **Generate New State**              | 87%                      | 100%                |
| **Admissible and Consistent Heuristc** | 93%                   | 60%                 |

# 7 CONCLUSIONS, LIMITATIONS, AND BROADER IMPACT

In this work, we introduced SearchBench, a pioneering benchmark designed to assess the reasoning capabilities of large language models (LLMs) in solving challenging and ubiquitous search problems using various text-based and code-based prompting methods. We demonstrated that advanced LLMs can, to some extent, successfully solve search problems by implementing structured algorithms, especially when the models' implementations are verified against unit tests. This suggests a potential future path for automating the addition of new problem types to SearchBench. Specifically, we could leverage LLMs to generate evaluation and instance generation pipelines for new problem types based on a natural language description of the problem. Our MSMT approach could then verify the accuracy of these pipelines using comprehensive unit tests, significantly streamlining the process of scaling the dataset to include new problem types. Limitations and broader impact are discussed in the Appendix Sec. D.

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

# Appendix

## Table of Contents

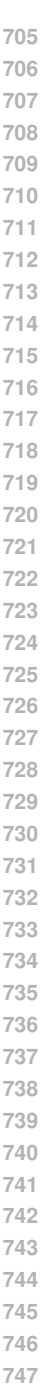

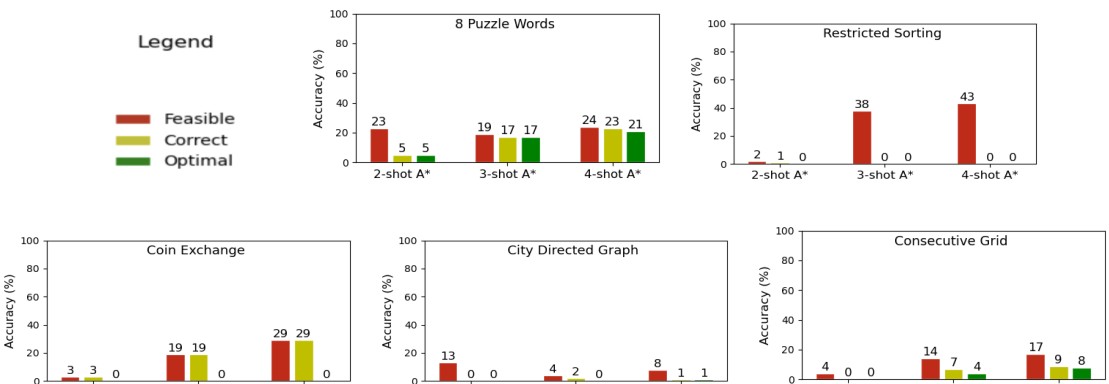

Figure 6: Comparing 2-shot, 3-shot and 4-shot performance of GPT4 between A*-prompting approaches.

## A   N-SHOT ABLATION EXPERIMENTS

To examine the effect of different numbers of demonstrations on GPT-4's performance using A* and MSMT A* prompting methods, we performed ablation experiments with 2-shot and 3-shot A* prompts. 4-shot is upper limit on the number of in-context examples due to the context length constraints of the models, including GPT-4. In all few-shot experiments, the examples used in the prompts were not from the evaluated problem category. The results, summarized in Fig. 6, show a consistent trend of performance improvement with the addition of more examples, as expected.

## B   DETAILED ANALYSIS OF GPT-4'S PERFORMANCE ON SEARCHBENCH

Tab. 3 details GPT-4 code-based method performance for each of SearchBench's 11 problems. Consistently 4-shot A* prompting outperforms 0-shot code for most problems. Interestingly for problems in the pathfinding category, prompting GPT4 with 0-shot code outperforms A* prompting.

Examining closer, GPT-4 mainly uses DFS for pathfinding in 0-shot code. While simpler than A*, DFS doesn't guarantee optimal solutions, as reflected in GPT-4's high feasible and correct rates but lower optimal rates. Implementing A* with an admissible and consistent heuristic requires the model to implement a more complex strategy in the code involving additional constraints and more sophisticated data structures. This increases the likelihood of reasoning or coding errors, which could explain the dip in GPT-4's performance using A* prompting compared to 0-shot code when solving these problems.

Figure 7 further analyzes the relationship between problem difficulty (quantified by state space size of the problem) and the performance of GPT-4. As observed, the model's performance is generally higher on easier problems, particularly in terms of the rate of correct solutions. This is expected, as easier problems have a smaller state space to explore. However, the performance of the model does not change drastically across different difficulty levels. This indicates that the combinatorial problems in SearchBench are intrinsically hard for LLMs to solve in text due to the requirement for backtracking. Moreover, the difference in implementing an A* search algorithm for a difficult or easy instance of SearchBench is limited to encoding the initial and goal states. The rest of the algorithm implementation task remains the same. This is the reason why the model's performance is comparable across different difficulty levels, both using text-based and code-based methods.

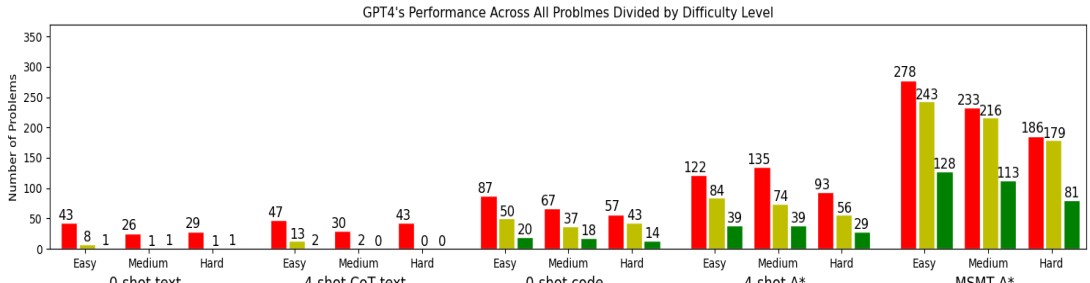

Figure 7: Count of feasible, correct, and optimal solutions generated by GPT4 via code-based methods for 3 levels of problem difficulty.

| Problem | 0-shot code | | | 4-shot A* | | | MSMT A* | | |
|---|---|---|---|---|---|---|---|---|---|
| 8 Puzzle | F: 3 | C: 0 | O: 0 | F: 63 | C: 60 | O: 60 | F: 76 | C: 68 | O: 68 |
| 8 Puzzle Words | F: 5 | C: 5 | O: 5 | F: 24 | C: 23 | O: 21 | F: 66 | C: 65 | O: 65 |
| Color Sorting | F: 17 | C: 1 | O: 1 | F: 41 | C: 35 | O: 6 | F: 91 | C: 91 | O: 0 |
| Restricted Sorting | F: 32 | C: 0 | O: 0 | F: 43 | C: 0 | O: 0 | F: 66 | C: 0 | O: 0 |
| Water Jug | F: 7 | C: 7 | O: 6 | F: 8 | C: 8 | O: 0 | F: 95 | C: 95 | O: 0 |
| Coin Exchange | F: 2 | C: 1 | O: 0 | F: 31 | C: 31 | O: 0 | F: 95 | C: 95 | O: 0 |
| Traffic | F: 65 | C: 50 | O: 13 | F: 24 | C: 5 | O: 5 | F: 65 | C: 60 | O: 60 |
| Trampoline Matrix | F: 27 | C: 27 | O: 22 | F: 51 | C: 4 | O: 4 | F: 57 | C: 53 | O: 46 |
| City Directed Graph | F: 29 | C: 28 | O: 1 | F: 7 | C: 0 | O: 0 | F: 55 | C: 51 | O: 45 |
| Magic Square | F: 3 | C: 1 | O: 0 | F: 8 | C: 5 | O: 0 | F: 14 | C: 14 | O: 0 |
| Consecutive Grid | F: 15 | C: 2 | O: 0 | F: 17 | C: 9 | O: 8 | F: 27 | C: 27 | O: 27 |

Table 3: GPT-4's performance when prompted with our code-based approaches, on each problem type. The values are percentages of the feasible (F), correct (C), and optimal (O) solutions.

## C  COMPUTE TIME OF LLM-GENERATED CODES

In this section, we analyze the computation time of programs generated by LLMs that produce correct solutions. We compare this time to the duration required to calculate the optimal solution for the problem instance using our fast A* implementation. This comparison provides insights into the efficiency of the algorithms generated by the LLMs. The average compute time of LLM-generated codes, normalized against the compute time of our A* implementation for the given instance, is reported in Fig. 8.

Our findings indicate that LLM-generated implementations are significantly slower than our A* implementation. Specifically, GPT-4's A* implementations were 213 times slower than the optimal A* solution, suggesting that GPT-4's heuristics are still less efficient. Additionally, on average, GPT-4's 0-shot code generations that return a correct solution run 900 times slower than the optimal A* implementation. These results underscore the intrinsic difficulty of SearchBench problems, even when addressed through code generation.

## D  LIMITATIONS AND BROADER IMPACT

**Limitations:** The primary challenge in developing the SearchBench dataset was scaling the number of problem types. Designing unique search problems and creating pipelines to generate numerous instances

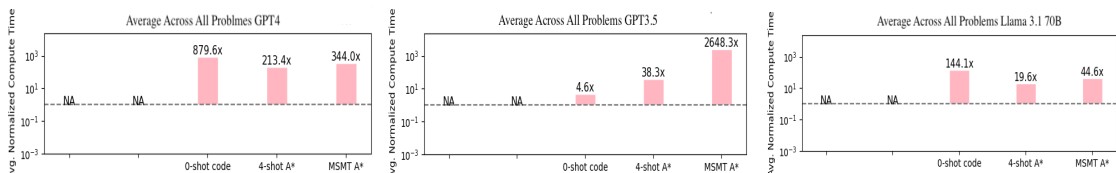

Figure 8: Average compute time of codes returning a correct solution normalized against the compute time of out A* implementation for all problems using GPT4, GPT3.5, Llama 3.1 70B.

with guaranteed solutions is both time-consuming and complex. Additionally, implementing a fast, instance-agnostic A* algorithm and developing evaluation pipelines to assess LLM-proposed solutions on multiple criteria further adds to the complexity.

However, we have shown that advanced LLMs can implement structured algorithms with scaled reliability, especially when generations are verified against unit tests as in our MSMT approach. This opens up the possibility of automating the addition of new problem types a to SearchBench.

**Broader Impact:** Our research, which aims to assist the development of models capable of general reasoning and reliable problem-solving, has the potential to yield significant societal benefits. Combinatorial problems, like those in our dataset, are fundamental in fields such as robotics, logistics, network design, and industrial optimization. Developing models that can tackle unique versions of these problems by designing efficient algorithms or performing systematic searches end-to-end could greatly enhance AI's applicability across various domains. However, this improvement in the reasoning capabilities of language models could also lead to job displacement, as these models could increasingly automate complex tasks traditionally performed by humans.

# E  SEARCHBENCH VARIABLES

Table continued in the next page.

| **Variables** | |
|---|---|
| **diff_sorted_id** | A unique numeric identifier assigned to each problem instance within a specific problem type. These identifiers are ordered by difficulty level, that is the problem instance with diff_sorted_id of 1 is easier than the instance with diff_sorted_id of 50. |
| **problem_statement** | A natural language description that outlines the problem to be solved. The problem statement is the sole piece of information given to language models when they are instructed to solve SearchBench problems. |
| **problem_type** | Indicates the problem type, out of 11 problem types in Search-Bench, that this particular problem is an instance of. |
| **problem_category** | The specific category, out of the five predefined problem categories in SearchBench, to which this problem belongs. |
| **relative_diff_score** | A numeric score that indicates the difficulty of this problem instance relative to other instances within the same problem type. This value is not comparable across different problem types. |
| **opt_solution** | A list of actions that, starting from the given initial state, lead to the goal state with the minimum cost as defined by the problem's criteria. |
| **opt_solution_cost** | The cost of the optimal solution for this problem instance. |
| **opt_solution_compute_t** | The time, in seconds, that our instance-agnostic A* implementation for the problem type took to solve this specific problem instance. |
| **solution_depth** | The number of actions required to reach the goal state from the given initial state with the minimum cost. This metric can be used to calculate an upper bound on the size of the search tree, represented as $b^d$, for this instance, where, b is an upper bound on the branching factor of the tree, which indicates the maximum number of actions leading to successor states from any given state, and d is the solution depth, representing the number of actions in the optimal solution. |
| **max_successor_states** | The maximum number of successor states that can be reached from any given state in this problem. This value is an upper bound on the branching factor of the state search tree for this problem. |

Table 4: This table provides a description of each column in SearchBench. Each row in SeacrhBench is an specific problem instance, and columns are fields of each instance.

| Variables | |
|---|---|
| **num_vars_per_state** | An upper bound on the number of variables in each state of the problem. Given that the number of states grows exponentially for SearchBench problems, this value provides an estimate of the memory required to traverse the search tree of the problem. |
| **is_feasible_args** | A list of variables of the problem instance that must be passed to the 'is_feasible' function of the evaluation pipeline to determine whether a suggested solution adheres to the rules and constraints of the problem. |
| **is_correct_args** | A list of variables in the problem statement of this instance that must be passed as arguments to the 'is_correct' function in the evaluation pipeline, in order to evaluate the correctness of a suggested solution. |
| **A\*_args** | Variables of this problem instance that must be passed to our A\* implementation for the problem type to obtain the optimal solution for the instance. |

## F  SEARCH TREE SIZE ANALYSIS

Table 5: Statistics of metrics pertaining to the search-tree-size of a specific instance, compared across all instances within SearchBench.

| Statistics | | | | | | | |
|---|---|---|---|---|---|---|---|
| **name** | **type** | **min** | **median** | **max** | **mean** | **standard deviation** | **missing** |
| opt_solution_compute_t | float (seconds) | 0.018 | 0.068 | 599.044 | 17.363 | 67.513 | 0% |
| solution_depth | int | 4 | 14 | 46 | 15.516 | 7.89 | 0% |
| max_successor_states | int | 4 | 12 | 132 | 24.633 | 24.622 | 0% |
| num_vars_per_state | int | 2 | 13 | 60 | 14.785 | 12.05 | 0% |

Figure 9 presents the relationship between the size of the state search tree and the difficulty levels of instances in SearchBench. It displays the average solution-depth and max_successor_state (normalized against the maximum and minimum solution_depth and max_successor_state across all instances in SearchBench) for one problem type from each of the five categories in SearchBench. Additionally, it shows the time our A\* algorithm took to navigate the search tree for instances of variable difficulty (compute time is averaged across instances with the same difficulty). We used a machine with 96 64-bit Intel Xeon Gold 5220R CPUs with a maximum speed of 4GHz, and 71.5 MiB Level 3 cache to run the A\* implementations.

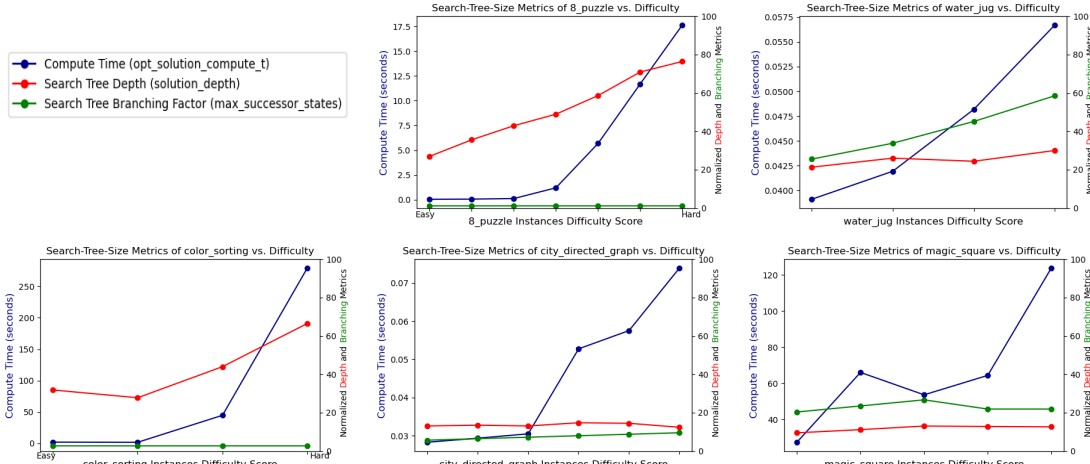

Figure 9: The plots depict the correlation between the increasing difficulty level and the corresponding increase in three metrics: the average depth of the solutions, the branching factor of the state search tree, and the exponential growth of the time required by our A* algorithm to solve the instances, demonstrated across five problem types in SearchBench.

The figure shows that the solution depth increases linearly with the difficulty scores of problem instances. However, for the city graph, it remains relatively constant, suggesting that the optimal number of hops to reach a destination node from a start node is consistent for our chosen range of directed graph connectivity and sizes (10 to 15 nodes).The max_successor_states, which represents the upper bound on the number of actions leading to successor states from each state, either remains constant or grows linearly with increasing difficulty level. This metric indicates the branching factor of the search tree size.

However, the compute time required to navigate this search tree grows much faster, exponentially, for most problems, as expected, given the search tree size is $b^d$, where b is the branching factor, and d is the solution depth. It's worth noting that we used a fast heuristic A* algorithm, which doesn't navigate the full search tree. An exhaustive algorithm like BFS, which explores every node, would result in a much faster exponential growth of compute times. In our experiments, a BFS implementation didn't finish executing even for some of the easiest instances within a 12-hour window.

## G GPT4'S MSMT A* IMPLEMENTATIONS FOR TWO INSTANCES OF EACH PROBLEM TYPE

In this section, we present the A* algorithm generated by GPT-4 using the MSMT A* prompting method, which successfully passed the unit tests. Additionally, we include GPT-4's implementation of the 'initialize' function for a more challenging instance, generated in the second stage of the MSMT.

To facilitate the model's reasoning process when mapping the problem to a graph, designing steps of the A* algorithm, and reasoning about the admissibility or consistency of the heuristic, we employ a Chain of Thought (CoT) reasoning in text as comments in our in-context examples. We provide additional information before each code segment in our prompts that walk through the reasoning steps required to arrive at the strategy implemented in the code. Below, you can observe GPT-4's comments that outline the intermediate reasoning steps the model generated to implement various code sections.

The implementations begin on the next page.

# 8 PUZZLE

> **8_puzzle instance (diff_sorted_id: 1)**: In the 8-puzzle game, you are given a grid with numbered square tiles arranged randomly and one tile missing. The goal is to arrange the tiles in descending order by sliding them into the empty space. The tiles can move in 4 directions: left, right, up, and down. Given the initial state of the puzzle below, where the empty spot is represented as "_", provide the shortest list of tiles that need to be swapped with the empty spot to achieve the goal state. The goal state is when all tiles are in descending order, with the largest number in the top left corner, and the empty spot is in the bottom right corner. The solution should be a list of numbers in Python format, where each number represents the number on the tile that the empty spot is swapped with at each turn. Initial state of the puzzle: [[12, 55, '_'], [35, 97, 43], [17, 25, 9]]

A* algorithm generated by GPT4 for the problem above using A* MSMT approach that passed the unit tests:

```python
import heapq

def initialize():
    # Define the initial state of the puzzle as a 2d tuple
    initial_state = ((12, 55, '_'), (35, 97, 43), (17, 25, 9))
    num_rows = 3
    num_cols = 3
    goal_state = ((97, 55, 43), (35, 25, 17), (12, 9, '_'))
    visited_costs = {}
    visited_costs[initial_state] = 0
    queue = [(0, 0, [], initial_state)]
    return initial_state, num_rows, num_cols, goal_state, visited_costs, queue

def a_star():
    initial_state, num_rows, num_cols, goal_state, visited_costs, queue = initialize()
    while queue:
        _, g, actions, state = heapq.heappop(queue)
        # If the current state is the goal state, return the actions taken
        if state == goal_state:
            return actions
        # Generate all possible actions from the current state, which includes swapping the
    empty spot with any of the 4 neighboring tiles
        empty_spot_coord = [(i, j) for i, row in enumerate(state) for j, element in enumerate(
    row) if element == '_'][0]
        for d_row, d_col in [(0, -1), (0, 1), (1, 0), (-1, 0)]:
            new_row, new_col = empty_spot_coord[0] + d_row, empty_spot_coord[1] + d_col
            # Check if the new coordinate is valid, ie if the new tile is within the bounds of
    the puzzle
            if 0 <= new_row < num_rows and 0 <= new_col < num_cols:
                # Generate the new state
                new_state = [list(row[:]) for row in state]
                # Swap the empty spot with the new tile
                new_state[empty_spot_coord[0]][empty_spot_coord[1]], new_state[new_row][new_col
    ] = new_state[new_row][new_col], new_state[empty_spot_coord[0]][empty_spot_coord[1]]
                new_state = tuple(tuple(row) for row in new_state)
                # The cost so far is the number of swaps made, as the task is to minimize the
    number of swaps required
                new_cost = g + 1
                if new_state not in visited_costs or new_cost < visited_costs[new_state]:
                    visited_costs[new_state] = new_cost
                    # The action taken is the number on the tile that was swapped with the
    empty spot
                    heapq.heappush(queue, (g + heuristic(new_state, goal_state), new_cost,
    actions + [int(new_state[empty_spot_coord[0]][empty_spot_coord[1]])], new_state))
    return None
```

Continuation of GPT-4's implementation:

```python
def heuristic(state, goal):
    # An admissible and consistent heuristic for this problem is the sum of the Manhattan
      distances from each tile in its current position to its position in the goal state
    # This heuristic relaxes the constraint that only the empty spot can be moved, and assumes
      that each tile can be moved to its goal position independently of the others
    # Thus the heuristic reports a lower estimate on the cost of reaching the goal state and is
      admissible
    # The heuristic is consistent because the estimated cost from the current state to the goal
      can never be greater than the sum of cost from the current node to a successor node plus
      the estimated cost from the successor node to the goal because the cost of moving the
      empty spot to an adjacent tile is 1, which is equal to the decrease in the Manhattan
      distance of the tile that was swapped with the empty spot
    # Also the cost of the goal state is 0, as all tiles are in their goal positions
    h = 0
    for i in range(len(state)):
        for j in range(len(state[0])):
            if state[i][j] != '_':
                goal_coord = [(x, y) for x, row in enumerate(goal) for y, element in enumerate(
    row) if element == state[i][j]][0]
                h += abs(i - goal_coord[0]) + abs(j - goal_coord[1])
    return h

print(a_star())
```

Table 6: The A* algorithm was generated by GPT4 using the MSMT A* approach. For 8_puzzle problem type, GPT4's A* implementation for the first instance passed the unit tests.

**8_puzzle instance (diff_sorted_id: 94):** In the 8-puzzle game, you are given a grid with numbered square tiles arranged randomly and one tile missing. The goal is to arrange the tiles in descending order by sliding them into the empty space. The tiles can move in 4 directions: left, right, up, and down. Given the initial state of the puzzle below, where the empty spot is represented as "_", provide the shortest list of tiles that need to be swapped with the empty spot to achieve the goal state. The goal state is when all tiles are in descending order, with the largest number in the top left corner, and the empty spot is in the bottom right corner. The solution should be a list of numbers in Python format, where each number represents the number on the tile that the empty spot is swapped with at each turn. Initial state of the puzzle: [[98, 30, 85, 44, 59, 40], [56, 68, 50, 61, 47, 15], [37, 48, 24, 81, 22, '_']]

'initialize' function generated by GPT4 for the problem above in the second stage of the MSMT approach

```python
def initialize():
    # Define the initial state of the puzzle as a 2d tuple
    initial_state = (('98', '30', '85', '44', '59', '40'),
                     ('56', '68', '50', '61', '47', '15'),
                     ('37', '48', '24', '81', '22', '_'))
    num_rows = 3
    num_cols = 6
    # Define the goal state of the puzzle as a 2d tuple
    goal_state = (('98', '85', '68', '61', '59', '56'),
                  ('50', '48', '47', '44', '40', '37'),
                  ('30', '24', '22', '15', '81', '_'))

    visited_costs = {}
    visited_costs[initial_state] = 0
    queue = [(0, 0, [], initial_state)]
    return initial_state, num_rows, num_cols, goal_state, visited_costs, queue
```

Table 7: GPT4 was only successful in generating a feasible solution for this instance using the MSMT A* approach. The other four code and text-based prompting methods used in our experiments resulted in infeasible solutions.

# 8 PUZZLE WORDS

**8_puzzle_words instance (diff_sorted_id: 1)**: In the game 'Sort the Chars', we are given a table of n by m dimensions. This table contains n words, each with m characters, except for the first word which has m - 1 characters. Each character is written on a separate tile. The objective of the game is to rearrange the characters such that row i spells the i-th word in the list, with the blank tile ('_') placed in the top left corner of the board in the end. We can rearrange the tiles by swapping the blank space with any of its 4 diagonal neighboring tiles. Given the list of words and initial state of the board below, where the black space is represented as '_', what is the shortest list of swap actions (reported in python syntax) that can sort the board into the given list of target words? The list must only include the 4 diagonal swap directions: up-right, down-right, up-left, or down-left, representing the direction in which the blank space was swpped in. Target words: cam, hill, pray, doer The initial board: [['i', 'c', 'a', 'm'], ['h', 'p', 'l', 'o'], ['_', 'r', 'a', 'y'], ['d', 'l', 'e', 'r']]

A* algorithm generated by GPT4 for the problem above using A* MSMT approach that passed the unit tests:

```python
import heapq

def initialize():
    # Define the initial state of the board as a 2d tuple
    initial_state = (('i', 'c', 'a', 'm'), ('h', 'p', 'l', 'o'), ('_', 'r', 'a', 'y'), ('d', 'l', '
      e', 'r'))
    num_rows = 4
    num_cols = 4
    # Define the goal state of the board as a 2d tuple
    goal_state = (('_', 'c', 'a', 'm'), ('h', 'i', 'l', 'l'), ('p', 'r', 'a', 'y'), ('d', 'o', 'e', '
      r'))
    visited_costs = {}
    visited_costs[initial_state] = 0
    queue = [(0, 0, [], initial_state)]
    return initial_state, num_rows, num_cols, goal_state, visited_costs, queue

def a_star():
    initial_state, num_rows, num_cols, goal_state, visited_costs, queue = initialize()
    while queue:
        _, g, actions, state = heapq.heappop(queue)
        # If the current state is the goal state, return the actions taken
        if state == goal_state:
            return actions
        # Generate all possible actions from the current state, which includes swapping the blank
      space with any of its 4 diagonal neighbors
        for d_row, d_col in [(-1, 1), (1, 1), (-1, -1), (1, -1)]:
            # Get the coordinate of the blank space
            blank_coord = [(i, j) for i, row in enumerate(state) for j, element in enumerate(row)
      if element == '_'][0]
            new_row, new_col = blank_coord[0] + d_row, blank_coord[1] + d_col
            # Check if the new coordinate is valid, ie if the new position is within the bounds of
      the board
            if 0 <= new_row < num_rows and 0 <= new_col < num_cols:
                # Generate the new state
                new_state = [list(row[:]) for row in state]
                new_state[blank_coord[0]][blank_coord[1]], new_state[new_row][new_col] = new_state[
      new_row][new_col], new_state[blank_coord[0]][blank_coord[1]]
                new_state = tuple(tuple(row) for row in new_state)
                # The cost so far is the number of actions taken, as the task is to minimize the
      number of actions required to sort the board
                new_cost = g + 1
                if new_state not in visited_costs or new_cost < visited_costs[new_state]:
                    visited_costs[new_state] = new_cost
                    # The direction of the swap action is the direction in which the blank space
      was moved
                    if d_row == -1 and d_col == 1:
                        action = 'up-right'
                    elif d_row == 1 and d_col == 1:
                        action = 'down-right'
                    elif d_row == -1 and d_col == -1:
                        action = 'up-left'
                    elif d_row == 1 and d_col == -1:
                        action = 'down-left'
                    heapq.heappush(queue, (g + heuristic(new_state, goal_state), new_cost, actions
      + [action], new_state))
    return None
```

25

The GPT4's implementation of the A* algorithm continues on the next page.

Continuation of GPT-4's implementation:

```python
def heuristic(state, goal):
    # An admissible and consistent heuristic for this problem is the count of tiles that are
      not in their goal position
    # This heuristic relaxes the constraint that only the blank space can be moved, and only to
      a diagonal position
    # It is admissible because it never overestimates the cost to reach the goal, as each
      misplaced tile must be moved at least once
    # It's consistent because moving a tile reduces the heuristic cost of the successor node by
      a max of 1 (if the moved tile is placed in its goal position), which is equal to the cost
      of reaching the successor node
    # Thus h(n) is always less than or equal to c(n, n')(equal to 1) + h(n')
    # And the cost of the goal state is 0, as all tiles will be in their goal position
    h = 0
    for i in range(len(state)):
        for j in range(len(state[0])):
            if state[i][j] != goal[i][j]:
                h += 1
    return h

print(a_star())
```

Table 8: The A* algorithm was generated by GPT4 using the MSMT A* approach. For 8_puzzle_words problem type, GPT4's A* implementation for the first instance passed the unit tests.

**8_puzzle_words instance (diff_sorted_id: 93)**: In the game 'Sort the Chars', we are given a table of n by m dimensions. This table contains n words, each with m characters, except for the first word which has m - 1 characters. Each character is written on a separate tile. The objective of the game is to rearrange the characters such that row i spells the i-th word in the list, with the blank tile ('_') placed in the top left corner of the board in the end. We can rearrange the tiles by swapping the blank space with any of its 4 diagonal neighboring tiles. Given the list of words and initial state of the board below, where the black space is represented as '_', what is the shortest list of swap actions (reported in python syntax) that can sort the board into the given list of target words? The list must only include the 4 diagonal swap directions: up-right, down-right, up-left, or down-left, representing the direction in ehich the blank space was swpped in. Target words: helen, adelea, bumble, iodism, trullo The initial board: [['d', 'h', '_', 'l', 'e', 'n'], ['a', 'a', 'e', 'b', 'e', 'u'], ['m', 'u', 'i', 'b', 'e', 'e'], ['i', 'o', 'd', 'm', 's', 'l'], ['t', 'r', 'l', 'l', 'l', 'o']]

'initialize' function generated by GPT4 for the problem above in the second stage of the MSMT approach

```python
def initialize():
    # Define the initial state of the board as a 2d tuple
    initial_state = (('d', 'h', '_', 'l', 'e', 'n'),
                     ('a', 'a', 'e', 'b', 'e', 'u'),
                     ('m', 'u', 'i', 'b', 'e', 'e'),
                     ('i', 'o', 'd', 'm', 's', 'l'),
                     ('t', 'r', 'l', 'l', 'l', 'o'))
    num_rows = 5
    num_cols = 6
    # Define the goal state of the board as a 2d tuple
    goal_state = (('_', 'h', 'e', 'l', 'e', 'n'),
                  ('a', 'd', 'e', 'l', 'e', 'a'),
                  ('b', 'u', 'm', 'b', 'l', 'e'),
                  ('i', 'o', 'd', 'i', 's', 'm'),
                  ('t', 'r', 'u', 'l', 'l', 'o'))
    visited_costs = {}
    visited_costs[initial_state] = 0
    queue = [(0, 0, [], initial_state)]
    return initial_state, num_rows, num_cols, goal_state, visited_costs, queue
```

Table 9: GPT4 was only successful in generating an optimal solution for this instance using the MSMT A* approach. The other four code and text-based prompting methods used in our experiments resulted in infeasible solutions.

COIN EXCHANGE

> **coin_exchange instance (diff_sorted_id: 1)**: In the 'taxed coin exchange' problem, you are required to choose a subset of coins from this list [14, 2, 5, 8, 65, 5, 16, 2, 5, 2, 12, 17, 16, 15, 20, 4, 19, 9, 10, 64, 2, 10, 14, 2, 17], such that the sum of the chosen coins adds up to 208. Each coin in the list is unique and can only be used once. Also coins carry a tax value. The tax values for each coin is 20: 13, 4: 2, 17: 16, 19: 8, 64: 4, 5: 3, 14: 10, 10: 9, 65: 11, 16: 2, 12: 7, 8: 5, 2: 2, 15: 14, 9: 8, where the tax for coins of the same value is the same. Also, if the coin chosen is smaller than the previous one, it must have an even value, otherwise, if the coin is larger than or equal to the previous coin chosen, it must have an odd value. The objective is to determine which subset of coins should be selected to minimize the total tax paid. The solution should be presented as a list of numbers, representing the value of the coins chosen in order, with the first coins chosen being in index 0, formatted in Python syntax.

A* algorithm generated by GPT4 for the problem above using A* MSMT approach that passed the unit tests:

```python
import heapq

def initialize():
    # Define the initial state of the problem, which includes the coins available, the sum of the
      coins chosen so far, the tax paid so far, and the value of the last coin chosen
    initial_state = ((14, 2, 5, 8, 65, 5, 16, 2, 5, 2, 12, 17, 16, 15, 20, 4, 19, 9, 10, 64, 2, 10,
      14, 2, 17), 0, 0, 0)
    goal_sum = 208
    tax_values = {20: 13, 4: 2, 17: 16, 19: 8, 64: 4, 5: 3, 14: 10, 10: 9, 65: 11, 16: 2, 12: 7, 8:
      5, 2: 2, 15: 14, 9: 8}
    visited_costs = {}
    visited_costs[initial_state] = 0
    queue = [(0, 0, [], initial_state)]
    return initial_state, goal_sum, tax_values, visited_costs, queue

def a_star():
    initial_state, goal_sum, tax_values, visited_costs, queue = initialize()
    while queue:
        _, g, actions, state = heapq.heappop(queue)
        # Unpack the information encoded in each state
        coins, sum_coins, tax_paid, last_coin = state
        # If the sum of the coins chosen so far equals the goal sum, return the coins chosen
        if sum_coins == goal_sum:
            return actions
        # If the sum of the coins chosen so far is less than the goal sum, generate all possible
      actions from the current state, which includes choosing any of the remaining coins
        elif sum_coins < goal_sum:
            for coin in coins:
                # Check if the new state, containing the new coin, would be valid; ie the coin must
      be even if it is smaller than the last coin chosen, and odd if it is larger than or equal to
      the last coin chosen
                if (coin < last_coin and coin % 2 == 0) or (coin >= last_coin and coin % 2 == 1):
                    # Generate the new state
                    new_state = (tuple(c for c in coins if c != coin), sum_coins + coin, tax_paid +
      tax_values[coin], coin)
                    # The additional cost of this state is the tax paid for the new coin as we are
      trying to minimize the total tax paid
                    new_cost = g + tax_values[coin]
                    if new_state not in visited_costs or new_cost < visited_costs[new_state]:
                        visited_costs[new_state] = new_cost
                        # The new coin must be added to the actions
                        heapq.heappush(queue, (g + heuristic(new_state, goal_sum), new_cost,
      actions + [coin], new_state))
    return None
```

The GPT4's implementation of the A* algorithm continues on the next page.

Continuation of GPT-4's implementation:

```python
def heuristic(state, goal_sum):
    # An admissible and consistent heuristic for this problem is the difference between the
      goal sum and the sum of the coins chosen so far
    # The heuristic assumes we can reach the goal sum without paying any additional tax, thus
      reporting a lower cost estimate to reach the goal state, making it admissible
    # The heuristic is consistent, ie non-decreasing along the path to the goal state, because
      the cost of choosing a coin is the tax paid for that coin, which is always greater than or
       equal to 0, the decrease in the difference between the goal sum and the sum of the coins
      chosen so far
    # Also the cost of the goal state is 0, as the sum of the coins chosen equals the goal sum
      and there are no more coins to choose

    return goal_sum - state[1]

print(a_star())
```

Table 10: The A* algorithm was generated by GPT4 using the MSMT A* approach. For coin_exchange problem type, GPT4's A* implementation for the first instance passed the unit tests.

**coin_exchange instance (diff_sorted_id: 104)**: In the 'taxed coin exchange' problem, you are required to choose a subset of coins from this list [18, 23, 23, 36, 27, 10, 25, 14, 6, 43, 6, 31, 5, 43, 3, 13, 16, 8, 11, 6, 39, 44, 41, 28, 12, 8, 1, 14, 22, 23, 11, 24, 37, 9, 2, 31, 27, 10, 9, 38, 37, 4, 4, 17, 18, 36, 18, 32, 3, 12, 4, 42, 8, 4, 4, 16, 30, 40, 13, 41], such that the sum of the chosen coins adds up to 441. Each coin in the list is unique and can only be used once. Also coins carry a tax value. The tax values for each coin is 6: 4, 18: 3, 38: 1, 23: 2, 16: 8, 12: 2, 4: 2, 24: 13, 28: 17, 30: 17, 1: 1, 9: 9, 27: 8, 11: 4, 37: 16, 41: 19, 13: 3, 31: 8, 44: 8, 22: 7, 43: 13, 10: 3, 17: 8, 36: 7, 39: 3, 25: 17, 40: 10, 8: 6, 32: 16, 14: 4, 2: 1, 42: 3, 3: 2, 5: 2, where the tax for coins of the same value is the same. Also, if the coin chosen is smaller than the previous one, it must have an even value, otherwise, if the coin is larger than or equal to the previous coin chosen, it must have an odd value. The objective is to determine which subset of coins should be selected to minimize the total tax paid. The solution should be presented as a list of numbers, representing the value of the coins chosen in order, with the first coins chosen being in index 0, formatted in Python syntax.

'initialize' function generated by GPT4 for the problem above in the second stage of the MSMT approach

```python
def initialize():
    def initialize():
    # Define the initial state of the problem, which includes the coins available, the sum of
      the coins chosen so far, the tax paid so far, and the value of the last coin chosen
    initial_state = ((18, 23, 23, 36, 27, 10, 25, 14, 6, 43, 6, 31, 5, 43, 3, 13, 16, 8, 11, 6,
        39, 44, 41, 28, 12, 8, 1, 14, 22, 23, 11, 24, 37, 9, 2, 31, 27, 10, 9, 38, 37, 4, 4, 17,
        18, 36, 18, 32, 3, 12, 4, 42, 8, 4, 4, 16, 30, 40, 13, 41), 0, 0, 0)
    goal_sum = 441
    tax_values = {6: 4, 18: 3, 38: 1, 23: 2, 16: 8, 12: 2, 4: 2, 24: 13, 28: 17, 30: 17, 1: 1,
        9: 9, 27: 8, 11: 4, 37: 16, 41: 19, 13: 3, 31: 8, 44: 8, 22: 7, 43: 13, 10: 3, 17: 8, 36:
        7, 39: 3, 25: 17, 40: 10, 8: 6, 32: 16, 14: 4, 2: 1, 42: 3, 3: 2, 5: 2}
    visited_costs = {}
    visited_costs[initial_state] = 0
    queue = [(0, 0, [], initial_state)]
    return initial_state, goal_sum, tax_values, visited_costs, queue
```

Table 11: GPT4 was only successful in generating a correct (but non-optimal) solution for this instance of the coin_exchange problem type using the MSMT A* approach. The other four code and text-based prompting methods used in our experiments resulted in infeasible solutions.

## WATER JUG

> **water_jug instance (diff_sorted_id: 1)**: In the 'taxed coin exchange' problem, you are required to choose a subset of coins from this list [14, 2, 5, 8, 65, 5, 16, 2, 5, 2, 12, 17, 16, 15, 20, 4, 19, 9, 10, 64, 2, 10, 14, 2, 17], such that the sum of the chosen coins adds up to 208. Each coin in the list is unique and can only be used once. Also coins carry a tax value. The tax values for each coin is 20: 13, 4: 2, 17: 16, 19: 8, 64: 4, 5: 3, 14: 10, 10: 9, 65: 11, 16: 2, 12: 7, 8: 5, 2: 2, 15: 14, 9: 8, where the tax for coins of the same value is the same. Also, if the coin chosen is smaller than the previous one, it must have an even value, otherwise, if the coin is larger than or equal to the previous coin chosen, it must have an odd value. The objective is to determine which subset of coins should be selected to minimize the total tax paid. The solution should be presented as a list of numbers, representing the value of the coins chosen in order, with the first coins chosen being in index 0, formatted in Python syntax.

A* algorithm generated by GPT4 for the problem above using A* MSMT approach that passed the unit tests:

```python
import heapq

def initialize():
    # Define the initial state of the buckets as a tuple
    initial_state = (0, 0, 0)
    # Define the goal state of the buckets as a tuple
    goal_state = (274, 297, 343)
    # Define the capacities of the jugs as a list
    jug_capacities = [41, 26, 44, 75, 40, 136]
    # Initialize a dictionary to store the cost of reaching each visited state
    visited_costs = {}
    visited_costs[initial_state] = 0
    # Initialize a priority queue of states not yet visited, with the initial state as the first
    #   element. The priority of each element is the cost to reach that state (g) + the estimate
    #   remaining cost (h) to reach the goal
    # Record the actions required to get to each state in a list; no actions performed to reach the
    #    initial state
    queue = [(0, 0, [], initial_state)]
    return initial_state, goal_state, jug_capacities, visited_costs, queue

def a_star():
    # The initialize function initializes and returns the visited_costs dictionary and the priority
    #    queue and encodes all of the variables given in the problem (ie the initial and goal state
    #    of the buckets and the capacities of the jugs)
    initial_state, goal_state, jug_capacities, visited_costs, queue = initialize()
    # While there are un-visited states
    while queue:
        # Pop the state with the lowest sum of the cost so far and estimated cost to the goal from
        #    the queue
        _, g, actions, state = heapq.heappop(queue)
        # We can check if the current state is the goal state with a simple equality check, as the
        #    goal state is predefined
        if state == goal_state:
            return actions
        # Generate all possible actions from the current state, which includes filling each of the
        #    3 buckets with each of the 6 jugs, and emptying each of the 3 buckets with each of the 6 jugs
        for i in range(len(jug_capacities)):
            for j in range(len(state)):
                # Check if the new state would be valid, ie if the bucket can be filled with the
                #    jug without overflowing and if the amount of water in each bucket does not exceed the amount
                #    of water in the bucket placed after it
                if state[j] + jug_capacities[i] <= goal_state[j] and (j == len(state) - 1 or state[
                    j] + jug_capacities[i] <= state[j + 1]):
                    # The action is valid, generate the new state
                    new_state = list(state)
                    new_state[j] += jug_capacities[i]
                    new_state = tuple(new_state)
                    # The cost so far is the number of actions performed, as our objective is to
                    #    minimize the number of actions required to reach the goal state
                    new_cost = g + 1
                    # If the new state is unvisited or we found a new path with a lower cost to
                    #    reach this state, add it to the queue of not-yet-visited states
                    if new_state not in visited_costs or new_cost < visited_costs[new_state]:
                        visited_costs[new_state] = new_cost
                        heapq.heappush(queue, (g + heuristic(new_state, goal_state), new_cost,
                            actions + [('+', jug_capacities[i], j + 1)], new_state))
```

30

The GPT4's implementation of the A* algorithm continues on the next page.

**Continuation of GPT-4's implementation:**

```python
                # Similarly, check if the new state would be valid, ie if the bucket can be
        emptied with the jug and if the amount of water in each bucket does not exceed the amount
        of water in the bucket placed after it
                if state[j] - jug_capacities[i] >= 0 and (j == 0 or state[j] - jug_capacities[i
        ] >= state[j - 1]):
                    # The action is valid, generate the new state
                    new_state = list(state)
                    new_state[j] -= jug_capacities[i]
                    new_state = tuple(new_state)
                    # The cost so far is the number of actions performed, as our objective is
        to minimize the number of actions required to reach the goal state
                    new_cost = g + 1
                    # If the new state is unvisited or we found a new path with a lower cost to
         reach this state, add it to the queue of not-yet-visited states
                    if new_state not in visited_costs or new_cost < visited_costs[new_state]:
                        visited_costs[new_state] = new_cost
                        heapq.heappush(queue, (g + heuristic(new_state, goal_state), new_cost,
        actions + [('-', jug_capacities[i], j + 1)], new_state))
    return None

def heuristic(state, goal):
    # An admissible and consistent heuristic is the sum of the differences between the current
      and goal amount of water in each bucket
    # The heuristic relaxes the constraint that the amount of water in each bucket can not at
      any point in time exceed the amount of water in the bucket placed after it and presumes we
       can fill or empty the buckets directly to their goal amount
    # Thus the heuristic reports a lower estimate on the cost to reach goal state and is
      admissible
    # The heuristic is consistent because the cost of filling or emptying a bucket is always 1,
       which is exactly the decrease or increase in the difference between the current and goal
      amount of water in the bucket, if the bucket is filled or emptied, respectively, otherwise
       the estimated cost of the successor node is the same or higher, and the heuristic
      estimate for the goal state is 0, as the difference between the current and goal amount of
       water in each bucket would be 0 in the goal state.
    return sum(abs(a - b) for a, b in zip(state, goal))

print(a_star())
```

Table 12: The A* algorithm was generated by GPT4 using the MSMT A* approach. GPT4's first A* implementation for the water_jug problem type passed the unit tests.

**water_jug instance (diff_sorted_id: 100)**: Given 9 labeled water jugs with capacities 126, 120, 127, 110, 54, 39, 38, 47, 139, 99 liters, we aim to fill 4 unlabeled buckets, numbered 1 to 4 and arranged in a line in ascending order, with 219, 240, 312, 411 liters of water respectively. The amount of water in each unlabeled bucket can not at any point in time exceed the amount of water in the bucket placed after it. Jugs can only be filled to the top and emptied completely, and the unlabeled buckets cannot be overfilled. An action, represented as a tuple ('+', X, Y) or ('-', X, Y), involves adding to or removing water from the unlabeled bucket numbered Y, using the jug with capacity X. Determine the shortest sequence of actions needed to fill the buckets as specified, and present the solution as a list of action tuples in Python syntax.

'initialize' function generated by GPT4 for the problem above in the second stage of the MSMT approach

```python
def initialize():
    # Define the initial state of the buckets as a tuple
    initial_state = (0, 0, 0, 0)
    # Define the goal state of the buckets as a tuple
    goal_state = (219, 240, 312, 411)
    # Define the capacities of the jugs as a list
    jug_capacities = [126, 120, 127, 110, 54, 39, 38, 47, 139, 99]
    # Initialize a dictionary to store the cost of reaching each visited state
    visited_costs = {}
    visited_costs[initial_state] = 0
    # Initialize a priority queue of states not yet visited, with the initial state as the
    #   first element. The priority of each element is the cost to reach that state (g) + the
    #   estimate remaining cost (h) to reach the goal
    # Record the actions required to get to each state in a list; no actions performed to reach
    #   the initial state
    queue = [(0, 0, [], initial_state)]
    return initial_state, goal_state, jug_capacities, visited_costs, queue
```

Table 13: GPT4 successfully generated a correct (but non-optimal) solution for this water_jug problem instance using the MSMT A* and A* prompting approach. The other three baseline code and text-based prompting methods resulted in infeasible solutions.

## RESTRICTED SORTING

**restricted_sorting instance (diff_sorted_id: 3)**: In 'Restricted Sorting', there are 6 stacks each with a capacity of 4 blocks, with 3 stacks filled with blocks of varying shades and the remaining are empty. The challenge is to sort the blocks by shade such that if a stack is not empty, it is stacked with 3 blocks of a single shade. The player can only transfer one block at a time from the top of a stack to an empty stack or to a stack that has only blocks of that shade, without exceeding the stacks' capacity. Transferring blocks to certain stacks is more expensive than others. The cost of moving one block to the top of each stack is: 0: 3, 1: 6, 2: 5, 3: 6, 4: 1, 5: 1, where the keys are the index of each stack, indexing from 0. The cost of moving a block is always at least 1. The solution should be a list of tuples, each containing, first, the index of the stack from which a block is picked up from and, second, the index of the stack to which it is transferred, indexing from 0. Given the initial state of the stacks, represented by the lists below (with the leftmost item being the shade of the topmost block in each stack)(and the first stack being the stack at index 0), what is the list of transfer pairs (reported in python syntax) with the least possible cost, that will result in all the blocks being correctly sorted? [[], ['Yellow', 'Green', 'Red', 'Red'], ['Blue', 'Green', 'Red', 'Green'], ['Yellow', 'Blue', 'Yellow', 'Blue'], [], []]

A* algorithm generated by GPT4 for the problem above using A* MSMT approach that passed the unit tests:

```python
import heapq

def initialize():
    # Define the initial state of the stacks as a tuple of tuples
    initial_state = ((), ('Yellow', 'Green', 'Red', 'Red'), ('Blue', 'Green', 'Red', 'Green'), ('
        Yellow', 'Blue', 'Yellow', 'Blue'), (), ())
    num_stacks = 6
    stack_capacity = 4
    # Define the cost of moving a block to each stack
    stack_costs = {0: 3, 1: 6, 2: 5, 3: 6, 4: 1, 5: 1}
    visited_costs = {}
    visited_costs[initial_state] = 0
    queue = [(0, 0, [], initial_state)]
    return initial_state, num_stacks, stack_capacity, stack_costs, visited_costs, queue

def a_star():
    initial_state, num_stacks, stack_capacity, stack_costs, visited_costs, queue = initialize()
    while queue:
        _, g, actions, state = heapq.heappop(queue)
        # If all the stacks are empty or contain blocks of a single shade, return the actions taken
        if all(len(stack) == 0 or len(set(stack)) == 1 for stack in state):
            return actions
        # Generate all possible actions from the current state, which includes moving the top block
        from any stack to any other stack
        for from_stack_ind in range(num_stacks):
            # Check if the stack has any blocks to move
            if state[from_stack_ind]:
                for to_stack_ind in range(num_stacks):
                    # Check if the stack can accept more blocks and if the block can be moved to
        the stack
                    if (len(state[to_stack_ind]) < stack_capacity and
                        (not state[to_stack_ind] or state[to_stack_ind][-1] == state[from_stack_ind
        ][-1])):
                        # Generate the new state
                        new_state = list(list(stack) for stack in state)
                        block = new_state[from_stack_ind].pop()
                        new_state[to_stack_ind].append(block)
                        new_state = tuple(tuple(stack) for stack in new_state)
                        # The cost of the new state is the cost of moving a block to the stack
                        new_cost = g + stack_costs[to_stack_ind]
                        if new_state not in visited_costs or new_cost < visited_costs[new_state]:
                            visited_costs[new_state] = new_cost
                            h = heuristic(new_state)
                            heapq.heappush(queue, (new_cost + h, new_cost, actions + [(
        from_stack_ind, to_stack_ind)], new_state))
    return None
```

The GPT4's implementation of the A* algorithm continues on the next page.

Continuation of GPT-4's implementation:

```python
def heuristic(state):
    # The heuristic function can be the sum of the number of blocks in each stack that are not
      of the same shade as the top block
    # This heuristic is admissible because it presumes we can move the blocks to any stack,
      even if the stack is full or contains blocks of a different shade, thus reporting a lower
      cost estimate to reach the goal state
    # The heuristic is consistent, ie non-decreasing along the path to the goal state, because
      the cost of moving a block from one stack to another is always at least 1, which is
      exactly the decrease in the heuristic value, if the block is moved to a stack with blocks
      of the same shade, otherwise the heuristic value of the successor node is the same or
      higher, and the heuristic estimate for the goal state is 0, as all the blocks in each
      stack are of the same shade in the goal state
    h = 0
    for stack in state:
        if stack:
            h += len([block for block in stack if block != stack[-1]])
    return h

print(a_star())
```

Table 14: The A* algorithm was generated by GPT4 using the MSMT A* approach. GPT4's third A* implementation for the restricted_sorting problem type passed the unit tests.

**restricted_sorting instance (diff_sorted_id: 57)**: In 'Restricted Sorting', there are 8 stacks each with a capacity of 5 blocks, with 4 stacks filled with blocks of varying shades and the remaining are empty. The challenge is to sort the blocks by shade such that if a stack is not empty, it is stacked with 4 blocks of a single shade. The player can only transfer one block at a time from the top of a stack to an empty stack or to a stack that has only blocks of that shade, without exceeding the stacks' capacity. Transferring blocks to certain stacks is more expensive than others. The cost of moving one block to the top of each stack is: 0: 2, 1: 3, 2: 9, 3: 8, 4: 3, 5: 7, 6: 9, 7: 5, where the keys are the index of each stack, indexing from 0. The cost of moving a block is always at least 1. The solution should be a list of tuples, each containing, first, the index of the stack from which a block is picked up from and, second, the index of the stack to which it is transferred, indexing from 0. Given the initial state of the stacks, represented by the lists below (with the leftmost item being the shade of the topmost block in each stack)(and the first stack being the stack at index 0), what is the list of transfer pairs (reported in python syntax) with the least possible cost, that will result in all the blocks being correctly sorted? [[], [], ['Green', 'Red', 'Black', 'Blue', 'Blue'], ['Yellow', 'Blue', 'Blue', 'Green', 'Black'], ['Yellow', 'Yellow', 'Yellow', 'Red', 'Black'], [], [], ['Red', 'Green', 'Green', 'Red', 'Black']]

'initialize' function generated by GPT4 for the problem above in the second stage of the MSMT approach

```python
def initialize():
    # Define the initial state of the stacks as a tuple of tuples
    initial_state = ((), (), ('Green', 'Red', 'Black', 'Blue', 'Blue'),
                     ('Yellow', 'Blue', 'Blue', 'Green', 'Black'),
                     ('Yellow', 'Yellow', 'Yellow', 'Red', 'Black'), (), (),
                     ('Red', 'Green', 'Green', 'Red', 'Black'))
    num_stacks = 8
    stack_capacity = 5
    # Define the cost of moving a block to each stack
    stack_costs = {0: 2, 1: 3, 2: 9, 3: 8, 4: 3, 5: 7, 6: 9, 7: 5}
    visited_costs = {}
    visited_costs[initial_state] = 0
    queue = [(0, 0, [], initial_state)]
    return initial_state, num_stacks, stack_capacity, stack_costs, visited_costs, queue
```

Table 15: GPT4 successfully generated a feasible (but not correct) solution for this restricted_sorting problem instance using the MSMT A* and A* prompting approach. The other three baseline code and text-based prompting methods resulted in infeasible solutions.

## COLOR SORTING

> **color_sorting instance (diff_sorted_id: 1)**: The game of 'Sort It' begins with 3 tubes, each filled with 4 balls of different colors. The goal is to sort the balls by color, with each tube containing balls of only one color. Only one ball can be moved at a time, taken from the top of one tube and placed on top of another. The capacity of each tube (maximum number of balls we can fit in each tube) is 6 balls. It is not allowed to place a ball in a tube that already has 6 balls. The solution should be a list of tuples, each containing, first, the index of the tube from which a ball is taken and, second, the index of the tube to which it is moved, indexing from 0. Given the initial state of the tubes, represented by the lists below (with the leftmost item being the color of the topmost ball in each tube), what is the shortest list of move tuples that will result in all the balls being correctly sorted? [['Green', 'Red', 'Green', 'Red'], ['Blue', 'Blue', 'Red', 'Green'], ['Red', 'Blue', 'Green', 'Blue']]

A* algorithm generated by GPT4 for the problem above using A* MSMT approach that passed the unit tests:

```python
import heapq

def initialize():
    # Define the initial state and the goal state of the tubes, represented as 3x4 tuples
    initial_state = (('Green', 'Red', 'Green', 'Red'), ('Blue', 'Blue', 'Red', 'Green'), ('Red', '
      Blue', 'Green', 'Blue'))
    goal_state = (('Red', 'Red', 'Red', 'Red'), ('Green', 'Green', 'Green', 'Green'), ('Blue', '
      Blue', 'Blue', 'Blue'))
    # Encoding other variables given in the problem statement
    num_tubes = 3
    tube_capacity = 6
    # Initialize a dictionary to store the cost of reaching each visited state
    visited_costs = {}
    visited_costs[initial_state] = 0
    # Initialize a priority queue of states not yet visited, with the initial state as the first
      element. The priority of each element is the cost to reach that state (g) + the estimate
      remaining cost (h) to reach the goal
    # Record the moves required to get to each state in a list; no moves performed to reach the
      initial state
    queue = [(0, 0, [], initial_state)]
    return initial_state, goal_state, num_tubes, tube_capacity, visited_costs, queue

def a_star():
    # The initialize function initializes and returns the visited_costs dictionary and the priority
      queue and encodes all of the variables given in the problem (ie the initial and goal state
      of the tubes and the tube capacity)
    initial_state, goal_state, num_tubes, tube_capacity, visited_costs, queue = initialize()
    # While there are un-visited states
    while queue:
        # Pop the state with the lowest sum of the cost so far and estimated cost to the goal from
          the queue
        _, g, actions, state = heapq.heappop(queue)
        # We can check if the current state is the goal state with a simple equality check, as the
          goal state is predefined
        if state == goal_state:
            return actions
        # Generate all valid actions from the current state, which includes moving the top ball
          from any tube to the top of any other tube
        for from_tube in range(num_tubes):
            for to_tube in range(num_tubes):
                # Check if the move is valid, ie if the from_tube is not empty and the to_tube is
                  not full
                if state[from_tube] and len(state[to_tube]) < tube_capacity:
                    # The action is valid, generate the new state
                    new_state = [list(tube[:]) for tube in state]
                    ball_to_be_moved = new_state[from_tube].pop(0)
                    # Do the move
                    new_state[to_tube].insert(0, ball_to_be_moved)
                    new_state = tuple(tuple(tube) for tube in new_state)
                    # The cost so far is the number of moves made, as our objective is to minimize
                      the number of moves required to reach the goal state
                    new_cost = g + 1
                    # If the new state is unvisited or we found a new path with a lower cost to
                      reach this state, add it to the queue of not-yet-visited states
                    if new_state not in visited_costs or new_cost < visited_costs[new_state]:
                        visited_costs[new_state] = new_cost
                        heapq.heappush(queue, (g + heuristic(new_state, goal_state), new_cost,
                          actions + [(from_tube, to_tube)], new_state))
    return None
```

The GPT4's implementation of the A* algorithm continues on the next page.

Continuation of GPT-4's implementation:

```python
def heuristic(state, goal):
    # An admissible and consistent heuristic is the sum of the number of misplaced balls in
      each tube
    # The heuristic relaxes the constraint that a ball can only be moved from the top of a tube
       and presumes we can move the balls to their goal position by taking them from any
      position in the tube
    # Thus the heuristic reports a lower estimate on the cost to reach goal state and is
      admissible
    # The heuristic is consistent because the cost of moving a ball to a neighboring tube is
      always 1, which is exactly the decrease in the number of misplaced balls, if the ball is
      moved toward its goal position, otherwise the estimated cost of the successor node is the
      same or higher, and the heuristic estimate for the goal state is 0, as the number of
      misplaced balls would be 0 in the goal state.
    h = 0
    for i in range(len(state)):
        for j in range(len(state[i])):
            # Can't compare integers with "_" when finding the goal position of each ball, thus
          ignore the "_" ball
            if state[i][j] != '_':
                # Add the the number of misplaced balls in each tube to the estimate
                h += sum(1 for ball in state[i] if ball != goal[i][0])
    return h

print(a_star())
```

Table 16: The A* algorithm was generated by GPT4 using the MSMT A* approach. GPT4's A* implementation for the first instance of the color_sorting problem type passed the unit tests.

**color_sorting instance (diff_sorted_id: 97)**: The game of 'Sort It' begins with 3 tubes, each filled with 6 balls of different colors. The goal is to sort the balls by color, with each tube containing balls of only one color. Only one ball can be moved at a time, taken from the top of one tube and placed on top of another. The capacity of each tube (maximum number of balls we can fit in each tube) is 9 balls. It is not allowed to place a ball in a tube that already has 9 balls. The solution should be a list of tuples, each containing, first, the index of the tube from which a ball is taken and, second, the index of the tube to which it is moved, indexing from 0. Given the initial state of the tubes, represented by the lists below (with the leftmost item being the color of the topmost ball in each tube), what is the shortest list of move tuples that will result in all the balls being correctly sorted? [['Red', 'Red', 'Blue', 'Red', 'Blue', 'Green'], ['Red', 'Green', 'Blue', 'Blue', 'Green', 'Red'], ['Blue', 'Green', 'Green', 'Red', 'Blue', 'Green']]

'initialize' function generated by GPT4 for the problem above in the second stage of the MSMT approach

```
def initialize():
    # Define the initial state and the goal state of the tubes, represented as 3x6 tuples
    initial_state = (('Red', 'Red', 'Blue', 'Red', 'Blue', 'Green'),
                     ('Red', 'Green', 'Blue', 'Blue', 'Green', 'Red'),
                     ('Blue', 'Green', 'Green', 'Red', 'Blue', 'Green'))
    goal_state = (('Red', 'Red', 'Red', 'Red', 'Red', 'Red'),
                  ('Green', 'Green', 'Green', 'Green', 'Green', 'Green'),
                  ('Blue', 'Blue', 'Blue', 'Blue', 'Blue', 'Blue'))
    # Encoding other variables given in the problem statement
    num_tubes = 3
    tube_capacity = 9
    # Initialize a dictionary to store the cost of reaching each visited state
    visited_costs = {}
    visited_costs[initial_state] = 0
    # Initialize a priority queue of states not yet visited, with the initial state as the
      first element. The priority of each element is the cost to reach that state (g) + the
      estimate remaining cost (h) to reach the goal
    # Record the moves required to get to each state in a list; no moves performed to reach the
      initial state
    queue = [(0, 0, [], initial_state)]
    return initial_state, goal_state, num_tubes, tube_capacity, visited_costs, queue
```

Table 17: GPT4 successfully generated a correct (but non-optimal) solution for this color_sorting problem instance using MSMT A* approach. Prompting GPT4 with all other four code and text-based prompting methods resulted in infeasible solutions.

MAGIC SQUARE

> **magic_square instance (diff_sorted_id: 1)**: The game of 'Sort It' begins with 3 tubes, each filled with 4 balls of different colors. The goal is to sort the balls by color, with each tube containing balls of only one color. Only one ball can be moved at a time, taken from the top of one tube and placed on top of another. The capacity of each tube (maximum number of balls we can fit in each tube) is 6 balls. It is not allowed to place a ball in a tube that already has 6 balls. The solution should be a list of tuples, each containing, first, the index of the tube from which a ball is taken and, second, the index of the tube to which it is moved, indexing from 0. Given the initial state of the tubes, represented by the lists below (with the leftmost item being the color of the topmost ball in each tube), what is the shortest list of move tuples that will result in all the balls being correctly sorted? [['Green', 'Red', 'Green', 'Red'], ['Blue', 'Blue', 'Red', 'Green'], ['Red', 'Blue', 'Green', 'Blue']]

A* algorithm generated by GPT4 for the problem above using A* MSMT approach that passed the unit tests:

```python
import heapq

def initialize():
    # Define the initial state of the grid, the goal sums of the specified row, column, and
     diagonal, and the range of unique integers that can be used to replace 'x's
    initial_state = (('35', 'x', '46'), ('x', 'x', 'x'), ('x', 'x', 'x'))
    goal_sums = {'row1': 103, 'col1': 107, 'diag': 124}
    num_range = set(range(29, 49))
    # Initialize a dictionary to store the cost of reaching each visited state
    visited_costs = {}
    visited_costs[initial_state] = 0
    # Initialize a priority queue of states not yet visited, with the initial state as the first
     element. The priority of each element is the cost to reach that state (g) + the estimate
     remaining cost (h) to reach the goal
    # Record the actions required to get to each state in a list; no actions performed to reach
     the initial state
    queue = [(0, 0, [], initial_state)]
    return initial_state, goal_sums, num_range, visited_costs, queue

def a_star():
    # The initialize function initializes and returns the visited_costs dictionary and the
     priority queue and encodes all of the variables given in the problem (ie the initial state of
      the grid, the goal sums, and the range of unique integers)
    initial_state, goal_sums, num_range, visited_costs, queue = initialize()
    # While there are un-visited states
    while queue:
        # Pop the state with the lowest sum of the cost so far and estimated cost to the goal from
         the queue
        _, g, actions, state = heapq.heappop(queue)
        # Check if the current state is the goal state
        # The goal state is where the sum of the specified row, column, and diagonal equal the
         goal sums, and there are no 'x's in the grid
        if all(cell != 'x' for row in state for cell in row) and sum(int(cell) for cell in state
        [1]) == goal_sums['row1'] and sum(int(state[i][1]) for i in range(3)) == goal_sums['col1']
        and sum(int(state[i][2-i]) for i in range(3)) == goal_sums['diag']:
            return actions
        # If the state has at least 1 remaining unknown number, ie 'x', generate all possible
        actions from the current state, which includes replacing the next x in the grid with any of
        avaiable unique integers (all numbers in the range - numbers present in the current state) in
         the range
        if any(cell == 'x' for row in state for cell in row):
            # Find the next 'x' in the grid
            for i in range(3):
                for j in range(3):
                    if state[i][j] == 'x':
                        # Generate the set of available numbers by subtracting the set of numbers
        present in the current state from the set of all numbers in the range
                        available_nums = num_range - set(int(cell) for row in state for cell in
        row if cell != 'x')
                        for num in available_nums:
                            # Generate the new state
                            new_state = [list(row) for row in state]
                            new_state[i][j] = str(num)
                            new_state = tuple(tuple(row) for row in new_state)
                            # The cost so far is the sum of the numbers in the grid, as the task
        is to minimize the sum of all of the numbers in the grid
                            new_cost = g + num
                            # If the new state is unvisited or we found a new path with a lower
        cost to arrive at this state, add it to the queue of un-visited states
```

The GPT4's implementation of the A* algorithm continues on the next page.

**Continuation of GPT-4's implementation:**

```python
                        if new_state not in visited_costs or new_cost < visited_costs[
    new_state]:
                            visited_costs[new_state] = new_cost
                            heapq.heappush(queue, (new_cost + heuristic(new_state,
    goal_sums), new_cost, actions + [(i, j, num)], new_state))
                        # Stop searching for 'x's after the first 'x' is found and all
    possible actions from it are generated
                        break
                else:
                    continue
                break
    return None

def heuristic(state, goal_sums):
    # An admissible and consistent heuristic for this problem is the sum of the absolute
     differences between the current sums and the goal sums of the specified row, column, and
     diagonal
    # The heuristic relaxes the constraints that each number in the grid must be unique and
     that the sum of all of the numbers in the grid must be minimized; ie It presumes we can
     add or subtract any amount to the current sums to reach the goal sums
    # Thus the heuristic reports a lower estimate on the cost of reaching the goal state and
     is admissible
    # The heuristic is consistent because the estimated cost from the current state to the
     goal can never be greater than the sum of cost from the current node to a successor node
     plus the estimated cost from the successor node to the goal because the cost of reaching a
      successor node is the number added to the grid, which is always greater than or equal to
     1, the decrease in the absolute difference between the current and goal sums
    h = 0
    if any(cell == 'x' for row in state for cell in row):
        h += abs(sum(int(cell) for cell in state[1] if cell != 'x') - goal_sums['row1'])
        h += abs(sum(int(state[i][1]) for i in range(3) if state[i][1] != 'x') - goal_sums['
    col1'])
        h += abs(sum(int(state[i][2-i]) for i in range(3) if state[i][2-i] != 'x') - goal_sums
    ['diag'])
    return h

print(a_star())
```

Table 18: The A* algorithm was generated by GPT4 using the MSMT A* approach. GPT4's first attempt at implementing the A* algorithm for the first instance of the magic_square problem type passed the unit tests.

**magic_square instance (diff_sorted_id: 34)**: In the magic square problem, a 3x3 grid is filled with unique integers ranging from 40 to 89. Some numbers are already given, while others are unknown and represented as 'x'. Sum of column 1 (counting from 0) must be 183, and sum of row 1 must be 194. Also, the sum of the numbers in the diagonal from the top right to the bottom left corner of the grid should equal 161. The goal is to find unique integers (ie each number can be in the final grid only once) in the given range to replace with 'x's in the grid below such that the sum of the specified rows, columns, and diagonal equals the given amounts and the sum of all of the numbers in the grid is as low as possible. The solution should be provided as a list of tuples in Python syntax. Each tuple should contain three numbers for each 'x' position: the row index, the column index (both starting from 0), and the value of the unique integer replaced with 'x' at that position.\n\nGrid:\n [['x' 'x' 'x']\n ['x' 'x' 'x']\n ['40' 'x' '71']]

'initialize' function generated by GPT4 for the problem above in the second stage of the MSMT approach

```
def initialize():
    # Define the initial state of the grid, the goal sums of the specified row, column, and
     diagonal, and the range of unique integers that can be used to replace 'x's
    initial_state = (('x', 'x', 'x'), ('x', 'x', 'x'), ('40', 'x', '71'))
    goal_sums = {'row1': 194, 'col1': 183, 'diag': 161}
    num_range = set(range(40, 90))
    # Initialize a dictionary to store the cost of reaching each visited state
    visited_costs = {}
    visited_costs[initial_state] = 0
    # Initialize a priority queue of states not yet visited, with the initial state as the first
     element. The priority of each element is the cost to reach that state (g) + the estimate
     remaining cost (h) to reach the goal
    # Record the actions required to get to each state in a list; no actions performed to reach
     the initial state
    queue = [(0, 0, [], initial_state)]
    return initial_state, goal_sums, num_range, visited_costs, queue
```

Table 19: GPT4 successfully generated a correct (but non-optimal) solution for this magic_square problem instance using MSMT A* approach. Prompting GPT4 with all other four code and text-based prompting methods resulted in infeasible solutions.

## CONSECUTIVE GRID

---

**consecutive_grid instance (diff_sorted_id: 13)**: We have a 3x3 numerical grid, with numbers ranging from 21 to 59 (21 included in the range but 59 is not included). The numbers in each row and column must be strictly increasing or decreasing. This means that either first > second > third or first < second < third in each row and column. If a grid cell is marked with an 'x', the number in that position is hidden. The objective is to replace the 'x's with unique integers from the given range, ensuring that each number only appears once in the grid. The replacements must maintain the consecutive order in each row and column. Additionally, the sum of the numbers in the topmost row plus the numbers in the rightmost column plus the numbers in the diagonal connecting the top-left corner of the grid to its bottom-right corner should be minimized. The solution should be given as a list of tuples in Python syntax. Each tuple should represent the replacement of a number with an 'x' number and contain three elements: the row index of the 'x', the column index of the 'x' (both starting from 0), and the value of the number that replaces the 'x'. The initial state of the grid is as follows: \n\nGrid:\n [['29' '45' 'x']\n ['x' 'x' 'x']\n ['x' '43' '34']]

---

A* algorithm generated by GPT4 for the problem above using A* MSMT approach that passed the unit tests:

```python
import heapq
import numpy as np

def initialize():
    # Define the initial state of the grid, as a 2d tuple
    initial_state = (('29', '45', 'x'), ('x', 'x', 'x'), ('x', '43', '34'))
    # Define the range of numbers that can be used to replace 'x'
    num_range = set(range(21, 59))
    # Define the dimensions of the grid
    num_rows = 3
    num_cols = 3
    # Initialize a dictionary to store the cost of reaching each visited state
    visited_costs = {}
    visited_costs[initial_state] = 0
    # Initialize a priority queue of states not yet visited, with the initial state as the first
    # element. The priority of each element is the cost to reach that state (g) + the estimate
    # remaining cost (h) to reach the goal
    # Record the actions required to get to each state in a list; no actions performed to reach the
    # initial state
    queue = [(0, 0, [], initial_state)]
    return initial_state, num_range, num_rows, num_cols, visited_costs, queue

def a_star():
    # The initialize function initializes and returns the visited_costs dictionary and the priority
    # queue and encodes all of the variables given in the problem (ie the initial state of the grid,
    # the range of numbers, and the dimensions of the grid)
    initial_state, num_range, num_rows, num_cols, visited_costs, queue = initialize()
    # While there are un-visited states
    while queue:
        # Pop the state with the lowest sum of the cost so far and estimated cost to the goal from
        # the queue
        _, g, actions, state = heapq.heappop(queue)
        # If the current state does not have any 'x's, ie all the numbers have been filled in, and
        # the numbers in each row and column are consecutive, return the actions taken to reach this
        # state
        if all(all(cell != 'x' for cell in row) for row in state) and all(all(int(row[i]) < int(row[
        i + 1]) for i in range(len(row) - 1)) or all(int(row[i]) > int(row[i + 1]) for i in range(len(
        row) - 1)) for row in state) and all(all(int(state[i][col]) < int(state[i + 1][col]) for i in
        range(len(state) - 1)) or all(int(state[i][col]) > int(state[i + 1][col]) for i in range(len(
        state) - 1)) for col in range(len(state[0]))):
            return actions
        # If the state has at least 1 remaining unknown number, ie 'x', generate all possible
        actions from the current state, which includes replacing the next x in the grid with any of
        avaiable unique integers (all numbers in the range - numbers present in the current state) in
        the range
        if any('x' in row for row in state):
            # Find the next 'x' in the grid
            for row_ind in range(num_rows):
                for col_ind in range(num_cols):
                    if state[row_ind][col_ind] == 'x':
                        # Generate all possible actions from the current state, which includes
        replacing the 'x' with any of the available unique integers in the range
                        for num in num_range - set(cell for row in state for cell in row if cell !=
        'x'):
                            # Generate the new state
                            new_state = [list(row[:]) for row in state]
                            new_state[row_ind][col_ind] = num
                            new_state = tuple(tuple(row) for row in new_state)
                            # The cost so far is the sum of the numbers in the topmost row, the
        rightmost column, and the diagonal from the top-left corner to the bottom-right corner of the
        grid
```

The GPT4's implementation of the A* algorithm continues on the next page.

**Continuation of GPT-4's implementation:**

```python
                        new_cost = sum(int(cell) for cell in new_state[0] if cell != 'x')
        + sum(int(new_state[i][-1]) for i in range(num_rows) if new_state[i][-1] != 'x') + sum(int
        (new_state[i][i]) for i in range(num_rows) if new_state[i][i] != 'x')
                        # If the new state is unvisited or we found a new path with a
        lower cost to reach this state, add it to the queue of not-yet-visited states
                        if new_state not in visited_costs or new_cost < visited_costs[
        new_state]:
                            visited_costs[new_state] = new_cost
                            heapq.heappush(queue, (new_cost + heuristic(new_state),
        new_cost, actions + [(row_ind, col_ind, num)], new_state))
                    # Once we have generated all possible actions from the current state,
        break the loop
                        break
                else:
                    continue
                break
        return None

def heuristic(state):
    # An admissible and consistent heuristic for this problem is the sum of the minimum
     remaining numbers that can be used to replace the 'x's in the grid
    # This heuristic is admissible because it never overestimates the cost to reach the goal,
     as the minimum remaining numbers are always less than or equal to the actual numbers that
     will replace the 'x's
    # The heuristic is consistent because the estimated cost from the current state to the
     goal can never be greater than the sum of cost from the current node to a successor node
     plus the estimated cost from the successor node to the goal because the cost of replacing
     an 'x' with a number is the number itself, which is always greater than or equal to the
     minimum remaining number
    return sum(sorted(set(range(21, 59)) - set(int(cell) for row in state for cell in row if
     cell != 'x'))[:sum(cell == 'x' for row in state for cell in row)])

print(a_star())
```

Table 20: The A* algorithm was generated by GPT4 using the MSMT A* approach. GPT4's 13-th attempt at implementing the A* algorithm for consecutive_grid problem type passed the unit tests.

**consecutive_grid instance (diff_sorted_id: 43)**: We have a 3x3 numerical grid, with numbers ranging from 7 to 60 (7 included in the range but 60 is not included). The numbers in each row and column must be strictly increasing or decreasing. This means that either first > second > third or first < second < third in each row and column. If a grid cell is marked with an 'x', the number in that position is hidden. The objective is to replace the 'x's with unique integers from the given range, ensuring that each number only appears once in the grid. The replacements must maintain the consecutive order in each row and column. Additionally, the sum of the numbers in the topmost row plus the numbers in the rightmost column plus the numbers in the diagonal connecting the top-left corner of the grid to its bottom-right corner should be minimized. The solution should be given as a list of tuples in Python syntax. Each tuple should represent the replacement of a number with an 'x' number and contain three elements: the row index of the 'x', the column index of the 'x' (both starting from 0), and the value of the number that replaces the 'x'. The initial state of the grid is as follows: \n\nGrid:\n [['13' 'x' 'x']\n ['x' 'x' '27']\n ['x' '26' 'x']]

'initialize' function generated by GPT4 for the problem above in the second stage of the MSMT approach

```python
def initialize():
    # Define the initial state of the grid, as a 2d tuple
    initial_state = (('13', 'x', 'x'), ('x', 'x', '27'), ('x', '26', 'x'))
    # Define the range of numbers that can be used to replace 'x'
    num_range = set(range(7, 60))
    # Define the dimensions of the grid
    num_rows = 3
    num_cols = 3
    # Initialize a dictionary to store the cost of reaching each visited state
    visited_costs = {}
    visited_costs[initial_state] = 0
    # Initialize a priority queue of states not yet visited, with the initial state as the first
    #  element. The priority of each element is the cost to reach that state (g) + the estimate
    #  remaining cost (h) to reach the goal
    # Record the actions required to get to each state in a list; no actions performed to reach
    #  the initial state
    queue = [(0, 0, [], initial_state)]
    return initial_state, num_range, num_rows, num_cols, visited_costs, queue
```

Table 21: GPT4 successfully generated an optimal and correct solution for this consecutive_grid problem instance using MSMT A* approach. Prompting GPT4 with all other four code and text-based prompting methods resulted in infeasible solutions.

TRAFFIC

**traffic instance (diff_sorted_id: 1)**: Using the provided matrix map of a city, where numbers represent travel time in minutes (all numbers are positive integers) and 'x' marks closed workshops, find the quickest route for Ben to travel from his current workshop at index (5, 7) to his destination workshop at index (2, 1), indexing from 0. Ben's car can move north, south, east, or west from a given crossroad, provided there's no x in that direction. Also, there are 3 districts in the city with district 1 covering rows 0 to 2, district 2 covering rows 3 to 4, and district 3 covering rows 5 to 8. Ben has to visit at least 1 workshop in each district on his path to the destination. The roads are bidirectional. The answer should be a list of tuples (in Python syntax) indicating the index of workshops on Ben's path. The start and end workshops must be included in the path.\n[13 x x 3 10 20 3 x 5]\n[x x x x 20 16 x x 17]\n[3 14 18 8 1 20 14 x 7]\n[13 3 6 10 7 4 6 6 1]\n[10 12 2 x 11 x 10 8 11]\n[x x x 11 6 18 13 20 17]\n[x x 16 x 4 17 7 10 15]\n[x 7 16 6 19 4 7 x x]\n[x 11 18 x x x 3 8 x]

A* algorithm generated by GPT4 for the problem above using A* MSMT approach that passed the unit tests:

```python
import heapq

def initialize():
    # Define the initial state and the goal state of the problem, represented as tuples
    initial_state = (5, 7)
    goal_state = (2, 1)
    # Encoding other variables given in the problem statement
    num_rows = 9
    num_cols = 9
    matrix = [[13, 'x', 'x', 3, 10, 20, 3, 'x', 5],['x', 'x', 'x', 'x', 20, 16, 'x', 'x', 17],[3,
        14, 18, 8, 1, 20, 14, 'x', 7],[13, 3, 6, 10, 7, 4, 6, 6, 1],[10, 12, 2, 'x', 11, 'x', 10, 8,
        11],['x', 'x', 'x', 11, 6, 18, 13, 20, 17],['x', 'x', 16, 'x', 4, 17, 7, 10, 15],['x', 7, 16,
        6, 19, 4, 7, 'x', 'x'], ['x', 11, 18, 'x', 'x', 'x', 3, 8, 'x']]
    districts = {1: range(0, 3), 2: range(3, 5), 3: range(5, 9)}
    # Initialize a dictionary to store the cost of reaching each visited state
    visited_costs = {}
    visited_costs[initial_state] = 0
    # Initialize a priority queue of states not yet visited, with the initial state as the first
      element. The priority of each element is the cost to reach that state (g) + the estimate
      remaining cost (h) to reach the goal
    # Record the actions required to get to each state in a list; no actions performed to reach the
      initial state
    queue = [(0, 0, [], initial_state, {1: False, 2: False, 3: True})]
    return initial_state, goal_state, num_rows, num_cols, matrix, districts, visited_costs, queue

def a_star():
    initial_state, goal_state, num_rows, num_cols, matrix, districts, visited_costs, queue =
      initialize()
    while queue:
        _, g, actions, state, visited_districts = heapq.heappop(queue)
        # Check if the current state is the goal state
        if state == goal_state and all(visited_districts.values()):
            return actions
        # Generate all possible actions from the current state, which includes moving to any of the
        4 neighboring cells
        for d_row, d_col in [(0, -1), (0, 1), (1, 0), (-1, 0)]:
            new_row, new_col = state[0] + d_row, state[1] + d_col
            # Check if the new state would be valid, ie if the coordinate of the cell to move to is
        a valid coordinate within the bounds of the matrix and the cell is not 'x'
            if 0 <= new_row < num_rows and 0 <= new_col < num_cols and matrix[new_row][new_col] !=
        'x':
                # Generate the new state
                new_state = (new_row, new_col)
                # The cost so far is the sum of the travel times, as our objective is to minimize
        the total travel time
                new_cost = g + matrix[new_row][new_col]
                # Update the visited districts
                new_visited_districts = visited_districts.copy()
                for district, rows in districts.items():
                    if new_row in rows:
                        new_visited_districts[district] = True
                # If the new state is unvisited or we found a new path with a lower cost to reach
        this state, add it to the queue of not-yet-visited states
                if new_state not in visited_costs or new_cost < visited_costs[new_state]:
                    visited_costs[new_state] = new_cost
                    heapq.heappush(queue, (g + heuristic(new_state, goal_state), new_cost, actions
        + [new_state], new_state, new_visited_districts))
    return None
```

45

The GPT4's implementation of the A* algorithm continues on the next page.

**Continuation of GPT-4's implementation:**

```python
def heuristic(state, goal):
    # An admissible and consistent heuristic is the Manhattan distance (the shortest path) of
      the current position from the goal position
    # The heuristic relaxes the constraint that a cell can only be moved to if it's not 'x' and
       presumes we can move to the goal position by moving to any of the neighboring cells
    # Thus the heuristic reports a lower estimate on the cost to reach goal state and is
      admissible
    # The heuristic is consistent because the cost of moving to a neighboring coordinate is
      always greater than or equal to 1, which is exactly the decrease in the Manhattan distance
      , if the cell is moved toward its goal position, otherwise the estimated cost of the
      successor node is the same or higher, and he heuristic estimate for the goal state is 0,
      as the distance of the goal position from itself would be 0.
    h = abs(state[0] - goal[0]) + abs(state[1] - goal[1])
    return h

print(a_star())
```

Table 22: The A* algorithm was generated by GPT4 using the MSMT A* approach. GPT4's first implementation of the A* algorithm for traffic problem type passed the unit tests.

**traffic instance (diff_sorted_id: 104)**: Using the provided matrix map of a city, where numbers represent travel time in minutes (all numbers are positive integers) and 'x' marks closed workshops, find the quickest route for Ben to travel from his current workshop at index (7, 2) to his destination workshop at index (4, 14), indexing from 0. Ben's car can move north, south, east, or west from a given crossroad, provided there's no x in that direction. Also, there are 3 districts in the city with district 1 covering rows 0 to 4, district 2 covering rows 5 to 8, and district 3 covering rows 9 to 14. Ben has to visit at least 1 workshop in each district on his path to the destination. The roads are bidirectional. The answer should be a list of tuples (in Python syntax) indicating the index of workshops on Ben's path. The start and end workshops must be included in the path.\n[x x 7 18 11 4 9 x 15 x x x 1 x x]\n[x x 14 x 10 x x x 8 x x 16 1 x x]\n[17 17 x x x x x 7 6 12 x x x 7 x]\n[x x x x 10 x x 18 1 11 4 x 13 10 16]\n[x x x 12 1 x x x 2 7 1 x 2 x 5]\n[x x 8 x 12 x x 5 17 x 2 x 4 10 18]\n[x x 4 19 1 x 18 7 x 10 3 x 19 16 19]\n[6 12 15 16 5 9 16 18 10 15 5 x 5 14 x]\n[x 16 8 17 12 11 16 8 9 9 7 4 5 20 3]\n[19 17 15 x x 17 x 3 2 2 11 7 8 16 1]\n[13 4 17 x x x x 5 x 18 16 15 19 4 x]\n[11 x x x x x x 19 x x 4 3 x 3 x 16 18 x 12]\n[x 13 x x x x x 10 5 x 17 x x 7 x]

---

'initialize' function generated by GPT4 for the problem above in the second stage of the MSMT approach

```python
def initialize():
    # Define the initial state and the goal state of the problem, represented as tuples
    initial_state = (7, 2)
    goal_state = (4, 14)

    # Encoding other variables given in the problem statement
    num_rows = 15
    num_cols = 15
    matrix = [[ 'x', 'x', 7, 18, 11, 4, 9, 'x', 15, 'x', 'x', 'x', 1, 'x', 'x'], [ 'x', 'x', 14, 'x
        ', 10, 'x', 'x', 'x', 8, 'x', 'x', 16, 1, 'x', 'x'], [ 17, 17, 'x', 'x', 'x', 'x', 'x', 7, 6,
        12, 'x', 'x', 'x', 7, 'x'], [ 'x', 'x', 'x', 'x', 10, 'x', 'x', 18, 1, 11, 4, 'x', 13, 10,
        16], [ 'x', 'x', 'x', 12, 1, 'x', 'x', 'x', 2, 7, 1, 'x', 2, 'x', 5], [ 'x', 'x', 8, 'x', 12,
        'x', 'x', 5, 17, 'x', 2, 'x', 4, 10, 18], [ 'x', 'x', 4, 19, 1, 'x', 18, 7, 'x', 10, 3, 'x',
        19, 16, 19], [ 6, 12, 15, 16, 5, 9, 16, 18, 10, 15, 5, 'x', 5, 14, 'x'], [ 'x', 16, 8, 17,
        12, 11, 16, 8, 9, 9, 7, 4, 5, 20, 3], [ 19, 17, 15, 'x', 'x', 17, 'x', 3, 2, 2, 11, 7, 8, 16,
        1], [ 13, 4, 17, 'x', 'x', 'x', 'x', 5, 'x', 18, 16, 15, 19, 4, 'x'], [ 11, 'x', 'x', 'x', '
        x', 12, 'x', 'x', 'x', 'x', 'x', 'x', 19, 'x', 'x'], [ 18, 'x', 'x', 'x', 9, 19, 'x', 16, 6,
        'x', 9, 3, 16, 15, 'x'], [ 'x', 'x', 'x', 'x', 10, 'x', 'x', 4, 3, 'x', 3, 'x', 16, 18, 'x', 12], [
        'x', 13, 'x', 'x', 'x', 'x', 'x', 10, 5, 'x', 17, 'x', 'x', 7, 'x']]
    districts = {1: range(0, 5), 2: range(5, 9), 3: range(9, 15)}
    # Initialize a dictionary to store the cost of reaching each visited state
    visited_costs = {}
    visited_costs[initial_state] = 0
    # Initialize a priority queue of states not yet visited, with the initial state as the first
      element. The priority of each element is the cost to reach that state (g) + the estimate
      remaining cost (h) to reach the goal
    # Record the actions required to get to each state in a list; no actions performed to reach the
        initial state
    queue = [(0, 0, [initial_state], initial_state, {1: False, 2: True, 3: False})]
    return initial_state, goal_state, num_rows, num_cols, matrix, districts, visited_costs, queue
```

Table 23: GPT4 successfully generated an optimal and correct solution for this traffic problem instance using MSMT A* approach. Prompting GPT4 with A* results in a feasible but incorrect solution and all other three baseline code and text-based prompting methods resulted in infeasible solutions.

## TRAMPOLINE MATRIX

**trampoline_matrix instance (diff_sorted_id: 4)**: Alex is at a trampoline park with a grid of mini trampolines, arranged in a square of 9x9. Some trampolines are broken and unusable. A map of the park is provided below, with 1 indicating a broken trampoline and 0 indicating a functional one. Alex can jump to any of the eight adjacent trampolines, as long as they are not broken. However, Alex must make exactly 3 diagonal jumps, no more, no less, on his path to his destination. He is currently on the trampoline at position (8, 1) (positions are counted from 0, left to right, top to bottom) and wants to reach the trampoline at position (0, 5). What is the shortest sequence of trampolines he should jump on to reach his destination (including the first and final trampolines)? The answer should be a list of tuples, in Python syntax, indicating the row and column of each trampoline Alex jumps on. \n0 0 1 0 0 0 0 0 1\n0 1 0 1 1 0 0 0 0\n1 1 1 1 1 1 1 1 0\n1 1 0 1 1 0 0 0 0\n1 0 1 0 1 0 1 1 1\n1 0 1 0 0 0 1 0 1\n0 1 0 0 0 1 1 0 1\n0 0 1 0 1 1 1 1 1\n1 0 0 1 1 0 0 1 1

A* algorithm generated by GPT4 for the problem above using A* MSMT approach that passed the unit tests:

```python
import heapq

def initialize():
    # Define the initial state and the goal state of the problem, represented as tuples
    initial_state = (8, 1)
    goal_state = (0, 5)
    # Encoding other variables given in the problem statement
    num_rows = 9
    num_cols = 9
    park_map = [[0, 0, 1, 0, 0, 0, 0, 0, 1],[0, 1, 0, 1, 1, 0, 0, 0, 0],[1, 1, 1, 1, 1, 1, 1, 1,
      0],[1, 1, 0, 1, 1, 0, 0, 0, 0],[1, 0, 1, 0, 1, 0, 1, 1, 1],[1, 0, 1, 0, 0, 0, 1, 0, 1],[0, 1,
      0, 0, 0, 1, 1, 0, 1],[0, 0, 1, 0, 1, 1, 1, 1, 1],[1, 0, 0, 1, 1, 0, 0, 1, 1]]
    diagonal_jumps = 3
    # Initialize a dictionary to store the cost of reaching each visited state
    visited_costs = {}
    visited_costs[(initial_state, diagonal_jumps)] = 0
    # Initialize a priority queue of states not yet visited, with the initial state as the first
    #   element. The priority of each element is the cost to reach that state (g) + the estimate
    #   remaining cost (h) to reach the goal
    # Record the actions required to get to each state in a list; no actions performed to reach the
    #   initial state
    queue = [(0, 0, [], (initial_state, diagonal_jumps))]
    return initial_state, goal_state, num_rows, num_cols, park_map, diagonal_jumps, visited_costs,
      queue

def a_star():
    # The initialize function initializes and returns the visited_costs dictionary and the priority
    #   queue and encodes all of the variables given in the problem (ie the initial and goal state,
    #   dimensions of the park, park map, and number of diagonal jumps)
    initial_state, goal_state, num_rows, num_cols, park_map, diagonal_jumps, visited_costs, queue =
      initialize()
    # While there are un-visited states
    while queue:
        # Pop the state with the lowest sum of the cost so far and estimated cost to the goal from
      the queue
        _, g, actions, (state, diagonal_jumps_left) = heapq.heappop(queue)
        # We can check if the current state is the goal state with a simple equality check, as the
      goal state is predefined
        if state == goal_state and diagonal_jumps_left == 0:
            return actions
        # Generate all valid actions from the current state, which includes jumping to any of the 8
       neighboring trampolines
        # Generate the coordinates of the neighboring trampolines
        for d_row, d_col in [(0, -1), (0, 1), (1, 0), (-1, 0), (1, 1), (-1, -1), (1, -1), (-1, 1)]:
            new_row, new_col = state[0] + d_row, state[1] + d_col
            # Check if the jump is valid, ie if the coordinate of the trampoline to jump to is a
      valid coordinate within the bounds of the park and the trampoline is not broken
            if 0 <= new_row < num_rows and 0 <= new_col < num_cols and park_map[new_row][new_col]
      == 0:
                # The jump is valid, generate the new state
                new_state = (new_row, new_col)
                # The cost so far is the number of jumps made, as our objective is to minimize the
      number of jumps required to reach the goal state
                new_cost = g + 1
                # If the jump is diagonal, decrement the number of diagonal jumps left
                if d_row != 0 and d_col != 0:
                    new_diagonal_jumps_left = diagonal_jumps_left - 1
                else:
                    new_diagonal_jumps_left = diagonal_jumps_left
                # If the new state is unvisited or we found a new path with a lower cost to reach
      this state, add it to the queue of not-yet-visited states
```

The GPT4's implementation of the A* algorithm continues on the next page.

**Continuation of GPT-4's implementation:**

```python
                if (new_state, new_diagonal_jumps_left) not in visited_costs or new_cost <
    visited_costs[(new_state, new_diagonal_jumps_left)]:
                    visited_costs[(new_state, new_diagonal_jumps_left)] = new_cost
                    heapq.heappush(queue, (new_cost + heuristic(new_state, goal_state),
    new_cost, actions + [new_state], (new_state, new_diagonal_jumps_left)))
    return None

def heuristic(state, goal):
    # An admissible and consistent heuristic is the Manhattan distance (the shortest path) of
      the current position from the goal position
    # The heuristic relaxes the constraint that Alex must make exactly 3 diagonal jumps and
      presumes Alex can move to the goal position by jumping to any of the neighboring
      trampolines
    # Thus the heuristic reports a lower estimate on the cost to reach goal state and is
      admissible
    # The heuristic is consistent because the cost of moving to a neighboring coordinate is
      always 1, which is exactly the decrease in the Manhattan distance, if Alex moves toward
      the goal position, otherwise the estimated cost of the successor node is the same or
      higher, and he heuristic estimate for the goal state is 0, as the distance of the current
      position from the goal position would be 0 in the goal state.
    h = abs(state[0] - goal[0]) + abs(state[1] - goal[1])
    return h

print(a_star())
```

Table 24: The A* algorithm was generated by GPT4 using the MSMT A* approach. GPT4's fourth implementation of the A* algorithm for trampoline_matrix problem type passed the unit tests.

**trampoline_matrix instance (diff_sorted_id: 102)**: Alex is at a trampoline park with a grid of mini trampolines, arranged in a square of 15x15. Some trampolines are broken and unusable. A map of the park is provided below, with 1 indicating a broken trampoline and 0 indicating a functional one. Alex can jump to any of the eight adjacent trampolines, as long as they are not broken. However, Alex must make excatly 5 diagonal jumps, no more, no less, on his path to his destination. He is currently on the trampoline at position (14, 13) (positions are counted from 0, left to right, top to bottom) and wants to reach the trampoline at position (2, 5). What is the shortest sequence of trampolines he should jump on to reach his destination (including the first and final trampolines)? The answer should be a list of tuples, in Python syntax, indicating the row and column of each trampoline Alex jumps on. \n1 1 0 1 1 1 0 0 0 1 0 1 0 1 0\n0 1 0 0 1 0 0 0 0 1 0 1 1 1 0\n1 1 1 1 0 0 0 0 0 1 1 0 0 1\n1 1 1 1 0 0 1 1 0 1 1 1 1 0 0\n1 1 0 1 0 0 1 0 0 0 0 0 0 0 0\n1 0 1 0 1 1 1 0 1 0 1 0 0 1\n1 0 0 0 0 0 0 1 0 0 1 0 1 1\n0 1 1 1 1 1 0 0 1 1 0 1 0 1 0 1 1\n0 0 1 1 1 0 0 1 1 0 0 1 0 0 1\n1 1 1 1 1 0 1 0 0 1 1 0 0 0 0\n1 0 0 1 0 1 1 0 0 0 1 1 1 0 0\n0 0 0 1 0 0 1 1 0 1 0 0 0 0 0\n0 1 1 1 1 0 0 0 1 1 1 0 0 0 0\n1 1 1 1 1 0 0 0 1 1 0 0 0 0\n1 1 1 0 0 1 1 0 1 1 1 0 1 0 1

'initialize' function generated by GPT4 for the problem above in the second stage of the MSMT approach

```python
def initialize():
    # Define the initial state and the goal state of the problem, represented as tuples
    initial_state = (14, 13)
    goal_state = (2, 5)
    # Encoding other variables given in the problem statement
    num_rows = 15
    num_cols = 15
    park_map = [[1, 1, 0, 1, 1, 1, 0, 0, 0, 1, 0, 1, 0, 1, 0],[0, 1, 0, 0, 1, 0, 0, 0, 0, 1, 0, 1,
      1, 1, 0],[1, 1, 1, 1, 0, 0, 0, 0, 0, 0, 1, 1, 0, 0, 1],[1, 1, 1, 1, 0, 0, 1, 1, 0, 1, 1, 1,
      1, 0, 0],[1, 1, 0, 1, 0, 0, 1, 0, 0, 0, 0, 0, 0, 0],[1, 0, 1, 0, 1, 1, 1, 0, 1, 1, 0, 1,
      0, 0, 1],[1, 1, 0, 0, 0, 0, 0, 0, 1, 0, 0, 1, 0, 1, 1],[0, 1, 1, 1, 1, 1, 0, 0, 1, 1, 0, 1,
      0, 1, 1],[0, 0, 1, 1, 1, 0, 0, 1, 1, 0, 0, 1, 0, 0, 1],[1, 1, 1, 1, 1, 0, 1, 0, 0, 1, 1, 0,
      0, 0, 0],[1, 0, 0, 1, 0, 1, 1, 0, 0, 0, 1, 1, 1, 0, 0],[0, 0, 0, 1, 0, 0, 1, 1, 0, 1, 0, 0,
      0, 0, 0],[0, 1, 1, 1, 1, 0, 0, 0, 1, 1, 1, 0, 0, 0, 0],[1, 1, 1, 1, 1, 0, 0, 0, 1, 1, 0,
      0, 0, 0],[1, 1, 1, 0, 0, 1, 1, 0, 1, 1, 1, 0, 1, 0, 1]]
    diagonal_jumps = 5
    # Initialize a dictionary to store the cost of reaching each visited state
    visited_costs = {}
    visited_costs[(initial_state, diagonal_jumps)] = 0
    # Initialize a priority queue of states not yet visited, with the initial state as the first
      element. The priority of each element is the cost to reach that state (g) + the estimate
      remaining cost (h) to reach the goal
    # Record the actions required to get to each state in a list; no actions performed to reach the
      initial state
    queue = [(0, 0, [initial_state], (initial_state, diagonal_jumps))]
    return initial_state, goal_state, num_rows, num_cols, park_map, diagonal_jumps, visited_costs,
      queue
```

Table 25: GPT4 successfully generated a correct (but non-optimal) solution for this trampoline_matrix problem instance using MSMT A* approach. Prompting GPT4 with A* results in a feasible but incorrect solution and all other three baseline code and text-based prompting methods resulted in infeasible solutions.

CITY DIRECTED GRAPH

**city_directed_graph instance (diff_sorted_id: 5)**: We have a map of cities, each represented by a letter, and they are connected by one-way roads. The adjacency matrix below shows the connections between the cities. Each row and column represents a city, and a '1' signifies a direct road from the city of the row to the city of the column. The travel time between any two directly connected cities is the same. Currently, we are located in city 'H'. Our task is to visit city Z and city K excatly twice. Determine the quickest route that allows us to visit both these destination cities, ensuring that we stop at the two destinations twice on our path. The sequence in which we visit the destination cities is not important. However, apart from K and Z, we can only visit each city once on our path. Provide the solution as a list of the city names on our path, including the start, in Python syntax.\n\n K B Z S X H O E U V \nK 0 0 1 1 0 0 1 0 1 \nB 0 0 0 0 0 0 0 1 0 1 \nZ 0 1 0 1 0 1 1 0 0 1 \nS 0 1 1 0 1 0 0 0 0 0 \nX 1 1 0 0 0 0 0 0 0 1 \nH 0 0 0 1 0 0 1 0 1 0 \nO 0 0 1 1 1 0 0 0 0 0 \nE 1 0 0 0 0 0 1 0 0 1 \nU 0 0 0 0 0 0 1 0 0 0 \nV 0 0 0 0 1 0 0 0 0 0 \n

A* algorithm generated by GPT4 for the problem above using A* MSMT approach that passed the unit tests:

```python
import heapq

def initialize():
    # Define the initial state and the goal state of the puzzle, represented as 3x3 tuples
    initial_state = 'H'
    goal_state = ['K', 'K', 'Z', 'Z']
    # Encoding other variables given in the problem statement
    adjacency_matrix = [[0, 0, 1, 1, 0, 0, 1, 1, 0, 1], [0, 0, 0, 0, 0, 0, 0, 1, 0, 1], [0, 1, 0, 1,
        0, 1, 1, 0, 0, 1], [0, 1, 1, 0, 1, 0, 0, 0, 0, 0], [1, 1, 0, 0, 0, 0, 0, 0, 0, 1], [0, 0, 0, 1,
        0, 0, 1, 0, 1, 0], [0, 0, 1, 1, 1, 0, 0, 0, 0, 0], [1, 0, 0, 0, 0, 0, 1, 0, 0, 1], [0, 0, 0, 0,
        0, 0, 1, 0, 0, 0], [0, 0, 0, 0, 1, 0, 0, 0, 0, 0]]
    cities = ['K', 'B', 'Z', 'S', 'X', 'H', 'O', 'E', 'U', 'V']
    # Initialize a dictionary to store the cost of reaching each visited state
    visited_costs = {}
    visited_costs[(initial_state, tuple(goal_state))] = 0
    # Initialize a priority queue of states not yet visited, with the initial state as the first
    #   element. The priority of each element is the cost to reach that state (g) + the estimate
    #   remaining cost (h) to reach the goal
    # Record the actions required to get to each state in a list; no actions performed to reach the
    #    initial state
    queue = [(0, 0, [], (initial_state, tuple(goal_state)))]
    return initial_state, goal_state, adjacency_matrix, cities, visited_costs, queue

def a_star():
    # The initialize function initializes and returns the visited_costs dictionary and the priority
    #    queue and encodes all of the variables given in the problem (ie the initial and goal board
    #    and dimensions of the puzzle board)
    initial_state, goal_state, adjacency_matrix, cities, visited_costs, queue = initialize()
    # While there are un-visited states
    while queue:
        # Pop the state with the lowest sum of the cost so far and estimated cost to the goal from
        #   the queue
        _, g, actions, state = heapq.heappop(queue)
        current_city, remaining_cities = state
        # We can check if the current state is the goal state with a simple equality check, as the
        #   goal state is predefined
        if not remaining_cities:
            return actions
        # Generate all valid actions from the current state, which includes moving to any of the
        #   cities connected to the current city
        for i in range(len(cities)):
            if adjacency_matrix[cities.index(current_city)][i] == 1:
                next_city = cities[i]
                # Check if the next city is a destination city and we have not visited it twice
        #   already
                if next_city in remaining_cities:
                    new_remaining_cities = list(remaining_cities)
                    new_remaining_cities.remove(next_city)
                    new_state = (next_city, tuple(new_remaining_cities))
                    new_cost = g + 1
                    if new_state not in visited_costs or new_cost < visited_costs[new_state]:
                        visited_costs[new_state] = new_cost
                        heapq.heappush(queue, (new_cost + heuristic(new_state, goal_state),
        #   new_cost, actions + [next_city], new_state))
```

51

The GPT4's implementation of the A* algorithm continues on the next page.

**Continuation of GPT-4's implementation:**

```
                # If the next city is not a destination city and we have not visited it before,
        we can move to it
                elif next_city not in actions:
                    new_state = (next_city, remaining_cities)
                    new_cost = g + 1
                    if new_state not in visited_costs or new_cost < visited_costs[new_state]:
                        visited_costs[new_state] = new_cost
                        heapq.heappush(queue, (new_cost + heuristic(new_state, goal_state),
     new_cost, actions + [next_city], new_state))
    return None

def heuristic(state, goal_state):
    # An admissible and consistent heuristic is the number of remaining cities to visit
    # The heuristic relaxes the constraint that we can only move to cities connected to the
      current city and presumes we can move to any of the remaining cities directly from the
      current city
    # Thus the heuristic reports a lower estimate on the cost to reach goal state and is
      admissible
    # The heuristic is consistent because the cost of moving to a city is always 1, which is
      exactly the decrease in the number of remaining cities to visit, if the city is a
      destination city, otherwise the estimated cost of the successor node is the same or higher
      , and he heuristic estimate for the goal state is 0, as there are no remaining cities to
      visit in the goal state.
    return len(state[1])

print(a_star())
```

Table 26: The A* algorithm was generated by GPT4 using the MSMT A* approach. GPT4's fifth implementation of the A* algorithm for city_directed_graph problem type passed the unit tests.

**city_directed_graph instance (diff_sorted_id: 84)**: We have a map of cities, each represented by a letter, and they are connected by one-way roads. The adjacency matrix below shows the connections between the cities. Each row and column represents a city, and a '1' signifies a direct road from the city of the row to the city of the column. The travel time between any two directly connected cities is the same. Currently, we are located in city 'T'. Our task is to visit city H and city K excatly twice. Determine the quickest route that allows us to visit both these destination cities, ensuring that we stop at the two destinations twice on our path. The sequence in which we visit the destination cities is not important. However, apart from K and H, we can only visit each city once on our path. Provide the solution as a list of the city names on our path, including the start, in Python syntax.\n\n F O L P Y R Q B T V A K S H \nF 0 0 0 1 1 0 0 1 0 0 0 0 0 0 \nO 0 0 1 0 1 0 1 0 1 0 0 1 0 0 \nL 1 0 0 0 0 1 1 0 0 0 1 1 1 0 \nP 0 0 1 0 1 0 0 1 0 1 1 0 0 0 \nY 0 0 0 0 0 0 0 1 0 0 1 0 1 0 \nR 0 0 0 0 1 0 1 1 0 0 0 0 1 0 \nQ 0 0 0 1 0 0 0 0 0 1 0 1 \nB 0 1 1 0 0 0 1 0 0 1 0 0 1 1 \nT 1 0 0 0 0 0 1 0 0 1 0 0 0 0 \nV 0 0 0 0 0 0 0 0 1 0 0 1 1 1 \nA 1 1 0 0 0 0 1 1 1 0 0 0 1 \nK 1 0 0 1 1 0 0 0 1 0 1 0 0 0 \nS 0 1 0 0 0 0 1 0 1 1 1 1 0 0 \nH 1 0 0 1 0 1 0 0 0 0 0 0 1 0 \n

'initialize' function generated by GPT4 for the problem above in the second stage of the MSMT approach

```python
def initialize():
    # Define the initial state and the goal state of the puzzle
    initial_state = ('T', ('K', 'K', 'H', 'H'))
    goal_state = []
    # Encoding other variables given in the problem statement
    adjacency_matrix = [[0, 0, 0, 1, 1, 0, 0, 1, 0, 0, 0, 0, 0, 0], [0, 0, 1, 0, 1, 0, 1, 0, 1, 0,
        0, 1, 0, 0], [1, 0, 0, 0, 0, 1, 1, 0, 0, 0, 1, 1, 1, 0], [0, 0, 1, 0, 1, 0, 0, 1, 0, 1, 1, 0,
        0, 0], [0, 0, 0, 0, 0, 0, 0, 1, 0, 0, 1, 1, 0, 0, 0, 1,
        0], [0, 0, 0, 1, 0, 0, 0, 0, 0, 0, 0, 1, 0, 1], [0, 1, 1, 0, 0, 0, 1, 0, 0, 1, 0, 0, 1, 1],
        [1, 0, 0, 0, 0, 0, 1, 0, 0, 0, 0, 0, 0], [0, 0, 0, 0, 0, 0, 0, 0, 1, 0, 0, 1, 1, 1], [1,
        1, 0, 0, 0, 0, 1, 1, 1, 0, 0, 0, 1], [1, 0, 0, 1, 1, 0, 0, 0, 1, 0, 1, 0, 0, 0], [0, 1, 0,
        0, 0, 0, 1, 0, 1, 1, 1, 1, 0, 0], [1, 0, 0, 1, 0, 1, 0, 0, 0, 0, 0, 0, 1, 0]]
    cities = ['F', 'O', 'L', 'P', 'Y', 'R', 'Q', 'B', 'T', 'V', 'A', 'K', 'S', 'H']
    # Initialize a dictionary to store the cost of reaching each visited state
    visited_costs = {}
    visited_costs[initial_state] = 0
    # Initialize a priority queue of states not yet visited, with the initial state as the first
      element. The priority of each element is the cost to reach that state (g) + the estimate
      remaining cost (h) to reach the goal
    # Record the actions required to get to each state in a list; no actions performed to reach the
      initial state
    queue = [(0, 0, [initial_state[0]], initial_state)]
    return initial_state, goal_state, adjacency_matrix, cities, visited_costs, queue
```

Table 27: GPT4 successfully generated an optimal solution for this city_directed_graph problem instance using MSMT A* approach. Prompting GPT4 with all other four code and text-based methods resulted in infeasible solutions.

# H PROMPTS

In this section, we provide the complete in-context examples and instructions given to the LLMs in each of the five prompting methods used in our experiments. Additionally, we present GPT-4's generated response for a pathfinding problem using each of these five prompts.

0_SHOT TEXT

| |
|---|
| **0_shot text prompt**: Solve the problem below step-by-step, using only text. The solution should be presented right after the phrase /answer_list/. Ensure there is no text following the reported solution. The solution should be formatted as a Python list, and each element in the list should be of the data type specified in the problem. |
| **problem statement (city_directed_graph instance 8)**: We have a map of cities, each represented by a letter, and they are connected by one-way roads. The adjacency matrix below shows the connections between the cities. Each row and column represents a city, and a '1' signifies a direct road from the city of the row to the city of the column. The travel time between any two directly connected cities is the same. Currently, we are located in city 'U'. Our task is to visit city A and city X excatly twice. Determine the quickest route that allows us to visit both these destination cities, ensuring that we stop at the two destinations twice on our path. The sequence in which we visit the destination cities is not important. However, apart from X and A, we can only visit each city once on our path. Provide the solution as a list of the city names on our path, including the start, in Python syntax.\n\n Y O Q X K R A U M L \nY 0 1 0 0 0 1 1 1 1 1 \nO 0 0 0 0 0 0 1 0 0 0 \nQ 1 0 0 1 0 1 0 0 1 0 \nX 1 0 1 0 0 1 0 0 1 1 \nK 0 1 0 1 0 0 1 0 0 0 \nR 0 0 0 0 1 0 0 0 0 0 \nA 0 0 0 1 0 1 0 0 1 1 \nU 0 0 0 0 0 0 0 0 0 1 \nM 0 0 1 0 0 0 0 0 0 0 \nL 0 1 0 0 0 0 0 0 0 0 \n |
| **GPT4's generation**:
To solve this problem, we need to find the quickest route that allows us to visit both city A and city X twice, starting from city U. We can only visit each city once, except for city A and city X.
From the adjacency matrix, we can see that city U is only directly connected to city L. So, we start our journey from city U to city L.
From city L, we can only go to city O. So, our next stop is city O.
From city O, we can only go to city A. So, our next stop is city A. This is our first visit to city A.
From city A, we can go to either city X, city R, or city Y. But since we need to visit city X twice, we choose to go to city X next.  This is our first visit to city X.
From city X, we can go to either city Y, city Q, or city A. But since we need to visit city A twice, we choose to go to city A next.  This is our second visit to city A.
From city A, we can go to either city X, city R, or city Y. But since we need to visit city X twice, we choose to go to city X next.  This is our second visit to city X.
So, the quickest route that allows us to visit both city A and city X twice, starting from city U, is U -> L -> O -> A -> X -> A -> X.
/answer_list/ ['U', 'L', 'O', 'A', 'X', 'A', 'X'] |
| **LLM-solution extracted**: ['U', 'L', 'O', 'A', 'X', 'A', 'X']
**LLM-solution is not feasible** |

Table 28: GPT4's solution for a city_directed_graph problem using 0_shot text prompting method.

2538
2539

4_SHOT COT TEXT

2540
2541
2542
2543
2544
2545

**4_shot CoT text prompt**:The objective is to solve the problems below step-by-step, using only text. The solution should be presented right after the phrase /answer_list/. Ensure there is no text following the reported solution. The solution should be formatted as a Python list, and each element in the list should be of the data type specified in the problem.
Examples:

2546
2547
2548
2549
2550
2551
2552
2553

In the 8-puzzle game, you are given a grid with numbered square tiles arranged randomly and one tile missing. The goal is to arrange the tiles in descending order by sliding them into the empty space. The tiles can move in 4 directions: left, right, up, and down. Given the initial state of the puzzle below, where the empty spot is represented as "_", provide the shortest list of tiles that need to be swapped with the empty spot to achieve the goal state. The goal state is when all tiles are in descending order, with the largest number in the top left corner, and the empty spot is in the bottom right corner. The solution should be a list of numbers in Python format, where each number represents the number on the tile that the empty spot is swapped with at each turn. Initial state of the puzzle: [[55, 43, 17], [97, 35, 9], [12, 25, '_']]

2554
2555
2556
2557
2558
2559
2560
2561
2562
2563
2564
2565
2566

The goal state of the puzzle is [[97, 55, 43], [35, 25, 17], [12, 9, '_']].
To achieve this, we need to follow these steps:
1. Swap 9 with '_': [[55, 43, 17], [97, 35, '_'], [12, 25, 9]]
2. Swap 17 with '_': [[55, 43, '_'], [97, 35, 17], [12, 25, 9]]
3. Swap 43 with '_': [[55, '_', 43], [97, 35, 17], [12, 25, 9]]
4. Swap 55 with '_': [['_', 55, 43], [97, 35, 17], [12, 25, 9]]
5. Swap 97 with '_': [[97, 55, 43], ['_', 35, 17], [12, 25, 9]]
6. Swap 35 with '_': [[97, 55, 43], [35, '_', 17], [12, 25, 9]]
7. Swap 25 with '_': [[97, 55, 43], [35, 25, 17], [12, '_', 9]]
8. Swap 9 with '_': [[97, 55, 43], [35, 25, 17], [12, 9, '_']]
So, the list of tiles that need to be swapped with the empty spot to achieve the goal state is [9, 17, 43, 55, 97, 35, 25, 9].
The answer is /answer_list/[9, 17, 43, 55, 97, 35, 25, 9]

2567
2568
2569
2570
2571
2572
2573
2574
2575
2576
2577
2578
2579
2580
2581
2582
2583
2584

The game of 'Sort It' begins with 3 tubes, each filled with balls of different colors. The goal is to sort the balls by color, with each tube containing balls of only one color. Only one ball can be moved at a time, taken from the top of one tube and placed on top of another. All of the tubes have the same capacity of 6 balls. It is not allowed to place a ball in a tube that is already at capacity. The solution should be a list of tuples, each containing, first, the index of the tube from which a ball is taken and, second, the index of the tube to which it is moved, indexing from 0. Given the initial state of the tubes, represented by the lists below (with the leftmost item being the color of the topmost ball in each tube), what is the shortest list of move tuples that will result in all the balls being correctly sorted? [['Green', 'Red', 'Green', 'Red'], ['Blue', 'Blue', 'Red', 'Green'], ['Red', 'Blue', 'Green', 'Blue']]
The initial state of the tubes is:
Tube 0: ['Green', 'Red', 'Green', 'Red']
Tube 1: ['Blue', 'Blue', 'Red', 'Green']
Tube 2: ['Red', 'Blue', 'Green', 'Blue']
The goal is to sort the balls by color, with each tube containing balls of only one color. The moves are as follows:
1. Move 'Green' from Tube 0 to Tube 2: (0, 2) The state of the tubes now is:
Tube 0: ['Red', 'Green', 'Red'] Tube 1: ['Blue', 'Blue', 'Red', 'Green'] Tube 2: ['Green', 'Red', 'Blue', 'Green', 'Blue']
2. Move 'Red' from Tube 0 to Tube 1: (0, 1)
Tube 0: ['Green', 'Red'] Tube 1: ['Red', 'Blue', 'Blue', 'Red', 'Green'] Tube 2: ['Green', 'Red', 'Blue', 'Green', 'Blue']
3. Move 'Green' from Tube 0 to Tube 2: (0, 2) 55
Tube 0: ['Red'] Tube 1: ['Red', 'Blue', 'Blue', 'Red', 'Green'] Tube 2: ['Green', 'Green', 'Red', 'Blue', 'Green', 'Blue']

The 4_shot CoT text prompt continues on the next page.

**Continuation of the 4_shot CoT text:**

4. Move 'Red' from Tube 1 to Tube 0: (1, 0)

Tube 0: ['Red', 'Red'] Tube 1: ['Blue', 'Blue', 'Red', 'Green'] Tube 2: ['Green', 'Green', 'Red', 'Blue', 'Green', 'Blue']

5. Move 'Green' from Tube 2 to Tube 1: (2, 1)

Tube 0: ['Red', 'Red'] Tube 1: ['Green', 'Blue', 'Blue', 'Red', 'Green'] Tube 2: ['Green', 'Red', 'Blue', 'Green', 'Blue']

6. Move 'Green' from Tube 2 to Tube 1: (2, 1)

Tube 0: ['Red', 'Red'] Tube 1: ['Green', 'Green', 'Blue', 'Blue', 'Red', 'Green'] Tube 2: ['Red', 'Blue', 'Green', 'Blue']

7. Move 'Red' from Tube 2 to Tube 1: (2, 0)

Tube 0: ['Red', 'Red', 'Red'] Tube 1: ['Green', 'Green', 'Blue', 'Blue', 'Red', 'Green'] Tube 2: ['Blue', 'Green', 'Blue']

8. Move 'Green' from Tube 1 to Tube 0: (1, 0)

Tube 0: ['Green', 'Red', 'Red', 'Red'] Tube 1: ['Green', 'Blue', 'Blue', 'Red', 'Green'] Tube 2: ['Blue', 'Green', 'Blue']

9. Move 'Green' from Tube 1 to Tube 0: (1, 0)

Tube 0: ['Green', 'Green', 'Red', 'Red', 'Red'] Tube 1: ['Blue', 'Blue', 'Red', 'Green'] Tube 2: ['Blue', 'Green', 'Blue']

10. Move 'Blue' from Tube 2 to Tube 1: (2, 1)

Tube 0: ['Green', 'Green', 'Red', 'Red', 'Red'] Tube 1: ['Blue', 'Blue', 'Blue', 'Red', 'Green'] Tube 2: ['Green', 'Blue']

11. Move 'Green' from Tube 2 to Tube 0: (2, 0)

Tube 0: ['Green', 'Green', 'Green', 'Red', 'Red', 'Red'] Tube 1: ['Blue', 'Blue', 'Blue', 'Red', 'Green'] Tube 2: ['Blue']

12. Move 'Blue' from Tube 1 to Tube 2: (1, 2)

Tube 0: ['Green', 'Green', 'Green', 'Red', 'Red', 'Red'] Tube 1: ['Blue', 'Blue', 'Red', 'Green'] Tube 2: ['Blue', 'Blue']

13. Move 'Blue' from Tube 1 to Tube 2: (1, 2)

Tube 0: ['Green', 'Green', 'Green', 'Red', 'Red', 'Red'] Tube 1: ['Blue', 'Red', 'Green'] Tube 2: ['Blue', 'Blue', 'Blue',]

14. Move 'Blue' from Tube 1 to Tube 2: (1, 2)

Tube 0: ['Green', 'Green', 'Green', 'Red', 'Red', 'Red'] Tube 1: ['Red', 'Green'] Tube 2: ['Blue', 'Blue', 'Blue', 'Blue']

15. Move 'Red' from Tube 1 to Tube 2: (1, 2)

Tube 0: ['Green', 'Green', 'Green', 'Red', 'Red', 'Red'] Tube 1: ['Green'] Tube 2: ['Red', 'Blue', 'Blue', 'Blue', 'Blue']

16. Move 'Green' from Tube 0 to Tube 1: (0, 1)

Tube 0: ['Green', 'Green', 'Red', 'Red', 'Red'] Tube 1: ['Green', 'Green'] Tube 2: ['Red', 'Blue', 'Blue', 'Blue', 'Blue']

17. Move 'Green' from Tube 0 to Tube 1: (0, 1)

Tube 0: ['Green', 'Red', 'Red', 'Red'] Tube 1: ['Green', 'Green', 'Green'] Tube 2: ['Red', 'Blue', 'Blue', 'Blue', 'Blue']

18. Move 'Green' from Tube 0 to Tube 1: (0, 1)

Tube 0: ['Red', 'Red', 'Red'] Tube 1: ['Green', 'Green', 'Green', 'Green'] Tube 2: ['Red', 'Blue', 'Blue', 'Blue', 'Blue']

19. Move 'Red' from Tube 2 to Tube 0: (2, 0)

Tube 0: ['Red', 'Red', 'Red', 'Red'] Tube 1: ['Green', 'Green', 'Green', 'Green'] Tube 2: ['Blue', 'Blue', 'Blue', 'Blue']

The final state of the tubes is:

Tube 0: ['Red', 'Red', 'Red', 'Red']

Tube 1: ['Green', 'Green', 'Green', 'Green']

Tube 2: ['Blue', 'Blue', 'Blue', 'Blue']

So, the shortest list of move tuples that will result in all the balls being correctly sorted is [(0, 1), (0, 2), (1, 0), (2, 1), (2, 1), (2, 0), (1, 0), (1, 0), (2, 1), (2, 0), (1, 2), (1, 2), (1, 2), (1, 2), (0, 1), (0, 1), (0, 1), (2, 0)].

The answer is /answer_list/[(0, 1), (0, 2), (1, 0), (2, 1), (2, 1), (2, 0), (1, 0), (1, 0), (2, 1), (2, 0), (1, 2), (1, 2), (1, 2), (0, 1), (0, 1), (0, 1), (2, 0)]

The 4_shot CoT text prompt continues on the next page.

**Continuation of the 4_shot CoT text:**

———

Given 6 labeled water jugs with capacities 37, 133, 38, 72, 41, 23, 122 liters, we aim to fill 3 unlabeled buckets, numbered 1 to 3 and arranged in a line in ascending order, with 195, 224, 268 liters of water respectively. The amount of water in each unlabeled bucket can not at any point in time exceed the amount of water in the bucket placed before it. Jugs can only be filled to the top and emptied completely, and the unlabeled buckets cannot be overfilled. An action, represented as a tuple ('+', X, Y) or ('-', X, Y), involves adding to or removing water from the unlabeled bucket numbered Y, using the jug with capacity X. Determine the shortest sequence of actions needed to fill the buckets as specified, and present the solution as a list of action tuples in Python syntax.

The problem can be solved using a greedy algorithm. The idea is to start filling the last bucket that has not been filled yet in order to not violate the constraint on the relative amount of water in the buckets based on their order. Here are the steps to solve the problem:
1. Start with the last bucket (bucket 3) and find the largest jug that can be used to fill it without overfilling it or making it difficult to fill the remaining capacity using the given jugs. In this case, the largest jug that can be used is the 122-liter jug. So, the first action is ('+', 122, 3).
2. Repeat the process for the same bucket (bucket 3) until it is filled with 268 liters of water. Each time, find the largest jug that can be used without overfilling the bucket. The actions are ('+', 72, 3), ('+', 37, 3), ('+', 37, 3).
3. Move to the next bucket (bucket 2) and repeat the process until the bucket is filled with 224 liters of water. The actions are ('+', 122, 2),('+', 41, 2),('+', 38, 2), ('+', 23, 2).
4. Move to the last bucket (bucket 1) and repeat the process until the bucket is filled with 195 liters of water. The greedy actions are ('+', 122, 1), ('+', 72, 1).
5. However, the last bucket (bucket 1) is filled with 194 liters, 1 liter less than required. Since there is no 1 liter jug, the 38 liter jug can be used to add water, then 37 liters is removed using the 37 liter jug, adding a net of 1 liter to the bucket. We can not use the 38 liter jug to add water after filling the bucket with 194 liters of water because it will overflow. Thus the order of actions to fill the last bucket is: ('+', 122, 1), ('+', 72, 1), ('-', 37, 1), ('+', 38, 1).
The final sequence of actions is [('+', 122, 3), ('+', 72, 3), ('+', 37, 3), ('+', 37, 3), ('+', 122, 2),('+', 41, 2),('+', 38, 2), ('+', 23, 2), ('+', 122, 1), ('+', 38, 1), ('-', 37, 1), ('+', 72, 1)].
The answer is /answer_list/[('+', 122, 3), ('+', 72, 3), ('+', 37, 3), ('+', 37, 3), ('+', 122, 2),('+', 41, 2),('+', 38, 2), ('+', 23, 2), ('+', 122, 1), ('+', 72, 1), ('-', 37, 1), ('+', 38, 1)]

———

"In the magic square problem, a 4x4 grid is filled with unique integers ranging from 29 to 54. Some numbers are already given, while others are unknown and represented as 'x'. The sums of columns 1 to 2 (counting from 0) must be 148, 196 respectively, and the sums of rows 1 to 2 must be 187, 149 respectively. Also, the sum of the numbers in the diagonal from the top left to the bottom right corner of the grid should equal 166. The goal is to find unique integers in the given range to replace with 'x's in the grid below such that the sum of the specified rows, columns, and diagonal equals the given amounts and the sum of all of the numbers in the grid is as low as possible. The solution should be provided as a list of tuples in Python syntax. Each tuple should contain three numbers: the row index, the column index (both starting from 0), and the value of the unknown number at that position.\n\nGrid:\n [['47' 'x' 'x' '32']\n ['x' 'x' 'x' '49']\n ['x' '31' '50' 'x']\n ['x' 'x' '52' '30']]

Step 1: The sum of the diagonal is given with only 1 element missing (in index (1, 1)). The sum of the diagonal is 166 and the known values in the diagonal are 47, 50, and 30. So, the missing value is 166 - 47 - 50 - 30 = 39. So, we fill the position (1, 1) with 39.
Grid after Step 1:
[['47' 'x' 'x' '32']
['x' '39' 'x' '49']
['x' '31' '50' 'x']
['x' 'x' '52' '30']]
Step 2: The sum of the first row is 187 and the known values in the first row are 39 and 49. So, the sum of the missing values is 187 - 49 - 39 = 99. We fill the missing values with the unique integers that sum to 99, which are 46 and 53. So, we fill the positions (1, 0) and (1, 2) with 46 and 53 respectively.
Grid after Step 2:
[['47' 'x' 'x' '32']
['46' '39' '53' '49']
['x' '31' '50' 'x']
['x' 'x' '52' '30']]

**Continuation of the 4_shot CoT text:**
Step 3: The sum of the second column is given with only 1 element missing (in index (0, 2)). The sum of the second column is 196 and the known values in the diagonal are 53, 50, and 52. So, the missing value is 196 - 53 - 50 - 52 = 41. So, we fill the position (0, 2) with 41.
Grid after Step 3:
[['47' 'x' '41' '32']
['46' '39' '53' '49']
['x' '31' '50' 'x']
['x' 'x' '52' '30']]
Step 4: The sum of the second row is 149 and the known values in the second row are 31 and 50. So, the sum of the missing values is 149 - 31 - 50 = 68. We fill the missing values with the unique integers that sum to 68, which are 33 and 35. So, we fill the positions (2, 0) and (2, 3) with 33 and 35 respectively.
Grid after Step 4:
[['47' 'x' '41' '32']
['46' '39' '53' '49']
['33' '31' '50' '35']
['x' 'x' '52' '30']]
Step 5: The sum of the first column is 148 and the known values in the first column are 39 and 31. So, the sum of the missing values is 148 - 31 - 39 = 78. We fill the missing values with the unique integers that sum to 78, which are 34 and 44. So, we fill the positions (0, 1) and (3, 1) with 34 and 44 respectively.
Grid after Step 5:
[['47' '34' '41' '32']
['46' '39' '53' '49']
['33' '31' '50' '35']
['x' '44' '52' '30']]
Step 6: There is 1 unknown number left in the grid at position (3,0). We must fill this cell with the smallest unique integer which is 29.
Grid after Step 6:
[['47' '34' '41' '32']
['46' '39' '53' '49']
['33' '31' '50' '35']
['29' '44' '52' '30']]
So, the solution is [(0, 1, 34), (0, 2, 41), (1, 0, 46), (1, 1, 39), (1, 2, 53), (2, 0, 33), (2, 3, 35), (3, 0, 29), (3, 1, 44)].
The answer is /answer_list/[(0, 1, 34), (0, 2, 41), (1, 0, 46), (1, 1, 39), (1, 2, 53), (2, 0, 33), (2, 3, 35), (3, 0, 29), (3, 1, 44)]

———
Solve the problem below in the same format, step-by-step, using only text.

**problem statement (city_directed_graph instance 8)**: We have a map of cities, each represented by a letter, and they are connected by one-way roads. The adjacency matrix below shows the connections between the cities. Each row and column represents a city, and a '1' signifies a direct road from the city of the row to the city of the column. The travel time between any two directly connected cities is the same. Currently, we are located in city 'U'. Our task is to visit city A and city X excatly twice. Determine the quickest route that allows us to visit both these destination cities, ensuring that we stop at the two destinations twice on our path. The sequence in which we visit the destination cities is not important. However, apart from X and A, we can only visit each city once on our path. Provide the solution as a list of the city names on our path, including the start, in Python syntax.\n\n Y O Q X K R A U M L \nY 0 1 0 0 0 1 1 1 1 1 \nO 0 0 0 0 0 0 1 0 0 0 \nQ 1 0 0 1 0 1 0 0 1 0 \nX 1 0 1 0 0 1 0 0 1 1 \nK 0 1 0 1 0 0 1 0 0 0 \nR 0 0 0 0 1 0 0 0 0 0 \nA 0 0 0 1 0 1 0 0 1 1 \nU 0 0 0 0 0 0 0 0 0 1 \nM 0 0 1 0 0 0 0 0 0 0 \nL 0 1 0 0 0 0 0 0 0 0 \n

**GPT4's generation**:
```
The adjacency matrix represents a directed graph where each city is a node
and each road is a directed edge.  The task is to find a path that starts
at city 'U', visits cities 'A' and 'X' exactly twice, and visits all other
cities at most once.
We can solve this problem using depth-first search (DFS) with backtracking.
DFS is a graph traversal algorithm that explores as far as possible along
each branch before backtracking.
Here are the steps to solve the problem:
1.  Start at city 'U'.
2.  From city 'U', the only city we can go to is city 'L'. So, we go to city
'L'.
3.  From city 'L', we can go to city 'O'. So, we go to city 'O'.
4.  From city 'O', we can go to city 'A'. So, we go to city 'A'. This is the
first visit to city 'A'.
```

GPT4's generation for this problem using 4_shot CoT text prompting continues on the next page.

**Continuation of GPT4's generation:**
5. From city 'A', we can go to cities 'X', 'R', 'M', and 'L'. But
we have already visited city 'L', and we want to visit city 'X' last,
so we go to city 'R'.
6. From city 'R', we can go to city 'K'. So, we go to city 'K'.
7. From city 'K', we can go to cities 'O', 'X', and 'A'. But we
have already visited city 'O', and we want to visit city 'X' last,
so we go to city 'A'. This is the second visit to city 'A'.
8. From city 'A', we can go to cities 'X', 'R', 'M', and 'L'. But
we have already visited cities 'R', 'M', and 'L', so we go to city
'X'. This is the first visit to city 'X'.
9. From city 'X', we can go to cities 'Y', 'Q', 'R', 'M', and 'A'.
But we have already visited cities 'R', 'M', and 'A', so we go to
city 'Y'.
10. From city 'Y', we can go to cities 'O', 'R', 'A', 'U', 'M', and
'L'. But we have already visited cities 'O', 'R', 'A', 'U', 'M', and
'L', so we go to city 'Q'.
11. From city 'Q', we can go to cities 'Y', 'X', 'R', 'M', and 'L'.
But we have already visited cities 'Y', 'R', 'M', and 'L', so we go
to city 'X'. This is the second visit to city 'X'.
So, the quickest route that allows us to visit both cities 'A' and
'X' twice, ensuring that we stop at the two destinations twice on our
path, is ['U', 'L', 'O', 'A', 'R', 'K', 'A', 'X', 'Y', 'Q', 'X'].
The answer is /answer_list/['U', 'L', 'O', 'A', 'R', 'K', 'A', 'X', 'Y', 'Q', 'X']
**LLM-solution extracted**: ['U', 'L', 'O', 'A', 'R', 'K', 'A', 'X', 'Y', 'Q', 'X']
**LLM-solution is not feasible**

Table 29: GPT4's solution for a city_directed_graph problem using 4_shot CoT text prompting method. As shown above, in the CoT part of in-context examples, we use ASCII characters to represent the intermediate states of the problem after each action. Also as city_directed_graph is a pathfinding problem, the 4_shot CoT text prompt is constructed using one solved instance from each of the four other categories in SearchBench.

0_SHOT CODE

---

**0_shot code prompt**: Write a python script that solves the problem below. Explain the steps taken in comments:

**problem statement (city_directed_graph instance 8)**: We have a map of cities, each represented by a letter, and they are connected by one-way roads. The adjacency matrix below shows the connections between the cities. Each row and column represents a city, and a '1' signifies a direct road from the city of the row to the city of the column. The travel time between any two directly connected cities is the same. Currently, we are located in city 'U'. Our task is to visit city A and city X excatly twice. Determine the quickest route that allows us to visit both these destination cities, ensuring that we stop at the two destinations twice on our path. The sequence in which we visit the destination cities is not important. However, apart from X and A, we can only visit each city once on our path. Provide the solution as a list of the city names on our path, including the start, in Python syntax.\n\n Y O Q X K R A U M L \nY 0 1 0 0 0 1 1 1 1 1 \nO 0 0 0 0 0 0 1 0 0 0 \nQ 1 0 0 1 0 1 0 0 1 0 \nX 1 0 1 0 0 1 0 0 1 1 \nK 0 1 0 1 0 0 1 0 0 0 \nR 0 0 0 0 1 0 0 0 0 0 \nA 0 0 0 1 0 1 0 0 1 1 \nU 0 0 0 0 0 0 0 0 0 1 \nM 0 0 1 0 0 0 0 0 0 0 \nL 0 1 0 0 0 0 0 0 0 0 \n

**GPT4's generation**: This problem can be solved using a depth-first search (DFS) algorithm. The DFS algorithm is an algorithm for traversing or searching tree or graph data structures. The algorithm starts at the root (selecting some arbitrary node as the root in the case of a graph) and explores as far as possible along each branch before backtracking.
Here is a Python script that solves the problem:

```python
# Define the adjacency matrix
adjacency_matrix = {
    'Y': ['O', 'R', 'A', 'U', 'M', 'L'],
    'O': ['A'],
    'Q': ['Y', 'X', 'R', 'M'],
    'X': ['Y', 'Q', 'R', 'M', 'L'],
    'K': ['O', 'X', 'A'],
    'R': ['K'],
    'A': ['X', 'R', 'M', 'L'],
    'U': ['L'],
    'M': ['Q'],
    'L': ['O']
}

# Define the start city and the destination cities
start_city = 'U'
destination_cities = ['A', 'X']

# Define a function to find all paths
def find_all_paths(start, end, path=[]):
    path = path + [start]
    if start == end:
        return [path]
    if start not in adjacency_matrix:
        return []
    paths = []
    for node in adjacency_matrix[start]:
        if node not in path:
            newpaths = find_all_paths(node, end, path)
            for newpath in newpaths:
                paths.append(newpath)
    return paths

# Find all paths from the start city to the destination cities
paths = []
for city in destination_cities:
    paths.extend(find_all_paths(start_city, city))

# Filter the paths to only include those that visit both destination cities twice
valid_paths = [path for path in paths if path.count('A') == 2 and path.count('X') == 2]

# Print the shortest valid path
print(min(valid_paths, key=len))
```

**Continuation of GPT4's generation:**

This script first defines the adjacency matrix as a dictionary, where
each key is a city and each value is a list of cities that can be reached
directly from that city.  It then defines a function to find all paths
from a start city to an end city.  This function uses a depth-first search
algorithm to explore all possible paths.  The script then finds all paths
from the start city to each of the destination cities, and filters these
paths to only include those that visit both destination cities twice.
Finally, it prints the shortest valid path.

**LLM-code run result**: LLM program returned the following error: Traceback (most recent call last):
File "./city_directed_graph_results/gpt_4_0shot_full_code/gpt_4_0shot_code_python_scripts/problem_8.py",
line 44, in <module>
print(min(valid_paths, key=len)) ValueError: min() arg is an empty sequence
**Solution is not feasible**

Table 30: GPT4's solution for a city_directed_graph problem using 0_shot text prompting method.

4_SHOT A*

---

**4_shot A\* prompt**: The goal is to solve the problems given by implementing the A\* search algorithm in python. Examples:

In the 8-puzzle game, you are given a grid with numbered square tiles arranged randomly and one tile missing. The goal is to arrange the tiles in descending order by sliding them into the empty space. The tiles can move in 4 directions: left, right, up, and down. Given the initial state of the puzzle below, where the empty spot is represented as "_", provide the shortest list of tiles that need to be swapped with the empty spot to achieve the goal state. The goal state is when all tiles are in descending order, with the largest number in the top left corner, and the empty spot is in the bottom right corner. The solution should be a list of numbers in Python format, where each number represents the number on the tile that the empty spot is swapped with at each turn. Initial state of the puzzle: [[55, 43, 17], [97, 35, 9], [12, 25, '_']]
'''python

```python
import heapq

def a_star():
    # Define the initial state and the goal state of the puzzle, represented as 3x3 tuples
    initial_state = ((55, 43, 17), (97, 35, 9), (12, 25, '_'))
    goal_state = ((97, 55, 43), (35, 25, 17), (12, 9, '_'))
    # Encoding other variables given in the problem statement
    num_rows = 3
    num_cols = 3
    # Initialize a dictionary to store the cost of reaching each visited state
    visited_costs = {}
    visited_costs[initial_state] = 0
    # Initialize a priority queue of states not yet visited, with the initial state as the first
      element. The priority of each element is the cost to reach that state (g) + the estimate
      remaining cost (h) to reach the goal
    # Record the swaps required to get to each state in a list; no swaps performed to reach the
      initial state
    queue = [(0, 0, [], initial_state)]
    # While there are un-visited states
    while queue:
        # Pop the state with the lowest sum of the cost so far and estimated cost to the goal from
      the queue
        _, g, actions, state = heapq.heappop(queue)
        # We can check if the current state is the goal state with a simple equality check, as the
      goal state is predefined
        if state == goal_state:
            return actions
        # Generate all valid actions from the current state, which includes swapping any of the
      tiles neighboring the empty spot, with the empty spot
        # Generate the coordinates of the tiles neighboring "_"
        empty_row, empty_col = [(i, j) for i in range(num_rows) for j in range(num_cols) if state[i
      ][j] == '_'][0]
        for d_row, d_col in [(0, -1), (0, 1), (1, 0), (-1, 0)]:
            swap_row, swap_col = empty_row + d_row, empty_col + d_col
            # Check if the swap is valid, ie if the coordinate of the tile to be swapped is a valid
       coordinate within the bounds of the board
            if 0 <= swap_row < num_rows and 0 <= swap_col < num_cols:
                # The actions is valid, generate the new state
                new_state = [list(row[:]) for row in state]
                number_to_be_swapped = new_state[swap_row][swap_col]
                # Do the swap
                new_state[empty_row][empty_col], new_state[swap_row][swap_col] = new_state[swap_row
      ][swap_col], new_state[empty_row][empty_col]
                new_state = tuple(tuple(row) for row in new_state)
                # The cost so far is the number of swaps made, as our objective is to minimize the
      number of swaps required to reach the goal state
                new_cost = g + 1
                # If the new state is unvisited or we found a new path with a lower cost to reach
      this state, add it to the queue of not-yet-visited states
                if new_state not in visited_costs or new_cost < visited_costs[new_state]:
                    visited_costs[new_state] = new_cost
                    heapq.heappush(queue, (new_cost + heuristic(new_state, goal_state), new_cost,
      actions + [number_to_be_swapped], new_state))
    return None
```

---

The A\* prompt continues on the next page.

**Continuation of A\* prompt:**

```python
def heuristic(state, goal):
    # An admissible and consistent heuristic is the sum of the Manhattan distances (the
      shortest path) of each tile from its goal position
    # The heuristic relaxes the constraint that a tile can only be swapped with the empty spot
      and presumes we can move the tiles to their goal position by swapping them with any of the
      other tiles
    # Thus the heuristic reports a lower estimate on the cost to reach goal state and is
      admissible
    # The heuristic is consistent because the cost of moving a tile to a neighboring coordinate
      is always 1, which is exactly the decrease in the Manhattan distance, if the tile is
      moved toward its goal position, otherwise the estimated cost of the successor node is the
      same or higher, and he heuristic estimate for the goal state is 0, as the distance of each
      tile from its goal position would be 0 in the goal state.
    h = 0
    for i in range(len(state)):
        for j in range(len(state[i])):
            # Can't compare integers with "_" when finding the goal position of each tile, thus
          ignore the "_" tile
            if state[i][j] != '_':
                # Get goal position of each tile
                goal_row, goal_col = [(x, y) for x in range(len(goal)) for y in range(len(goal[
          x])) if goal[x][y] == state[i][j]][0]
                # Add the the Manhattan distance of the current and goal coordinates of the
          tile to the estimate
                h += abs(i - goal_row) + abs(j - goal_col)
    return h

print(a_star())
```
----

The game of 'Sort It' begins with 3 tubes, each filled with 4 balls of different colors. The goal is to sort the balls by color, with each tube containing balls of only one color. Only one ball can be moved at a time, taken from the top of one tube and placed on top of another. The capacity of each tube (maximum number of balls we can fit in each tube) is 6 balls. It is not allowed to place a ball in a tube that already has 6 balls. The solution should be a list of tuples, each containing, first, the index of the tube from which a ball is taken and, second, the index of the tube to which it is moved, indexing from 0. Given the initial state of the tubes, represented by the lists below (with the leftmost item being the color of the topmost ball in each tube), what is the shortest list of move tuples that will result in all the balls being correctly sorted? [['Green', 'Red', 'Green', 'Red'], ['Blue', 'Blue', 'Red', 'Green'], ['Red', 'Blue', 'Green', 'Blue']]

```python
import heapq
from collections import Counter

def a_star():
    # Define the initial state of the tubes, as a 2d tuple of color of the balls in tubes 0 to
      2
    initial_state = (('Green', 'Red', 'Green', 'Red'), ('Blue', 'Blue', 'Red', 'Green'), ('Red'
      , 'Blue', 'Green', 'Blue'))
    # Encoding other variables given in the problem statement
    num_tubes = 3
    capacity = 6
    # Initialize a dictionary to store the cost of reaching each visited state
    visited_costs = {}
    visited_costs[initial_state] = 0
    # Initialize a priority queue of states not yet visited, with the initial state as the
      first element. The priority of each element is the cost to reach that state (g) + the
      estimate remaining cost (h) to reach the goal
    # Record the actions required to get to each state in a list; no actions performed to reach
      the initial state
    queue = [(0, 0, [], initial_state)]
    # While there are un-visited states
    while queue:
        # Pop the state with the lowest sum of the cost so far and estimated cost to the goal
      from the queue
        _, g, actions, state = heapq.heappop(queue)
```

The A\* prompt continues on the next page.

**Continuation of A\* prompt:**

```python
        # Check of the current state is the goal state
        # The goal state is where each tube only contains balls of 1 single color
        if all(len(set(tube)) <= 1 for tube in state):
            return actions
        # Generate all possible actions from the current state, which includes moving a ball
    from any of the 3 tubes to another tube
        for from_tube_ind in range(num_tubes):
            for to_tube_ind in range(num_tubes):
                # Check if the new state would be valid, ie from_tube and to_tube must not be
    the same tube
                # And from_tube must at least have 1 ball to move and the to_tube cannot be at
    capacity
                if from_tube_ind != to_tube_ind and state[from_tube_ind] and len(state[
    to_tube_ind]) < capacity:
                    # Generate the new state
                    new_state = [list(tube[:]) for tube in state]
                    # The ball to move is the topmost ball in the from_tube, at index 0
                    ball_to_move = new_state[from_tube_ind].pop(0)
                    # Add the ball to the top of the to_tube
                    new_state[to_tube_ind].insert(0, ball_to_move)
                    new_state = tuple(tuple(tube) for tube in new_state)
                    # The cost so far is the number of moves made, as the task is to minimize
    the number of moves required
                    new_cost = g + 1
                    # If the new state is unvisited or we found a new path with a lower cost to
     arrive at this state, add it to the queue of un-visited states
                    if new_state not in visited_costs or new_cost < visited_costs[new_state]:
                        visited_costs[new_state] = new_cost
                        heapq.heappush(queue, (new_cost + heuristic(new_state), g + 1, actions
     + [(from_tube_ind, to_tube_ind)], new_state))
        return None

def heuristic(tubes):
    # An admissible and consistent heuristic for this problem is the count of balls that are
     not the same color as the most frequent color in their tube
    # This heuristic relaxes the constraint that only the ball at the top of the tube can be
     moved
    # It is admissible because it never overestimates the cost to reach the goal, as each
     mismatched ball must be moved at least once
    # It's consistent because moving a ball from one tube to another reduces the heuristic cost
      of the successor node by a max of 1 (if the moved ball's color matches the most common
     color in the new tube but not in the old one), which is equal to the cost of reaching the
     successor node
    # Thus h(s) is always less than or equal to c(s, n) (equal to 1) + h(n)
    h = 0
    for tube in tubes:
        if tube:
            # If there are ties in the frequency of colors, the most_commonm_color must be
     match the color of the balls lower that are in the tube, as moving lower balls is costlier
            reversed_tube = tube[:]
            reversed_tube = reversed_tube[::-1]
            # Get the most common color
            most_common_color = Counter(reversed_tube).most_common(1)[0][0]
            for ball in tube:
                if ball != most_common_color:
                    h += 1
    return h

print(a_star())
```
"""

______

Given 6 labeled water jugs with capacities 37, 133, 38, 72, 41, 23, 122 liters, we aim to fill 3 unlabeled buckets, numbered 1 to 3 and arranged in a line in ascending order, with 195, 224, 268 liters of water respectively. The amount of water in each unlabeled bucket can not at any point in time exceed the amount of water in the bucket placed before it. Jugs can only be filled to the top and emptied completely, and the unlabeled buckets cannot be overfilled. An action, represented as a tuple ('+', X, Y) or ('-', X, Y), involves adding to or removing water from the unlabeled bucket numbered Y, using the jug with capacity X. Determine the shortest sequence of actions needed to fill the buckets as specified, and present the solution as a list of action tuples in Python syntax.

The A\* prompt continues on the next page.

**Continuation of A\* prompt:** ```python

```python
from heapq import heappush, heappop

def a_star():
    # Define the capacities of the jugs, the goal state, and initial state, with states having
    an immutable data type
    jugs = [37, 133, 38, 72, 41, 23, 122]
    goal_state = (195, 224, 268)
    initial_state = (0, 0, 0)
    num_buckets = 3
    visited_costs = {}
    visited_costs[initial_state] = 0
    queue = []
    # No actions taken in the initial state
    heappush(queue, (0, 0, [], initial_state))
    while queue:
        _, g, actions, state = heappop(queue)
        # If the amount of water in the buckets in the current state equal the goal amounts,
    return the actions taken
        if state == goal_state:
            return actions
        # Generate all possible actions from the current state, which includes adding or
    subtracting water using any of the 6 jugs to any of the 3 buckets
        # Iterating through capacities of jugs and indices of buckets as the action tuples must
     include the operation ('+' or '-'), capacity of the jug used, and the index of the bucket
     affected
        for jug in jugs:
            for bucket_ind in range(num_buckets):
                # Check if adding water using the current jug results in a valid state, ie the
    addition must not result in overflowing any of the buckets
                if (state[bucket_ind] + jug <= goal_state[bucket_ind]):
                    temp_state = list(state)[:]
                    temp_state[bucket_ind] += jug
                    # And the new state must maintain the constraint on the relative amount of
    water in the buckets based on their order
                    if all(temp_state[i] <= temp_state[i + 1] for i in range(len(temp_state) -
    1)):
                        # Generate the new state
                        new_state = tuple(temp_state)
                        # The cost so far is the number of actions taken, as the task is to
    minimize the number of actions required  to fill the buckets with the designated amount of
     water
                        new_cost = g + 1
                        if new_state not in visited_costs or new_cost < visited_costs[new_state
    ]:
                            visited_costs[new_state] = new_cost
                            h = heuristic(state, goal_state, jugs)
                            # In the problem statement the buckets are indexed starting from 1,
     thus must add 1 to the bucket_ind
                            heappush(queue, (new_cost + h, new_cost,  actions + [('+', jug,
    bucket_ind+1)], new_state))
                # Check if removing water from the bucket results in a valid state. The buckest
     cannot have a negative amount of water
                if state[bucket_ind] - jug >= 0:
                    temp_state = list(state)[:]
                    temp_state[bucket_ind] -= jug
                    # The constraint on the relative amount of water in the buckets based on
    their order must hold after this action
                    if all(temp_state[i] <= temp_state[i + 1] for i in range(len(temp_state) -
    1)):
                        new_state = tuple(temp_state)
                        new_cost = g + 1
                        if new_state not in visited_costs or new_cost < visited_costs[new_state
    ]:
                            visited_costs[new_state] = new_cost
                            h = heuristic(state, goal_state, jugs)
                            heappush(queue, (new_cost + h, new_cost, actions + [('-', jug,
    bucket_ind+1)], new_state))
    return None

def heuristic(buckets_state, buckets_goal, jugs):
    # The heuristic function can be a simulation of filling buckets greedily, using the next
     largest jug repeatedly as long as the amount of water in the bucket does not exceed the
     goal amount
```

The A\* prompt continues on the next page.

**Continuation of A\* prompt:**

```python
    # This heuristic is admissible because it is greedy, always opting for the action that
      fills the buckets the most, ensuring it never overestimates the cost to reach the goal
    # The heuristic is consistent as the estimated cost of the next node is higher if water is
      removed from a bucket, or equal or less by at most 1 (equal to the cost of reaching the
      successor node, ie one action) as the maximum amount of water that can be added to the
      bucket is by using the largest jug that won't cause an overflow, which is exactly the jug
      used to fill the bucket in the heuristic. Thus h(n) can never be greater than c(n, n')(
      equal to 1) + h(n')
    h = 0
    # Sort the jugs by decreasing capacity
    jugs = sorted(jugs, reverse=True)
    # Iterate through the buckets
    for i in range(len(buckets_state)):
        bucket_fill = buckets_state[i]
        goal = buckets_goal[i]
        # Fill the bucket using the next largest jug as long as the bucket does not overflows
        for jug in jugs:
            while bucket_fill + jug < goal:
                bucket_fill += jug
                # Increment the estimated cost to the goal by 1 actions
                h += 1
    return h

print(a_star())
```
'''
________

In the magic square problem, a 4x4 grid is filled with unique integers ranging from 29 to 54. Some numbers are already given, while others are unknown and represented as 'x'. The sums of columns must be None, 148, 196, None for columns 0 to 3 respectively, and the sums of rows must be None, 187, 149, None for rows 0 to 3 respectively, where None means that we do not have any constraints on the sum of the numbers in the row or column at that index. Also, the sum of the numbers in the diagonal from the top left to the bottom right corner of the grid should equal 166. The goal is to find unique integers in the given range to replace with 'x's in the grid below such that the sum of the specified rows, columns, and diagonal equals the given amounts and the sum of all of the numbers in the grid is as low as possible. The solution should be provided as a list of tuples in Python syntax. Each tuple should contain three numbers: the row index, the column index (both starting from 0), and the value of the unknown number at that position.\n\nGrid:\n [[47 x x 32]\n [x x x 49]\n [x 31 50 x]\n [x x 52 30]]
'''python

```python
import heapq
import math
import numpy as np

def a_star():
    # Define the initial state of the grid as a 2d tuple
    initial_state = (('47', 'x', 'x', '32'),
                     ('x', 'x', 'x', '49'),
                     ('x', '31', '50', 'x'),
                     ('x', 'x', '52', '30'))
    num_rows = 4
    num_cols = 4
    row_sums = [None, 187, 149, None]
    col_sums = [None, 148, 196, None]
    diagonal_sum = 166
    # Create the set of the valid numbers that could be in the grid
    numbers = set(range(29, 54))
    visited_costs = {}
    visited_costs[initial_state] = 0
    queue = [(0, 0, [], initial_state)]
    while queue:
        _, g, actions, state = heapq.heappop(queue)
        # The goal state must not have any unknown numbers, thus we need to get (the
        coordinates of) the unknown numbers to check whether the current state is the goal state
        x_coords = [(i, j) for i, row in enumerate(state) for j, element in enumerate(row) if
        element == 'x']
        if not x_coords:
```

The A\* prompt continues on the next page.

**Continuation of A* prompt:**

```
            # Convert the cells of the state to ints to calculate and compare the sum of the
        specific positions in the current state with the given goal sums
            state_array = np.array([[int(element) for element in row] for row in state])
            if (np.all([i == j for i, j in zip(np.sum(state_array, axis=0), col_sums) if j])
        and
                np.all([i == j for i, j in zip(np.sum(state_array, axis=1), row_sums) if j])
        and
                np.trace(state_array) == diagonal_sum):
                return actions
          # If the state has at least 1 remaining unknown number, generate all possible actions
        from the current state, which includes replacing the next x in the grid with any of unique
         integers in the range
          else:
            first_x_coord = x_coords[0]
            # The number must be unique and not be present in any other cells of the grid
            used_numbers  = set(int(cell) for row in state for cell in row if cell != 'x')
            for number in numbers:
                # Check if the new state, containing the new number, would be valid; ie the
        number must be unique and the sum of specified positions must not exceed the goal sums
        with the addition of the new number
                sum_x_row_new_state = sum(int(cell) for cell in state[first_x_coord[0]] if cell
         != 'x') + number
                sum_x_col_new_state = sum(int(state[k][first_x_coord[1]]) for k in range(
        num_rows) if state[k][first_x_coord[1]] != 'x') + number
                sum_diag_new_state = sum(int(state[k][k]) for k in range(num_rows) if state[k][
        k] != 'x') + number
                if (number not in used_numbers and
                    # If the x is in one of the rows with a given sum, then the sum of the new
        row, with addition of the number, must not exceed the target sum
                    (row_sums[first_x_coord[0]] is None or sum_x_row_new_state <= row_sums[
        first_x_coord[0]]) and
                    # Similarly, if the x position is in a column or the diagonal with a goal
        sum
                    (col_sums[first_x_coord[1]] is None or sum_x_col_new_state <= col_sums[
        first_x_coord[1]]) and
                    (first_x_coord[0] != first_x_coord[1] or sum_diag_new_state <= diagonal_sum
        )):
                    # Generate the new state
                    new_state = [list(row[:]) for row in state]
                    new_state[first_x_coord[0]][first_x_coord[1]] = str(number)
                    new_state = tuple(tuple(row) for row in new_state)
                    # The additional cost of this state is the value of the number replaced
        with x as we are trying to minimize the sum of the numbers in the grid
                    new_cost = g + number
                    if new_state not in visited_costs or new_cost < visited_costs[new_state]:
                        visited_costs[new_state] = new_cost
                        # Relaxing the constraints on the columns sums and the diagonal sum to
        calculate the heuristic
                        h = heuristic(new_state, row_sums, numbers)
                        heapq.heappush(queue, (new_cost + h, new_cost, actions + [(
        first_x_coord[0], first_x_coord[1], number)], new_state))
    return None

def heuristic(state, row_sums, numbers):
    # Relax the columns and diagonal sum constraints to compute an admissible and consistent
      heuristic
    # This heuristic considers the sum of differences between the given and current row sums,
      for rows with a specified sum value that have at least one unknown number, filling other x
       with the smallest unique number
    # The heuristic assumes xs can be filled with the smallest unique number satisfying row sum
       constraints, even if column or diagonal constraints are violated, thus reporting a lower
      cost estimate to reach the goal state, making it admissible
    # The heuristic is consistent, ie non-decreasing along the path to the goal state, because
      the cost of replacing an x in the grid with a number is the number itself, which is always
       greater than or equal to the number used to fill that position in the heuristic
    # Also the cost of the goal state is 0, as the sum of the rows equals their goal sums and
      there are no unknown numbers to fill in the grid
```

**Continuation of A* prompt:**

```python
    # Get numbers not used in the state currently
    used_numbers = set(int(cell) for row in state for cell in row if cell != 'x')
    available_numbers = sorted(list(numbers - used_numbers))
    h = 0
    for i in range(len(state)):
        if row_sums[i] is not None:
            row_full = True
            sum_row = sum(int(cell) for cell in state[i] if cell != 'x')
            # Check to see if the row with a target some has any more xs
            for cell in state[i]:
                if cell == 'x':
                    row_full = False
            if not row_full:
                h += row_sums[i] - sum_row
            # Otherwise the sum of a row must equal the target sum or there's no path to reach
    the goal state from this state. Return math.inf
            else:
                if sum_row != row_sums[i]:
                    return math.inf
        # For other rows, greedily fill the x positions with the smallest unique number in the
    range
        else:
            for cell in state[i]:
                if cell == 'x' and available_numbers:
                    h += available_numbers.pop(0)
    return h

print(a_star())
```

‘‘‘

______

Solve the below problem in the same format by encoding the problem states as a graph and implementing an A* search algorithm. Explain the rational behind each line of code using comments. The program must print the answer in the instructed format as a python list.

**problem statement (city_directed_graph instance 8):** We have a map of cities, each represented by a letter, and they are connected by one-way roads. The adjacency matrix below shows the connections between the cities. Each row and column represents a city, and a '1' signifies a direct road from the city of the row to the city of the column. The travel time between any two directly connected cities is the same. Currently, we are located in city 'U'. Our task is to visit city A and city X excatly twice. Determine the quickest route that allows us to visit both these destination cities, ensuring that we stop at the two destinations twice on our path. The sequence in which we visit the destination cities is not important. However, apart from X and A, we can only visit each city once on our path. Provide the solution as a list of the city names on our path, including the start, in Python syntax.\n\n Y O Q X K R A U M L \nY 0 1 0 0 0 1 1 1 1 1 \nO 0 0 0 0 0 0 1 0 0 0 \nQ 1 0 0 1 0 1 0 0 1 0 \nX 1 0 1 0 0 1 0 0 1 1 \nK 0 1 0 1 0 0 1 0 0 0 \nR 0 0 0 0 1 0 0 0 0 0 \nA 0 0 0 1 0 1 0 0 1 1 \nU 0 0 0 0 0 0 0 0 0 1 \nM 0 0 1 0 0 0 0 0 0 0 \nL 0 1 0 0 0 0 0 0 0 0 \n

**GPT4's generation:**
‘‘‘python

```python
import heapq

def a_star():
    # Define the adjacency matrix of the graph
    adjacency_matrix = {
        'Y': ['O', 'R', 'A', 'U', 'M', 'L'],
        'O': ['A'],
        'Q': ['Y', 'X', 'R', 'M'],
        'X': ['Y', 'Q', 'R', 'M', 'L'],
        'K': ['O', 'X', 'A'],
        'R': ['K'],
        'A': ['X', 'R', 'M', 'L'],
        'U': ['L'],
        'M': ['Q'],
        'L': ['O']}
    # Define the initial state and the goal state
    initial_state = ('U', frozenset(), 0)
    goal_state = 2
    # Initialize a dictionary to store the cost of reaching each visited state
    visited_costs = {}
    visited_costs[initial_state] = 0
    # Initialize a priority queue of states not yet visited, with the initial state as the
     first element
    queue = [(0, 0, [], initial_state)]
```

**Continuation of GPT4's generation:**

```
    # While there are un-visited states
    while queue:
        # Pop the state with the lowest sum of the cost so far and estimated cost to the goal
     from the queue
        _, g, actions, state = heapq.heappop(queue)
        # Check if the current state is the goal state
        if state[2] == goal_state:
            return actions
        # Generate all possible actions from the current state, which includes moving to any
     of the neighboring cities
        for neighbor in adjacency_matrix[state[0]]:
            # Check if the new state would be valid, ie if the city has not been visited
     before or is one of the destination cities
            if neighbor not in state[1] or neighbor in ['A', 'X']:
                # Generate the new state
                new_state = (neighbor, state[1].union({neighbor}), state[2] + (1 if neighbor
     in ['A', 'X'] else 0))
                # The cost so far is the number of cities visited, as the task is to minimize
     the number of cities visited
                new_cost = g + 1
                # If the new state is unvisited or we found a new path with a lower cost to
     reach this state, add it to the queue of not-yet-visited states
                if new_state not in visited_costs or new_cost < visited_costs[new_state]:
                    visited_costs[new_state] = new_cost
                    heapq.heappush(queue, (new_cost + heuristic(new_state, goal_state),
     new_cost, actions + [neighbor], new_state))
    return None

def heuristic(state, goal):
    # An admissible and consistent heuristic is the number of destination cities not yet
     visited twice
    # The heuristic is admissible because it never overestimates the cost to reach the goal
     state, as each destination city must be visited twice
    # The heuristic is consistent because the cost of moving to a neighboring city is always
     1, which is exactly the decrease in the heuristic cost, if the city is a destination city
     and has not been visited twice before
    return goal - state[2]

print(a_star())
```
"""

**LLM-code run result**: ['L', 'O', 'A', 'X']
**Solution is feasible! Solution is not correct.**

Table 31: GPT4's solution for a city_directed_graph problem using A* prompting method.

MSMT A* SECOND STAGE

**MSMT A\* second stage prompt**: The goal is to implement the initialize function for the given A* algorithm. The initialize function must initialize the visited_costs dictionary and the priority queue and encode all of the information about the initial state and other variables given in the problem (ie a given matrix, eligible actions, goal coordinate, initial state of the board, etc).
Examples:

In the 8-puzzle game, you are given a grid with numbered square tiles arranged randomly and one tile missing. The goal is to arrange the tiles in descending order by sliding them into the empty space. The tiles can move in 4 directions: left, right, up, and down. Given the initial state of the puzzle below, where the empty spot is represented as "_", provide the shortest list of tiles that need to be swapped with the empty spot to achieve the goal state. The goal state is when all tiles are in descending order, with the largest number in the top left corner, and the empty spot is in the bottom right corner. The solution should be a list of numbers in Python format, where each number represents the number on the tile that the empty spot is swapped with at each turn. Initial state of the puzzle: [[55, 43, 17], [97, 35, 9], [12, 25, '_']]
'''python

```python
import heapq

def a_star():
    # The initialize function initializes and returns the visited_costs dictionary and the priority
    #   queue and encodes all of the variables given in the problem (ie the initial and goal board
    #   and dimensions of the puzzle board)
    initial_state, goal_state, num_rows, num_cols, visited_costs, queue = initialize()
    # While there are un-visited states
    while queue:
        # Pop the state with the lowest sum of the cost so far and estimated cost to the goal from
        #   the queue
        _, g, actions, state = heapq.heappop(queue)
        # We can check if the current state is the goal state with a simple equality check, as the
        #   goal state is predefined
        if state == goal_state:
            return actions
        # Generate all valid actions from the current state, which includes swapping any of the
        #   tiles neighboring the empty spot, with the empty spot
        # Generate the coordinates of the tiles neighboring "_"
        empty_row, empty_col = [(i, j) for i in range(num_rows) for j in range(num_cols) if state[i
            ][j] == '_'][0]
        for d_row, d_col in [(0, -1), (0, 1), (1, 0), (-1, 0)]:
            swap_row, swap_col = empty_row + d_row, empty_col + d_col
            # Check if the swap is valid, ie if the coordinate of the tile to be swapped is a valid
            #   coordinate within the bounds of the board
            if 0 <= swap_row < num_rows and 0 <= swap_col < num_cols:
                # The actions is valid, generate the new state
                new_state = [list(row[:]) for row in state]
                number_to_be_swapped = new_state[swap_row][swap_col]
                # Do the swap
                new_state[empty_row][empty_col], new_state[swap_row][swap_col] = new_state[swap_row
                    ][swap_col], new_state[empty_row][empty_col]
                new_state = tuple(tuple(row) for row in new_state)
                # The cost so far is the number of swaps made, as our objective is to minimize the
                #   number of swaps required to reach the goal state
                new_cost = g + 1
                # If the new state is unvisited or we found a new path with a lower cost to reach
                #   this state, add it to the queue of not-yet-visited states
                if new_state not in visited_costs or new_cost < visited_costs[new_state]:
                    visited_costs[new_state] = new_cost
                    heapq.heappush(queue, (new_cost + heuristic(new_state, goal_state), new_cost,
                        actions + [number_to_be_swapped], new_state))
    return None
```

**Continuation of MSMT A\* prompt:**

```python
def heuristic(state, goal):
    # An admissible and consistent heuristic is the sum of the Manhattan distances (the
      shortest path) of each tile from its goal position
    # The heuristic relaxes the constraint that a tile can only be swapped with the empty spot
      and presumes we can move the tiles to their goal position by swapping them with any of the
      other tiles
    # Thus the heuristic reports a lower estimate on the cost to reach goal state and is
      admissible
    # The heuristic is consistent because the cost of moving a tile to a neighboring coordinate
      is always 1, which is exactly the decrease in the Manhattan distance, if the tile is
      moved toward its goal position, otherwise the estimated cost of the successor node is the
      same or higher, and he heuristic estimate for the goal state is 0, as the distance of each
      tile from its goal position would be 0 in the goal state.
    h = 0
    for i in range(len(state)):
        for j in range(len(state[i])):
            # Can't compare integers with "_" when finding the goal position of each tile, thus
      ignore the "_" tile
            if state[i][j] != '_':
                # Get goal position of each tile
                goal_row, goal_col = [(x, y) for x in range(len(goal)) for y in range(len(goal[
      x])) if goal[x][y] == state[i][j]][0]
                # Add the the Manhattan distance of the current and goal coordinates of the
      tile to the estimate
                h += abs(i - goal_row) + abs(j - goal_col)
    return h

print(a_star())
```

The target initialize function:
'''python

```python
def initialize():
    # Define the initial state and the goal state of the puzzle, represented as 3x3 tuples
    initial_state = ((55, 17, 17), (97, 35, 9), (12, 25, '_'))
    goal_state = ((97, 55, 43), (35, 25, 17), (12, 9, '_'))
    # Encoding other variables given in the problem statement
    num_rows = 3
    num_cols = 3
    # Initialize a dictionary to store the cost of reaching each visited state
    visited_costs = {}
    visited_costs[initial_state] = 0
    # Initialize a priority queue of states not yet visited, with the initial state as the
      first element. The priority of each element is the cost to reach that state (g) + the
      estimate remaining cost (h) to reach the goal
    # Record the swaps required to get to each state in a list; no swaps performed to reach the
      initial state
    queue = [(0, 0, [], initial_state)]
    return initial_state, goal_state, num_rows, num_cols, visited_costs, queue
```

'''

————

The game of 'Sort It' begins with 3 tubes, each filled with 4 balls of different colors. The goal is to sort the balls by color, with each tube containing balls of only one color. Only one ball can be moved at a time, taken from the top of one tube and placed on top of another. The capacity of each tube (maximum number of balls we can fit in each tube) is 6 balls. It is not allowed to place a ball in a tube that already has 6 balls. The solution should be a list of tuples, each containing, first, the index of the tube from which a ball is taken and, second, the index of the tube to which it is moved, indexing from 0. Given the initial state of the tubes, represented by the lists below (with the leftmost item being the color of the topmost ball in each tube), what is the shortest list of move tuples that will result in all the balls being correctly sorted? [['Green', 'Red', 'Green', 'Red'], ['Blue', 'Blue', 'Red', 'Green'], ['Red', 'Blue', 'Green', 'Blue']]

The A\* prompt continues on the next page.

**Continuation of A\* prompt**:

```python
import heapq
from collections import Counter

def a_star():
    # The initialize function initializes and returns the visited_costs dictionary and the
      priority queue and encodes all of the variables given in the problem (ie the initial state
       of the tubes, number of tubes, and capacity of each tube)
    initial_state, num_tubes, capacity, visited_costs, queue = initialize()
    # While there are un-visited states
    while queue:
        # Pop the state with the lowest sum of the cost so far and estimated cost to the goal
          from the queue
        _, g, actions, state = heapq.heappop(queue)
        # Check of the current state is the goal state
        # The goal state is where each tube only contains balls of 1 single color
        if all(len(set(tube)) <= 1 for tube in state):
            return actions
        # Generate all possible actions from the current state, which includes moving a ball
          from any of the 3 tubes to another tube
        for from_tube_ind in range(num_tubes):
            for to_tube_ind in range(num_tubes):
                # Check if the new state would be valid, ie from_tube and to_tube must not be
                  the same tube
                # And from_tube must at least have 1 ball to move and the to_tube cannot be at
                  capacity
                if from_tube_ind != to_tube_ind and state[from_tube_ind] and len(state[
                  to_tube_ind]) < capacity:
                    # Generate the new state
                    new_state = [list(tube[:]) for tube in state]
                    # The ball to move is the topmost ball in the from_tube, at index 0
                    ball_to_move = new_state[from_tube_ind].pop(0)
                    # Add the ball to the top of the to_tube
                    new_state[to_tube_ind].insert(0, ball_to_move)
                    new_state = tuple(tuple(tube) for tube in new_state)
                    # The cost so far is the number of moves made, as the task is to minimize
                      the number of moves required
                    new_cost = g + 1
                    # If the new state is unvisited or we found a new path with a lower cost to
                      arrive at this state, add it to the queue of un-visited states
                    if new_state not in visited_costs or new_cost < visited_costs[new_state]:
                        visited_costs[new_state] = new_cost
                        heapq.heappush(queue, (new_cost + heuristic(new_state), g + 1, actions
                          + [(from_tube_ind, to_tube_ind)], new_state))
    return None

def heuristic(tubes):
    # An admissible and consistent heuristic for this problem is the count of balls that are
      not the same color as the most frequent color in their tube
    # This heuristic relaxes the constraint that only the ball at the top of the tube can be
      moved
    # It is admissible because it never overestimates the cost to reach the goal, as each
      mismatched ball must be moved at least once
    # It's consistent because moving a ball from one tube to another reduces the heuristic cost
       of the successor node by a max of 1 (if the moved ball's color matches the most common
      color in the new tube but not in the old one), which is equal to the cost of reaching the
      successor node
    # Thus h(s) is always less than or equal to c(s, n)(equal to 1) + h(n)
    h = 0
    for tube in tubes:
        if tube:
            # If there are ties in the frequency of colors, the most_commonm_color must be
              match the color of the balls lower that are in the tube, as moving lower balls is costlier
            reversed_tube = tube[:]
            reversed_tube = reversed_tube[::-1]
            # Get the most common color
            most_common_color = Counter(reversed_tube).most_common(1)[0][0]
            for ball in tube:
                if ball != most_common_color:
                    h += 1
    return h
print(a_star())
```

72

```
```

The MSMT A\* prompt continues on the next page.

**Continuation of MSMT A\* prompt**:
The target initialize function:
```python
def initialize():
    # Define the initial state of the tubes, as a 2d tuple of color of the balls in tubes 0 to
      2
    initial_state = (('Green', 'Red', 'Green', 'Red'), ('Blue', 'Blue', 'Red', 'Green'), ('Red'
      , 'Blue', 'Green', 'Blue'))
    # Encoding other variables given in the problem statement
    num_tubes = 3
    capacity = 6
    # Initialize a dictionary to store the cost of reaching each visited state
    visited_costs = {}
    visited_costs[initial_state] = 0

    # Initialize a priority queue of states not yet visited, with the initial state as the
      first element. The priority of each element is the cost to reach that state (g) + the
      estimate remaining cost (h) to reach the goal
    # Record the actions required to get to each state in a list; no actions performed to reach
      the initial state
    queue = [(0, 0, [], initial_state)]

    return initial_state, num_tubes, capacity, visited_costs, queue
```

_____

Given 6 labeled water jugs with capacities 37, 133, 38, 72, 41, 23, 122 liters, we aim to fill 3 unlabeled buckets, numbered 1 to 3 and arranged in a line in ascending order, with 195, 224, 268 liters of water respectively. The amount of water in each unlabeled bucket can not at any point in time exceed the amount of water in the bucket placed before it. Jugs can only be filled to the top and emptied completely, and the unlabeled buckets cannot be overfilled. An action, represented as a tuple ('+', X, Y) or ('-', X, Y), involves adding to or removing water from the unlabeled bucket numbered Y, using the jug with capacity X. Determine the shortest sequence of actions needed to fill the buckets as specified, and present the solution as a list of action tuples in Python syntax.
```python
from heapq import heappush, heappop

def a_star():
    jugs, goal_state, initial_state, num_buckets, visited_costs, queue = initialize()
    while queue:
        _, g, actions, state = heappop(queue)
        # If the amount of water in the buckets in the current state equal the goal amounts,
      return the actions taken
        if state == goal_state:
            return actions
        # Generate all possible actions from the current state, which includes adding or
      subtracting water using any of the 6 jugs to any of the 3 buckets
        # Iterating through capacities of jugs and index of buckets as the action tuples must
      include the operation ('+' or '-'), capacity of the jug used, and the index of the bucket
      affected
        for jug in jugs:
            for bucket_ind in range(num_buckets):
                # Check if adding water using the current jug results in a valid state, ie the
      addition must not result in overflowing any of the buckets
                if (state[bucket_ind] + jug <= goal_state[bucket_ind]):
                    temp_state = list(state)[:]
                    temp_state[bucket_ind] += jug
                    # And the new state must maintain the constraint on the relative amount of
      water in the buckets based on their order
                    if all(temp_state[i] <= temp_state[i + 1] for i in range(len(temp_state) -
      1)):
                        # Generate the new state
                        new_state = tuple(temp_state)
                        # The cost so far is the number of actions taken, as the task is to
      minimize the number of actions required to fill the buckets with the designated amount of
       water
                        new_cost = g + 1
```

73

The MSMT A\* prompt continues on the next page.

**Continuation of MSMT A\* prompt:**

```python
                            if new_state not in visited_costs or new_cost < visited_costs[new_state]:
                                visited_costs[new_state] = new_cost
                                h = heuristic(state, goal_state, jugs)
                                # In the problem statement the buckets are indexed starting from 1,
        thus must add 1 to the bucket_ind
                                heappush(queue, (new_cost + h, new_cost,  actions + [('+', jug,
        bucket_ind+1)], new_state))
                    # Check if removing water from the bucket results in a valid state. The buckest
        cannot have a negative amount of water
                    if state[bucket_ind] - jug >= 0:
                        temp_state = list(state)[:]
                        temp_state[bucket_ind] -= jug
                        # The constraint on the relative amount of water in the buckets based on their
        order must hold after this action
                        if all(temp_state[i] <= temp_state[i + 1] for i in range(len(temp_state) - 1)):
                            new_state = tuple(temp_state)
                            new_cost = g + 1
                            if new_state not in visited_costs or new_cost < visited_costs[new_state]:
                                visited_costs[new_state] = new_cost
                                h = heuristic(state, goal_state, jugs)
                                heappush(queue, (new_cost + h, new_cost, actions + [('-', jug,
        bucket_ind+1)], new_state))
    return None

def heuristic(buckets_state, buckets_goal, jugs):
    # The heuristic function can be a simulation of filling buckets greedily, using the next
      largest jug repeatedly as long as the amount of water in the bucket does not exceed the goal
      amount
    # This heuristic is admissible because it is greedy, always opting for the action that fills
      the buckets the most, ensuring it never overestimates the cost to reach the goal
    # The heuristic is consistent as the estimated cost of the next node is higher if water is
      removed from a bucket, or equal or less by at most 1 (equal to the cost of reaching the
      successor node, ie one action) as the maximum amount of water that can be added to the bucket
       is by using the largest jug that won't cause an overflow, which is exactly the jug used to
      fill the bucket in the heuristic. Thus h(n) can never be greater than c(n, n')(equal to 1) +
      h(n')
    h = 0
    # Sort the jugs by decreasing capacity
    jugs = sorted(jugs, reverse=True)
    # Iterate through the buckets
    for i in range(len(buckets_state)):
        bucket_fill = buckets_state[i]
        goal = buckets_goal[i]
        # Fill the bucket using the next largest jug as long as the bucket does not overflows
        for jug in jugs:
            while bucket_fill + jug < goal:
                bucket_fill += jug
                # Increment the estimated cost to the goal by 1 actions
                h += 1
    return h

print(a_star())
```
```

The target initialize function:
```python
def initialize():
    # Define the capacities of the jugs, the goal state, and initial state, with states having an
      immutable data type
    jugs = [37, 133, 38, 72, 41, 23, 122]
    goal_state = (195, 224, 268)
    initial_state = (0, 0, 0)
    num_buckets = 3
    visited_costs = {}
    visited_costs[initial_state] = 0
    queue = []
    # No actions taken in the initial state
    heappush(queue, (0, 0, [], initial_state))
    return jugs, goal_state, initial_state, num_buckets, visited_costs, queue
```
```

74

The MSMT A\* prompt continues on the next page.

**Continuation of MSMT A\* prompt:**
————

In the magic square problem, a 4x4 grid is filled with unique integers ranging from 29 to 54. Some numbers are already given, while others are unknown and represented as 'x'. The sums of columns must be None, 148, 196, None for columns 0 to 3 respectively, and the sums of rows must be None, 187, 149, None for rows 0 to 3 respectively, where None means that we do not have any constraints on the sum of the numbers in the row or column at that index. Also, the sum of the numbers in the diagonal from the top left to the bottom right corner of the grid should equal 166. The goal is to find unique integers in the given range to replace with 'x's in the grid below such that the sum of the specified rows, columns, and diagonal equals the given amounts and the sum of all of the numbers in the grid is as low as possible. The solution should be provided as a list of tuples in Python syntax. Each tuple should contain three numbers: the row index, the column index (both starting from 0), and the value of the unknown number at that position.\n\nGrid:\n [[47 x x 32]\n [x x x 49]\n [x 31 50 x]\n [x x 52 30]]

```python
import heapq
import math
import numpy as np

def a_star():
    initial_state, num_rows, num_cols, row_sums, col_sums, diagonal_sum, numbers, visited_costs,
      queue = initialize()
    while queue:
        _, g, actions, state = heapq.heappop(queue)
        # The goal state must not have any unknown numbers, thus we need to get (the coordinates of
        ) the unknown numbers to check whether the current state is the goal state
        x_coords = [(i, j) for i, row in enumerate(state) for j, element in enumerate(row) if
      element == 'x']
        if not x_coords:
            # Convert the cells of the state to ints to calculate and compare the sum of the
      specific positions in the current state with the given goal sums
            state_array = np.array([[int(element) for element in row] for row in state])
            if (np.all([i == j for i, j in zip(np.sum(state_array, axis=0), col_sums) if j]) and
                np.all([i == j for i, j in zip(np.sum(state_array, axis=1), row_sums) if j]) and
                np.trace(state_array) == diagonal_sum):
                return actions
        # If the state has at least 1 remaining unknown number, generate all possible actions from
      the current state, which includes replacing the next x in the grid with any of unique
      integers in the range
        else:
            first_x_coord = x_coords[0]
            # The number must be unique and not be present in any other cells of the grid
            used_numbers  = set(int(cell) for row in state for cell in row if cell != 'x')
            for number in numbers:
                # Check if the new state, containing the new number, would be valid; ie the number
      must be unique and the sum of specified positions must not exceed the goal sums with the
      addition of the new number
                sum_x_row_new_state = sum(int(cell) for cell in state[first_x_coord[0]] if cell !=
      'x') + number
                sum_x_col_new_state = sum(int(state[k][first_x_coord[1]]) for k in range(num_rows)
      if state[k][first_x_coord[1]] != 'x') + number
                sum_diag_new_state = sum(int(state[k][k]) for k in range(num_rows) if state[k][k]
      != 'x') + number
                if (number not in used_numbers and
                    # If the x is in one of the rows with a given sum, then the sum of the new row,
       with addition of the number, must not exceed the target sum
                    (row_sums[first_x_coord[0]] is None or sum_x_row_new_state <= row_sums[
      first_x_coord[0]]) and
                    # Similarly, if the x position is in a column or the diagonal with a goal sum
                    (col_sums[first_x_coord[1]] is None or sum_x_col_new_state <= col_sums[
      first_x_coord[1]]) and
                    (first_x_coord[0] != first_x_coord[1] or sum_diag_new_state <= diagonal_sum)):
                    # Generate the new state
                    new_state = [list(row[:]) for row in state]
                    new_state[first_x_coord[0]][first_x_coord[1]] = str(number)
                    new_state = tuple(tuple(row) for row in new_state)
                    # The additional cost of this state is the value of the number replaced with x
      as we are trying to minimize the sum of the numbers in the grid
                    new_cost = g + number
```

**Continuation of MSMT A\* prompt:**

```python
                        if new_state not in visited_costs or new_cost < visited_costs[new_state]:
                            visited_costs[new_state] = new_cost
                            # Relaxing the constraints on the columns sums and the diagonal sum to
        calculate the heuristic
                            h = heuristic(new_state, row_sums, numbers)
                            heapq.heappush(queue, (new_cost + h, new_cost, actions + [(
        first_x_coord[0], first_x_coord[1], number)], new_state))
        return None

def heuristic(state, row_sums, numbers):
    # Relax the columns and diagonal sum constraints to compute an admissible and consistent
      heuristic
    # This heuristic considers the sum of differences between the given and current row sums,
      for rows with a specified sum value that have at least one unknown number, filling other x
       with the smallest unique number
    # The heuristic assumes xs can be filled with the smallest unique number satisfying row sum
       constraints, even if column or diagonal constraints are violated, thus reporting a lower
      cost estimate to reach the goal state, making it admissible
    # The heuristic is consistent, ie non-decreasing along the path to the goal state, because
      the cost of replacing an x in the grid with a number is the number itself, which is always
       greater than or equal to the number used to fill that position in the heuristic
    # Also the cost of the goal state is 0, as the sum of the rows equals their goal sums and
      there are no unknown numbers to fill in the grid
    # Get numbers not used in the state currently
    used_numbers = set(int(cell) for row in state for cell in row if cell != 'x')
    available_numbers = sorted(list(numbers - used_numbers))
    h = 0
    for i in range(len(state)):
        if row_sums[i] is not None:
            row_full = True
            sum_row = sum(int(cell) for cell in state[i] if cell != 'x')
            # Check to see if the row with a target some has any more xs
            for cell in state[i]:
                if cell == 'x':
                    row_full = False
            if not row_full:
                h += row_sums[i] - sum_row
            # Otherwise the sum of a row must equal the target sum or there's no path to reach
        the goal state from this state. Return math.inf
            else:
                if sum_row != row_sums[i]:
                    return math.inf
        # For other rows, greedily fill the x positions with the smallest unique number in the
        range
        else:
            for cell in state[i]:
                if cell == 'x' and available_numbers:
                    h += available_numbers.pop(0)
    return h

print(a_star())
```
```

The target initialize function:
```python
def initialize():
    # Define the initial state of the grid as a 2d tuple
    initial_state = (('47', 'x', 'x', '32'),
                     ('x', 'x', 'x', '49'),
                     ('x', '31', '50', 'x'),
                     ('x', 'x', '52', '30'))
    num_rows = 4
    num_cols = 4
    row_sums = [None, 187, 149, None]
    col_sums = [None, 148, 196, None]
    diagonal_sum = 166
    # Create the set of the valid numbers that could be in the grid
    numbers = set(range(29, 54))
    visited_costs = {}
    visited_costs[initial_state] = 0
    queue = [(0, 0, [], initial_state)]
    return initial_state, num_rows, num_cols, row_sums, col_sums, diagonal_sum, numbers,
      visited_costs, queue
```
```

76

The MSMT A\* continues on the next page.

**Continuation of MSMT A\* prompt:**

**problem statement (city_directed_graph instance 8)**: We have a map of cities, each represented by a letter, and they are connected by one-way roads. The adjacency matrix below shows the connections between the cities. Each row and column represents a city, and a '1' signifies a direct road from the city of the row to the city of the column. The travel time between any two directly connected cities is the same. Currently, we are located in city 'U'. Our task is to visit city A and city X excatly twice. Determine the quickest route that allows us to visit both these destination cities, ensuring that we stop at the two destinations twice on our path. The sequence in which we visit the destination cities is not important. However, apart from X and A, we can only visit each city once on our path. Provide the solution as a list of the city names on our path, including the start, in Python syntax.\n\n Y O Q X K R A U M L \nY 0 1 0 0 0 1 1 1 1 1 \nO 0 0 0 0 0 0 0 1 0 0 0 \nQ 1 0 0 1 0 1 0 0 1 0 \nX 1 0 1 0 0 1 0 0 1 1 \nK 0 1 0 1 0 0 1 0 0 0 \nR 0 0 0 0 1 0 0 0 0 0 \nA 0 0 0 1 0 1 0 0 1 1 \nU 0 0 0 0 0 0 0 0 0 1 \nM 0 0 1 0 0 0 0 0 0 0 \nL 0 1 0 0 0 0 0 0 0 0 \n

**The A\* algorithm generated by GPT4 for the city_directed_graph problem type during the first stage of MSMT that passed the unit tests:** ```python

```python
import heapq

def a_star():
    # The initialize function initializes and returns the visited_costs dictionary and the priority
      queue and encodes all of the variables given in the problem (ie the initial and goal board
      and dimensions of the puzzle board)
    initial_state, goal_state, adjacency_matrix, cities, visited_costs, queue = initialize()
    # While there are un-visited states
    while queue:
        # Pop the state with the lowest sum of the cost so far and estimated cost to the goal from
        the queue
        _, g, actions, state = heapq.heappop(queue)
        current_city, remaining_cities = state
        # We can check if the current state is the goal state with a simple equality check, as the
        goal state is predefined
        if not remaining_cities:
            return actions
        # Generate all valid actions from the current state, which includes moving to any of the
        cities connected to the current city
        for i in range(len(cities)):
            if adjacency_matrix[cities.index(current_city)][i] == 1:
                next_city = cities[i]
                # Check if the next city is a destination city and we have not visited it twice
        already
                if next_city in remaining_cities:
                    new_remaining_cities = list(remaining_cities)
                    new_remaining_cities.remove(next_city)
                    new_state = (next_city, tuple(new_remaining_cities))
                    new_cost = g + 1
                    if new_state not in visited_costs or new_cost < visited_costs[new_state]:
                        visited_costs[new_state] = new_cost
                        heapq.heappush(queue, (new_cost + heuristic(new_state, goal_state),
        new_cost, actions + [next_city], new_state))
                # If the next city is not a destination city and we have not visited it before, we
        can move to it
                elif next_city not in actions:
                    new_state = (next_city, remaining_cities)
                    new_cost = g + 1
                    if new_state not in visited_costs or new_cost < visited_costs[new_state]:
                        visited_costs[new_state] = new_cost
                        heapq.heappush(queue, (new_cost + heuristic(new_state, goal_state),
        new_cost, actions + [next_city], new_state))
    return None

def heuristic(state, goal_state):
    # An admissible and consistent heuristic is the number of remaining cities to visit
    # The heuristic relaxes the constraint that we can only move to cities connected to the current
      city and presumes we can move to any of the remaining cities directly from the current city
    # Thus the heuristic reports a lower estimate on the cost to reach goal state and is admissible
    # The heuristic is consistent because the cost of moving to a city is always 1, which is
      exactly the decrease in the number of remaining cities to visit, if the city is a destination
      city, otherwise the estimated cost of the successor node is the same or higher, and he
      heuristic estimate for the goal state is 0, as there are no remaining cities to visit in the
      goal state.
    return len(state[1])

print(a_star())
```

77

```

The MSMT A\* continues on the next page.

**Continuation of MSMT A\* prompt:**

An example implementation of the target initialize function. Crucial: the value of variables must be updated according to the problem statement, the implementation above is with hypothetical values:

```python
def initialize():
    # Define the initial state and the goal state of the puzzle, represented as 3x3 tuples
    initial_state = 'H'
    goal_state = ['K', 'K', 'Z', 'Z']
    # Encoding other variables given in the problem statement
    adjacency_matrix = [[0, 0, 1, 1, 0, 0, 1, 1, 0, 1],
                        [0, 0, 0, 0, 0, 0, 0, 1, 0, 1],
                        [0, 1, 0, 1, 0, 1, 1, 0, 0, 1],
                        [0, 1, 1, 0, 1, 0, 0, 0, 0, 0],
                        [1, 1, 0, 0, 0, 0, 0, 0, 0, 1],
                        [0, 0, 0, 1, 0, 0, 1, 0, 1, 0],
                        [0, 0, 1, 1, 1, 0, 0, 0, 0, 0],
                        [1, 0, 0, 0, 0, 0, 1, 0, 0, 1],
                        [0, 0, 0, 0, 0, 1, 0, 0, 0, 0],
                        [0, 0, 0, 0, 1, 0, 0, 0, 0, 0]]
    cities = ['K', 'B', 'Z', 'S', 'X', 'H', 'O', 'E', 'U', 'V']
    # Initialize a dictionary to store the cost of reaching each visited state
    visited_costs = {}
    visited_costs[(initial_state, tuple(goal_state))] = 0
    # Initialize a priority queue of states not yet visited, with the initial state as the first
    #   element. The priority of each element is the cost to reach that state (g) + the estimate
    #   remaining cost (h) to reach the goal
    # Record the actions required to get to each state in a list; no actions performed to reach the
    #   initial state
    queue = [(0, 0, [], (initial_state, tuple(goal_state)))]
    return initial_state, goal_state, adjacency_matrix, cities, visited_costs, queue
```

Initialize the visited_costs dictionary and the priority queue and encode all the variables given of only the last problem in the initialize function as the examples provided above. The value of variables must match the information encoded in the problem statement.

The target initialize function:

**GPT4's generation:**

```python
def initialize():
    # Define the initial state and the goal state of the puzzle, represented as 3x3 tuples
    initial_state = ('U', ('A', 'A', 'X', 'X'))
    goal_state = []
    # Encoding other variables given in the problem statement
    adjacency_matrix = [[0, 1, 0, 0, 0, 1, 1, 1, 1, 1],
                        [0, 0, 0, 0, 0, 1, 0, 0, 0, 0],
                        [1, 0, 0, 1, 0, 1, 0, 0, 1, 0],
                        [1, 0, 1, 0, 0, 1, 0, 0, 1, 1],
                        [0, 1, 0, 1, 0, 0, 1, 0, 0, 0],
                        [0, 0, 0, 0, 1, 0, 0, 0, 0, 0],
                        [0, 0, 0, 1, 0, 1, 0, 0, 1, 1],
                        [0, 0, 0, 0, 0, 0, 0, 0, 0, 1],
                        [0, 0, 1, 0, 0, 0, 0, 0, 0, 0],
                        [0, 1, 0, 0, 0, 0, 0, 0, 0, 0]]
    cities = ['Y', 'O', 'Q', 'X', 'K', 'R', 'A', 'U', 'M', 'L']
    # Initialize a dictionary to store the cost of reaching each visited state
    visited_costs = {}
    visited_costs[initial_state] = 0
    # Initialize a priority queue of states not yet visited, with the initial state as the first
    #   element. The priority of each element is the cost to reach that state (g) + the estimate
    #   remaining cost (h) to reach the goal
    # Record the actions required to get to each state in a list; no actions performed to reach the
    #   initial state
    queue = [(0, 0, [initial_state[0]], initial_state)]
    return initial_state, goal_state, adjacency_matrix, cities, visited_costs, queue
```

**LLM-code run result**: ['U', 'L', 'O', 'A', 'X', 'Y', 'A', 'X']

**Solution is feasible! Solution is correct! Solution is optimal! Computation time of the LLM code is 97% of the our A\* code computation time.** 78

Table 32: GPT4's solution for a city_directed_graph problem using MSMT A\* prompting method.

## I  HOSTING, LICENSING, AND MAINTENANCE

We accept responsibility for any violations of rights that might have occurred in the curation of this dataset. We affirm that the dataset is composed solely of search problems and does not include any sensitive information. The data and code associated with SearchBench are licensed under the Creative Commons (CC BY-SA) license, ensuring open access and usability for the research community.

To ensure the long-term availability and preservation of the SearchBench dataset, we have hosted it on both Hugging Face and GitHub. Moreover, we will provide full access to the code for prompting and inference methods, as well as automated pipelines for generating and evaluating an arbitrary number of instances though these platforms, after the double blind review period. We are committed to maintaining the dataset on these platforms with continued open access. Additionally, we anticipate releasing future versions of this dataset with increased scalability.

