# OpenReview forum: "Navigating the Labyrinth: Evaluating and Enhancing LLMs’ Ability to Reason About Search Problems"
_ICLR.cc/2025/Conference — ICLR 2025 Conference Withdrawn Submission_

### Official Review · Reviewer_fqts · 2024-10-31

**Soundness:** 2
**Presentation:** 2
**Contribution:** 2
**Rating:** 3
**Confidence:** 4

**Summary:**

This paper presents SearchBench, a benchmark evaluating large language models (LLMs) on complex combinatorial search tasks that require multi-step reasoning and backtracking. With 11 unique problem types across five categories, SearchBench challenges LLMs by avoiding training contamination and requiring reasoning-intensive solutions. The authors introduce A* search prompting and a Multi-Stage-Multi-Try (MSMT) strategy, which breaks A* implementation into verifiable steps, improving GPT-4’s success rate to over 57% on some tasks. Despite these advances, results reveal LLM limitations in achieving optimal solutions in multi-hop complex problem-solving.

**Strengths:**

1. The use of classic optimization problems with unique rule modifications to prevent LLM familiarity from pre-training is well-designed.
2. SearchBench is a legit benchmark for evaluating LLMs on complex tasks; commendable effort in its generation and diversity.

**Weaknesses:**

1. Using demonstrations from other problem categories deviates from the few-shot learning definition and may serve as distractors[1], which could lower baseline performance.
2. MSMT A* lacks novelty, as combining code generation with external feedback (e.g., compiler feedback and unit test filtering) is now a standard technique in LLM optimization.
3. Presentation could improve: font issues in Figure 3, misuse of "accuracy" on the y-axis of Figure 5, and some redundancy in explanations.

[1] Shi, F., Chen, X., Misra, K., Scales, N., Dohan, D., Chi, E. H., ... & Zhou, D. (2023, July). Large language models can be easily distracted by irrelevant context. In International Conference on Machine Learning (pp. 31210-31227). PMLR.

**Questions:**

1. [Figure3] Are there results for GPT-o1 with A* and MSMT A*, and Code Llama with 0-shot text?
2. Line 268 suggests that models capable of solving SearchBench can generalize to other combinatorial problems; are there experimental results supporting this claim?
3. MSMT A* benefits from multiple tries and unit test prefiltering, which naturally boosts feasibility rates. Would giving other methods an equivalent number of trials make for a fairer comparison?

---

> ### Author Response · Authors · 2024-11-28
> **Rebuttal**
>
> Thank you for your time and effort in reviewing our paper. Our primary focus was to introduce a comprehensive search and planning benchmark for LLMs. The A* MSMT serves as a baseline on this benchmark, demonstrating the current models' ability to design sophisticated search algorithms, offloading the execution of numerous iterations of the algorithm to an external interpreter. However, the benchmark itself is our main contribution, and it remains challenging even for state-of-the-art models like GPT-4 and GPT-01, especially when tasked with solving problems end-to-end.
>
> >Using demonstrations from other problem categories deviates from the few-shot learning definition and may serve as distractors[1], which could lower baseline performance.
>
> We did provide results using zero-shot prompts in both text and code, where models were instructed to solve a given problem without any other information or examples. Figure 3 demonstrates that the performance of A* and MSMT A* is significantly higher than zero-shot performance. From this comparison we can draw the conclusion that the solved examples in the prompts do not act as distractions. While A* implementations for different problem types vary, the general structure of the algorithm and the conversion of problem states into graph nodes are consistent across combinatorial problems, aiding model performance.
>
> >MSMT A* lacks novelty, as combining code generation with external feedback (e.g., compiler feedback and unit test filtering) is now a standard technique in LLM optimization.
>
> While various prompting and inference methods have been used to enhance model reasoning, our contribution lies in the novel application of these methods using the A* search algorithm, a powerful yet complex algorithm to implement.
>
> Moreover, our main motivation for A* MSMT was to show that language models struggle with combinatorial problems due to the inherent nonlinearity of computations involved in evaluating each state within the problem's state space, such as calculating cost and heuristic of each node, generating child nodes, and determining their feasibility . Our results show that while LLMs struggle with performing these simple computations end to end, they excel at writing complex search plans when the execution of many iterations of these plans is offloaded to a Python interpreter, highlighting that nonlinearity is a bottleneck in LLM reasoning. Finally, it’s important to note that our primary contribution was presenting the challenging SearchBench benchmark.
>
> >Are there results for GPT-o1 with A* and MSMT A*, and Code Llama with 0-shot text?
>
> Code Llama is specifically trained for code generation, so we did not evaluate it on text-based approaches. At the time of our experiments, GPT-01 did not support the context length required for A* and MSMT A* approaches.
>
> >Line 268 suggests that models capable of solving SearchBench can generalize to other combinatorial problems; are there experimental results supporting this claim?
>
> Our claim that models capable of solving SearchBench can generalize to other combinatorial problems is based on the fact that each problem type in SearchBench is selected from representative categories in combinatorial problems. These problems have been uniquely modified to ensure they do not resemble previously solved problems, making them new combinatorial challenges. Moreover, given that NP-hard problems can be reduced to each other, solving these new problems suggests broader generalization capabilities.
>
> >MSMT A* benefits from multiple tries and unit test prefiltering, which naturally boosts feasibility rates. Would giving other methods an equivalent number of trials make for a fairer comparison?
>
> While using multiple tries for any method and averaging over the answers could improve performance, MSMT's strength lies in its ease of use and the simplicity of calculating unit tests. Implementing multiple tries for text-based approaches would require a method to evaluate and compare intermediate generations (note that the answer to our problems is a list of actions, making methods like averaging and majority vote inapplicable as the set of feasible and/or correct solutions is unbounded), which is complex and typically involves training a reward model. This is a complex task requiring supervised datasets and is outside the scope of our current work, our primary contribution was to introduce a robust and challenging benchmark of search and planning problems.
>
>
> We hope this clarifies our contributions and the rationale behind our approach. Thank you for your feedback.

---

### Official Review · Reviewer_1qot · 2024-11-03

**Soundness:** 3
**Presentation:** 3
**Contribution:** 2
**Rating:** 5
**Confidence:** 4

**Summary:**

The work investigates the ability of Large Language Models (LLMs) to solve complex combinatorial search problems that require multi-step reasoning and backtracking. The paper introduces a new benchmark referred as to SearchBench. This new benchmark dataset has 11 unique combinatorial problems that examines LLMs with tasks involving state-based search and backtracking. The authors analyze the feasibility, correctness, and optimality of solutions generated by LLMs. They reported that even advanced models like GPT-4 severely struggle with such tasks.
The authors then propose an A* prompting strategy to guide LLMs in implementing an informed search algorithm (A*). They also presented a Multi-Stage-Multi-Try (MSMT) approach that decomposes the A* algorithm into two stages with unit test verifications, significantly improving model performance. Experimental results show that the MSMT method helps models achieve higher accuracy. Despite these improvements, the authors still observed challenges in optimality and broader reasoning persist.
Overall, the work contributes SearchBench as a robust benchmark and MSMT A* prompting as an effective strategy for enhancing LLM reasoning capabilities on complex search problems.

**Strengths:**

The paper’s strengths are as follows:

1. The most important contribution of the paper is creation and introduction of SearchBench, presenting a broad set of combinatorial search problems dataset that extend beyond standard benchmarks. It assesses models based on feasibility, correctness, and optimality, providing a detailed assessment of LLM reasoning abilities in combinatorial problems.

2. The paper illustrates that LLMs often struggles in multi-step reasoning and backtracking tasks. As such, the paper underline major issues in current model capabilities. The challenges in SearchBench reflect real-world applications, such as pathfinding and puzzle-solving, that require systematic search and misstep correction.

3. The authors idea of using A* prompting strategy with a Multi-Stage-Multi-Try (MSMT) approach is interesting and shows substantial improvements relative to prompt based solutions alone.  MSMT’s staged and unit-tested code generation approach improves LLM performance, demonstrating a practical way to improve reasoning on complex tasks.

4. The paper provides evaluation of various large language models (e.g., GPT-4, Llama) and also studies various prompting techniques (e.g., 0-shot, Chain-of-Thought, A* prompting). Hence, the paper shows meaningful comparisons across models and prompt based strategies.

**Weaknesses:**

The main weaknesses of the paper:

1. The paper does not consider recent advance works on multi-step reasoning techniques and code synthesis methods using LLM. Instead, the authors solely use prompt based approaches. It is unclear how those advance methods will perform on the proposed dataset.
2. The paper conclusions may not hold for problems that do not have code based solutions. As such it is limited to certain types of problems that can be solved through coding.
3. The evaluations are non comprehensive. The authors give unfair advantage to their MSMT method as the method has prior knowledge about the type of the code it needs to sysnthesize, i.e., code for A-search algorithm. The evaluations should have included recent LLM works who also can synthesis codes and provided with such prior that the code is for A*search. Only then one can better appreciate the proposed code synthesis method.
4. The scalability of Multi-Stage-Multi-Try (MSMT) method is unclear as the method is complex and computationally demanding. Moreover, the simulation-based experiments lack real-world variability, making it difficult to evaluate how well the proposed methods would generalize to other real world problems.
5. This is less important in my overall rating. But, clearly, SearchBench is centered around combinatorial tasks. Although interesting dataset, it does not support how one could devise LLM methods with other reasoning challenges, such as open-domain problems.

**Questions:**

1. As explained in weakness, the scalability of MSMT is unclear. Can the authors comment as to why this method will not suffer from state space expansion problem when the problem scales to practical scenarios? Since SearchBench is limited to those problems that human can solve correctly, it appears that the scale of the problems are too small for real world problems.
2. It would be helpful to check and compare the performance of MSMT in other combinatorics tasks outside of the authors own SearchBench dataset.
3. The authors need to evaluate also the performance of other more recent methods in LLM reasoning that are not promote based solely, such as multistep reasoning, reward process modeling, planning with world model and deliberate reasoning techniques, on their SearchBench dataset and compare with those of MSMT. It is u fair to limit compare MSMT with prompt based approaches or LLM that is not boosted to synthesis codes. It is well known that LLm cannot do well in code generation for complex reasoning problems unless they are guided through multiple structured steps.
4. The paper could benefit from detailed analysis on error patterns, which could help identify specific failure areas in LLM reasoning and suggest targeted improvements.
5. What if we do not know the type of the problem and hence do not know if the A search algorithm is the solution. This limits the scope of the work. The paper does not address the issue of algorithm selection. In other words, the proposed MSMT method has the prior knowledge about the code type to generate.

---

> ### Author Response · Authors · 2024-11-28
> **Rebuttal**
>
> Thank you for reviewing our paper. Our main goal was to introduce a comprehensive search and planning benchmark for LLMs. The A* MSMT serves as a baseline, showcasing current models' ability to design sophisticated search algorithms, offloading iterations of search to an external interpreter. However, the benchmark itself is our primary contribution and remains challenging for state-of-the-art models like GPT-4 and GPT-01, especially when tasked to solve the problems end-to-end.
>
> >The paper does not consider recent advances in multi-step reasoning and code synthesis using LLMs
>
> In addition to presenting the challenging SearchBench benchmark, our contribution includes demonstrating the application of MSMT A* which highlights LLMs' stronger capability to write a complex search algorithm as opposed to performing non linear reasoning, which involves the iterative computations involved in evaluating each state within the problem's state space, such as calculating cost and heuristic of each node, generating child nodes, etc. If you could provide specific citations for the advanced methods you mentioned, it would help us understand your concerns better. Our MSMT approach aligns with recent advancements that leverage prompting and inference techniques to enhance reasoning.
>
> >The paper conclusions may not hold for problems that do not have code based solutions
>
> The assumption that problems can be solved through code is not overly restrictive. Many real-world problems can be modeled as state-base problems with a start state, end state, and a series of allowed actions and solved using search, allowing for the application of A* search, making our approach broadly applicable.
>
> >The authors give unfair advantage to their MSMT method as the method has prior knowledge about the type of the code it needs to sysnthesize
>
> We have evaluated the model's performance using 0-shot code prompting, where the model was not given any prior information about the problem-solving approach. This approach demonstrates the model's ability to generate solutions without explicit guidance or prior knowledge about how to solve these problems. Providing solved instances as a prompt, as in our A* and MSMT A* approach, is a common technique to enhance reasoning. Our main contribution is the benchmark itself, which remains challenging even with such guidance.
>
> >The scalability of MSMT method is unclear as the method is complex and computationally demanding
>
> The problems in SearchBench are NP-hard, representing some of the most challenging problems in theoretical computer science. These problems inherently require significant computational resources, even when the algorithms are implemented by experts in the field, regardless of the approach used. Our work highlights the current limitations of LLMs in solving such problems end-to-end, emphasizing the need for further research in this area.
>
> > it does not support how one could devise LLM methods with other reasoning challenges, such as open-domain problems
>
> By "open-domain problems," if you mean unbounded state-space problems like board games or simulation worlds, our A* MSMT approach is applicable there as well. In our method, we only assume that we are provided with a start state, an end state, and a set of allowed actions, which can be extracted from natural language or set based on common sense rules. Although our problem types are mostly bounded, their state space can be made arbitrarily large, making our approach generally applicable to such problems.
>
> >The authors need to evaluate also the performance of other more recent methods in LLM reasoning that are not promote based solely, such as multistep reasoning, reward process modeling
>
> Multistep reasoning is incorporated in our 4-shot CoT and A* implementations as comments. Reward-based methods require training both the language model and a reward model which is beyond the scope of our work. Our focus was on introducing a problem set that highlights the current challenges LLMs face in reasoning, and MSMT A* is one of the methods we used to show case that the nonlinearity involved in reasoning is the main bottleneck of LLMs which can be alleviated by using an external interpreter to execute many iterations of the LLM-generated algorithms.
>
> >What if we do not know the type of the problem and hence do not know if the A search algorithm is the solution
>
> A* is the most computationally efficient search algorithm that guarantees finding an optimal solution with an admissible and consistent heuristic. The combinatorial problems in our dataset would take days to solve using breadth-first search, and depth-first search does not guarantee optimal solutions. A* does not require specific assumptions about the problem compared to other search algorithms; it can be applied to any problem where intermediate states can be represented as a graph, with actions as links between nodes.
>
> We hope this clarifies our contributions and the rationale behind our approach.

---

### Official Review · Reviewer_pgfP · 2024-11-05

**Soundness:** 3
**Presentation:** 3
**Contribution:** 2
**Rating:** 3
**Confidence:** 3

**Summary:**

This paper introduces an approach to using LLM to solve search problems by prompting LLM to implement A*.

**Strengths:**

- The proposed method performs better than directly prompting LLMs to solve the problem or generate code to solve the problem.
- The paper also proposes a benchmark set with the hope of avoiding training contamination.

**Weaknesses:**

- Using LLMs to generate the A* implementation sounds like an overkill. One could consider simply prompting LLMs to generate the inputs and heuristic function to an existing A* implementation and then prompting LLMs again to interpret the output of the A* algorithm.
- It seems the paper transforms the effort of implementing the A* algorithm to the effort of implementing a prompting scheme to have LLM generate A*. From this perspective, I don't see a significant motivation to use the method proposed in the paper. On the other hand, if the motivation is to understand the capability of LLMs to solve these types of puzzles, it would be more interesting to consider the scenario where the LLM is not provided a hint about how the problem can be solved (i.e., with A*).

**Questions:**

I would consider an approach that does not resynthesize code that already exists (e.g., like A*) but only prompt LLMs for parameters to the A*.

---

> ### Author Response · Authors · 2024-11-28
> **Rebuttal**
>
> Thank you for your time and effort in reviewing our paper. Our primary focus was to introduce a comprehensive search and planning benchmark for LLMs. The A* MSMT serves as a baseline on this benchmark, demonstrating the current models' ability to design sophisticated search algorithms, offloading the execution of numerous iterations of the algorithm to an external interpreter. However, the benchmark itself is our main contribution, and it remains challenging even for state-of-the-art models like GPT-4 and GPT-01, especially when tasked with solving problems end-to-end.
>
>
> >Using LLMs to generate the A* implementation sounds like an overkill. One could consider simply prompting LLMs to generate the inputs and heuristic function to an existing A* implementation and then prompting LLMs again to interpret the output of the A* algorithm.
>
> Implementations of the A* search algorithms cannot be reduced to a few parameters; the algorithms must be constructed for each unique problem type. Each problem requires fundamental changes to the A* algorithm that are unique to each problem type, including how to construct the search graph, represent actions as nodes, select child nodes, and calculate costs. Our prompts, detailed in the Appendix, demonstrate that there is little in common between different A* implementations beyond the general structure of the algorithm. These differences extend beyond the heuristic function, making each implementation unique and unsuitable for reuse across different problem types in our benchmark.
>
> > if the motivation is to understand the capability of LLMs to solve these types of puzzles, it would be more interesting to consider the scenario where the LLM is not provided a hint about how the problem can be solved (i.e., with A*).
>
> We explored various zero-shot prompting schemes where no hint is provided to the model on how to solve the problems, such as zero-shot text and zero-shot code (Please refer to Figure 3 and teh methods and Experiments sections). The motivation for using A* was to highlight the challenges LLMs face with nonlinear reasoning. While these problems are straightforward for humans, requiring only basic algebra, LLMs struggle to solve them end to end due to the need for multiple iterations of simple computations. By prompting the model to write an A* algorithm, we offload executing many iterations of the search algorithm to an external engine, allowing the LLM to focus on a complex task that needs to be executed once rather than repeatedly.
>
> We hope this clarifies our contributions and the rationale behind our approach. Thank you for your feedback.

---

### Note · Authors · 2024-12-15

I have read and agree with the venue's withdrawal policy on behalf of myself and my co-authors.